# Rbfox1 is required for myofibril development and maintaining fiber type–specific isoform expression in *Drosophila* muscles

Elena Nikonova[1,*], Amartya Mukherjee[2,*], Ketaki Kamble[2,*], Christiane Barz[3], Upendra Nongthomba[2], Maria L Spletter[1]

**Protein isoform transitions confer muscle fibers with distinct properties and are regulated by differential transcription and alternative splicing. RNA-binding Fox protein 1 (Rbfox1) can affect both transcript levels and splicing, and is known to contribute to normal muscle development and physiology in vertebrates, although the detailed mechanisms remain obscure. In this study, we report that Rbfox1 contributes to the generation of adult muscle diversity in *Drosophila*. Rbfox1 is differentially expressed among muscle fiber types, and RNAi knockdown causes a hypercontraction phenotype that leads to behavioral and eclosion defects. Misregulation of fiber type–specific gene and splice isoform expression, notably loss of an indirect flight muscle–specific isoform of Troponin-I that is critical for regulating myosin activity, leads to structural defects. We further show that Rbfox1 directly binds the 3′-UTR of target transcripts, regulates the expression level of myogenic transcription factors myocyte enhancer factor 2 and Salm, and both modulates expression of and genetically interacts with the CELF family RNA-binding protein Bruno1 (Bru1). Rbfox1 and Bru1 co-regulate fiber type–specific alternative splicing of structural genes, indicating that regulatory interactions between FOX and CELF family RNA-binding proteins are conserved in fly muscle. Rbfox1 thus affects muscle development by regulating fiber type–specific splicing and expression dynamics of identity genes and structural proteins.**

## Introduction

Muscles are an ideal model to understand the strategies involved in the generation of diversity within a tissue, as they are developmentally patterned with distinct morphologies and diverse contractile properties (Spletter & Schnorrer, 2014). Muscle fiber types

are heterogeneous, displaying protein isoform-specific signatures that impart distinctive functionalities to meet diverse physiological demands (Armstrong & Phelps, 1984; Bottinelli & Reggiani, 2000; Bottinelli, 2001; Schiaffino & Reggiani, 2011; Schiaffino et al, 2020). Composite muscle fiber profiles are a result of coordinated regulation of gene expression (Firulli & Olson, 1997; Black & Olson, 1998; Majesky, 2007), co-integrated with protein isoform transitions facilitated by alternative splicing (Smith et al, 1989; Guo et al, 2010; Kalsotra & Cooper, 2011; Nikonova et al, 2020), accompanied by post-translational modifications (Anthony et al, 2002; Michele & Campbell, 2003; Wells et al, 2003; Nayak & Amrute-Nayak, 2020). The underlying molecular changes are initially regulated by the intrinsic developmental program (Firulli & Olson, 1997; Kablar & Rudnicki, 2000), and later modulated by nerve stimulation, physiological demands, and pathophysiological conditions (Hughes et al, 1993; Pette & Staron, 2001; Schiaffino et al, 2007; Pistoni et al, 2010). The process of protein isoform expression needs to be tightly regulated to have a functionally relevant outcome (Smith et al, 1989; Firulli & Olson, 1997; Black & Olson, 1998; Anthony et al, 2002; Majesky, 2007; Guo et al, 2010; Kalsotra & Cooper, 2011).

Protein isoform expression is regulated by a diverse array of RNA-binding proteins (RBPs). RBPs regulate the process of alternative splicing by binding to *cis*-intronic or -exonic elements in target RNAs to generate the splicing profile of a given cell type (Kalsotra & Cooper, 2011; Fu & Ares, 2014). RBPs can also regulate translation level by binding to 3′-UTR elements and subsequently associating with translation factors, P-granules, or components of the nonsense-mediate decay pathway (Hentze et al, 2018; Kishor et al, 2019; Ho et al, 2021). RBPs are thus key mediators of eukaryotic genome information during development, and are essential for establishing, refining, and maintaining tissue and fiber type–specific properties (Lunde et al, 2007; Spletter & Schnorrer, 2014; Nikonova et al, 2019; Grifone et al, 2020). The salience of RBP function is illustrated by observations that alternative splicing and protein isoform expression patterns are disrupted in diseases from

[1]Department of Physiological Chemistry, Biomedical Center, Ludwig-Maximilians-Universität München, Martinsried-Planegg, Germany  [2]Department of Molecular Reproduction, Development and Genetics (MRDG), Indian Institute of Science, Bangalore, India  [3]Muscle Dynamics Group, Max Planck Institute of Biochemistry, Martinsried-Planegg, Germany

Correspondence: maria.spletter@bmc.med.lmu.de; upendra@iisc.ac.in
*Elena Nikonova, Amartya Mukherjee, and Ketaki Kamble contributed equally to this work.

cardiomyopathy to cancer (Ravanidis et al, 2018; Bessa et al, 2020; Picchiarelli & Dupuis, 2020), and that loss of RBP function leads to severe neuromuscular disorders, such as myotonic dystrophy, amyotrophic lateral sclerosis, and spinal motor atrophy (Nikonova et al, 2019; López-Martínez et al, 2020; Picchiarelli & Dupuis, 2020). It is therefore critically important to understand RBP function in detail.

RNA-binding Fox protein 1 (Rbfox1, the first identified member of the FOX family of RBPs) is an RBP with a single, highly conserved RNA recognition motif domain that binds to 5′-UGCAUG-3′ motifs (Jin et al, 2003; Auweter et al, 2006). Rbfox1 binding to introns causes context-dependent exon retention or skipping, depending on if it binds upstream or downstream of an alternative exon (Nakahata & Kawamoto, 2005; Fukumura et al, 2007), whereas 3′-UTR binding is reported to modulate mRNA stability (Carreira-Rosario et al, 2016). Rbfox1 may additionally influence transcription networks by binding transcriptional regulators (Usha & Shashidhara, 2010; Wei et al, 2016; Shukla et al, 2017). In vertebrates, Rbfox1 has been shown to either cooperatively or competitively regulate splicing with other RBPs, such as SUP-12, ASD-1, MBNL1, NOVA, PTBP, CELF1/2, and PSF (Klinck et al, 2014; Conboy, 2017), as well as to be involved in cross-regulatory interactions with CELF and MBNL family proteins (Gazzara et al, 2017; Sellier et al, 2018). This context-dependent nature of Rbfox1 function, as well as integration with other splicing networks and the conservation of such regulatory interactions, is not yet fully understood.

Rbfox1 plays an important role in regulating the development of both neurons and muscle (Conboy, 2017). In vertebrates, Rbfox1 is necessary for proper neuronal migration and axonal growth (Hamada et al, 2016), is induced by stress (Amir-Zilberstein et al, 2012), and modulates the splicing of genes involved in axonal depolarization (Lee et al, 2009; Gehman et al, 2011). Rbfox1 was recently shown to regulate sensory neuron specification in Drosophila (Shukla et al, 2017) and brain development in the mosquito Aedes aegypti (Mysore et al, 2021), suggesting its function in neuronal development is conserved. In vertebrate muscle, Rbfox1 binding sites are enriched around developmentally regulated, alternatively spliced exons in the heart (Kalsotra et al, 2008), and Rbfox1 regulates alternative splicing of structural proteins as well as proteins in the calcium signaling pathway in skeletal muscle (Pedrotti et al, 2015). This function is disease relevant, as Rbfox1-mediated splicing is implicated in the regulation of cardiac failure (Gao et al, 2016), and Rbfox is down-regulated in the mouse model of Facio-scapulo-humeral dystrophy (Pistoni et al, 2010). Moreover, Rbfox1-mutant mice display myofiber and sarcomeric defects and impaired muscle function (Pedrotti et al, 2015), and Rbfox1 is necessary for maintaining skeletal muscle mass (Singh et al, 2018). Because of these pleiotropic phenotypes, and often multi-layered regulatory mechanisms, the exact role of Rbfox1 in muscle development and physiology is still not fully understood. Moreover, the interpretation of mutant phenotypes and regulatory interactions is complicated by the heterogeneous fiber type composition of vertebrate muscles, and the presence of other FOX family members, notably Rbfox2 (Conboy, 2017; Singh et al, 2018; Begg et al, 2020), that have similar functions.

Invertebrate models with simpler, less redundant genomes, such as Drosophila or Caenorhabditis elegans, are powerful systems to investigate conserved, in vivo functions of RBPs in muscle (Nikonova et al, 2019). Muscle structure, as well as the mechanism of actomyosin contractility, is highly conserved (Lemke & Schnorrer, 2017; Dasbiswas et al, 2018), and studies of alternative splicing regulation and fiber type–specific protein isoform function have proven highly informative (Plantié et al, 2015; Jagla et al, 2017; Jawkar & Nongthomba, 2020). Although the C. elegans homolog FOX1 has been shown to result in egg-laying defects and regulate a muscle-specific splice event in egl-15 (Kuroyanagi et al, 2006), the role of Rbfox1 in Drosophila muscle remains largely unknown. The Drosophila genome contains a single copy of the Rbfox1 gene (also known as Ataxin-2–binding protein 1, A2BP1) (Kuroyanagi, 2009). We previously reported that RNAi-mediated knockdown of Rbfox1 leads to a loss of flight and short sarcomeres in flight muscle (Nikonova et al, 2019), motivating our present work to explore the detailed role of Rbfox1 in regulating muscle development in flies. Drosophila muscles are of two major types, fibrillar and tubular. The asynchronous, stretch-activated fibrillar indirect flight muscles (IFMs), comprising the dorsal longitudinal (DLMs) and dorso-ventral muscle groups, are physiologically similar to vertebrate cardiac muscles (Pringle, 1981; Peckham et al, 1990; Swank et al, 2006). Tubular muscles, constituting all other body muscles in the fly, are synchronous and resemble vertebrate skeletal muscle (de la Pompa et al, 1989; Nikonova et al, 2020). Drosophila muscles also have a uniform fiber type within a muscle fascicle (Bernstein et al, 1993; Spletter & Schnorrer, 2014), precluding the complication of heterogeneous muscle fiber composition typical of mammalian muscles.

In this study, we present the first detailed investigation of the role of Rbfox1 in sculpting the diversity and function of the Drosophila adult musculature. We show that Rbfox1 plays a conserved role in development of both fibrillar and tubular muscle fiber types. Impairment of Rbfox1 function in the IFMs causes muscle hyper-contraction resulting from the mis-splicing and the stoichiometric imbalance of structural proteins, such as Troponin-I (TnI). We present evidence that Rbfox1 regulates fiber type–specific isoform expression on multiple levels. It regulates mRNA transcript levels through direct 3′-UTR binding, as well as indirectly through regulation of transcription factors, including spalt major (Salm) and Myocyte enhancer factor 2 (Mef2), identifying a novel link between RNA regulation and transcriptional refinement of fiber type identity in Drosophila muscle. Rbfox1 further exhibits level-dependent, cross-regulatory interactions with Salm as well as the CELF family RBP Bruno1 (Bru1). Rbfox1 and Bru1 genetically interact in IFMs, and co-regulate alternative splicing of fiber type–specific events in structural genes. Our results demonstrate the conservation of an ancient regulatory network between FOX and CELF family proteins in muscle, and establish a central role for Rbfox1 in fiber type–specific RNA regulation in Drosophila myogenesis.

## Results

### Rbfox1 is differentially expressed between tubular and fibrillar muscles

To evaluate the expression pattern of Rbfox1 in Drosophila muscle, we used the protein trap Rbfox1[CC00511] (Rbfox1-GFP) fly line (Kelso et al, 2004) to track GFP-tagged Rbfox1 protein expression. We observed GFP signal in cells associated with the hinge region of third instar larvae (L3) wing discs (Fig 1A), confirming a previous finding of

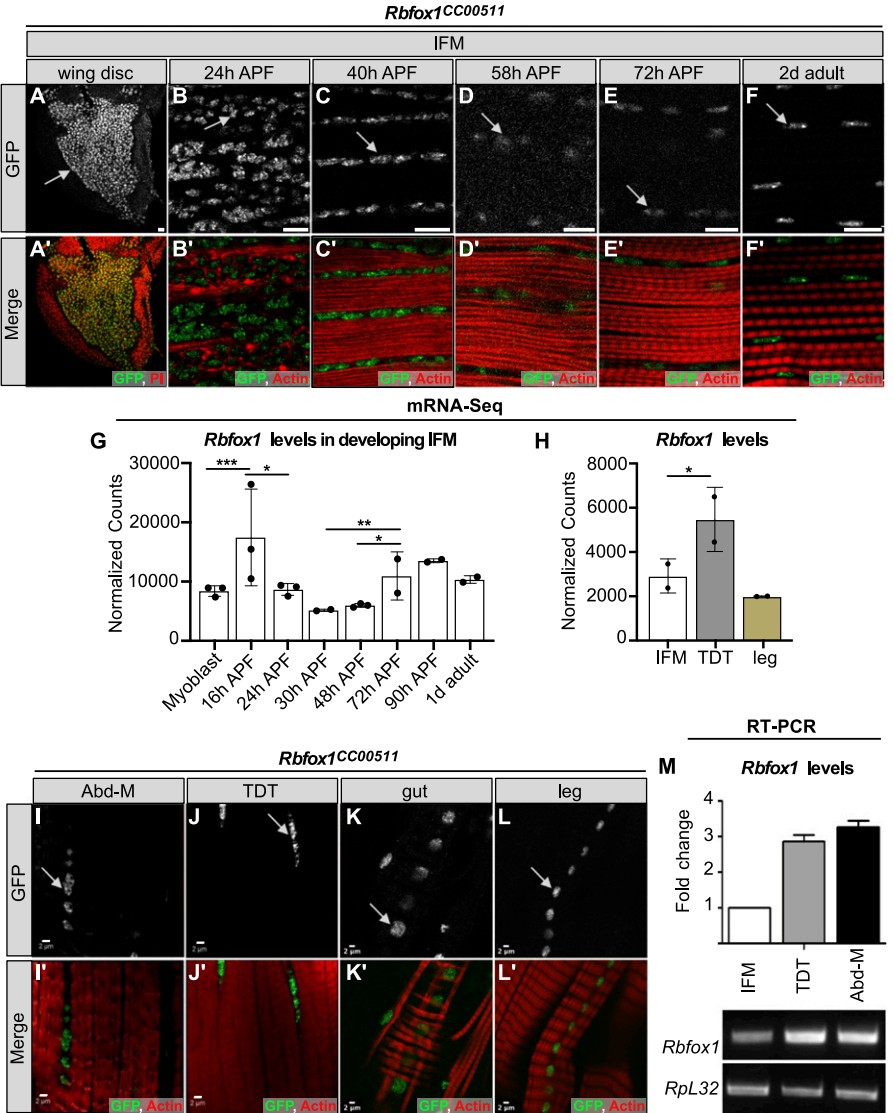

**Figure 1. Rbfox1 is differentially expressed between fibrillar and tubular muscle.**
**(A, B, C, D, E, F)** The *Rbfox1^CC00511^* (Rbfox1-GFP) protein trap line was used to study expression of Rbfox1. **(A, A')** Wing discs of L3 larvae (propidium iodide, red). **(B, B')** Indirect flight muscles (IFMs) at 24 h after puparium formation (APF) show Rbfox1 expression in completely split templates. **(C, C')** IFMs at 40 h APF with Rbfox1 expression during initiation of assembly of sarcomere structure. **(D, D', E, E')** IFMs at 58 and 72 h APF during sarcomere maturation. **(F, F')** Rbfox1 is expressed in 2-d-old adult IFMs. (Arrows indicate GFP positive nuclei. GFP, green; phalloidin-stained actin, red; Scale bars = 10 μm.). **(G, H)** mRNA-Seq data from *w^1118^* reported as normalized counts show differential expression of *Rbfox1* across IFM development (G) and between 1 d adult fiber types (H). Significance calculated with DESeq2 (*$P < 0.01$, **$P < 0.001$, ***$P < 0.0001$). **(I, J, K, L)** Confocal microscopy of the *Rbfox1*–GFP (*Rbfox1^CC00511^*) line shows Rbfox1 expression in adult tubular muscles including abdominal muscles, tergal depressor of the trochanter, gut and leg. **(I', J', K', L')** Merged channel images in I', J', K' and L' show GFP in green and phalloidin-stained actin in red. Scale bars = 2 μm. **(M)** qPCR and representative semi-quantitative gel images showing relative expression of *Rbfox1* at the mRNA level in adult *Canton-S* across muscle fiber types. *RpL32*, also known as *RP49*, was used as a normalizing control.
Source data are available online for this figure.

Rbfox1 expression in myoblasts (Usha & Shashidhara, 2010). In pupae, Rbfox1 protein is detectable in IFM nuclei at all stages of adult myofiber development: at 24 h after puparium formation (APF) in IFMs undergoing splitting and myoblast fusion (Fig 1B), at 40 h APF during sarcomere assembly (Fig 1C), at 58 and 72 h as sarcomeres undergo maturation (Fig 1D and E), and in 2-d-old adult IFMs (Fig 1F). We also detect continual expression of *Rbfox1* in IFMs at the RNA level based on mRNA-Seq data (Fig 1G). Interestingly, we observed a dip in Rbfox1 expression levels from 50 to 72 h APF in the middle of IFM development at both the protein and the mRNA levels.

We next examined Rbfox1 expression in other types of somatic muscle. Rbfox1-GFP can be detected in the nuclei of all muscles examined, including the tubular abdominal muscles (Abd-M), the tergal depressor of the trochanter (TDT or jump muscle), the gut, and the leg muscles (Fig 1I–L). Likewise, *Rbfox1* mRNA is detected in all muscles tested, including IFM, TDT, legs, and abdomen (Figs 1H

and M and S1A and C). *Rbfox1* mRNA is expressed at significantly higher levels in tubular TDT than in fibrillar IFMs, as revealed by mRNA-Seq (Fig 1H) and RT–PCR (Figs 1M and S1C), and displays preferential exon use between these two fiber types (Fig S1B). As leg muscle and Abd-M samples cannot be dissected to the same purity as IFM and TDT, mRNA levels in these samples may not accurately represent muscle-specific *Rbfox1* expression. Taken together, these data demonstrate that although *Rbfox1* is expressed in all types of muscle in *Drosophila*, the expression level is regulated both in a temporal and muscle type–specific manner.

## Rbfox1 function in muscle is necessary for viability and pupal eclosion

To evaluate Rbfox1 function in muscle development, we tested several tools to reduce Rbfox1 levels. We used the deGradFP system, which was developed to specifically target GFP-fused proteins

(Caussinus et al, 2012), to knockdown *Rbfox1*$^{CC00511}$ (Rbfox1-GFP). We also used three UAS-RNAi (IR) hairpins targeting *Rbfox1*, including *Rbfox1*-RNAi (Usha & Shashidhara, 2010), *Rbfox1*-IR$^{27286}$, and *Rbfox1*-IR$^{KK110518}$ (Nikonova et al, 2019) (Fig S1A). As detailed below, these tools produced different levels of knockdown, and phenotypes of different severity. We used additional genetic manipulations to exercise temporal and spatial control to evaluate Rbfox1 phenotypes in select muscles with different levels of Rbfox1 knockdown.

We started by inducing deGradFP using the constitutive muscle driver Mef2-Gal4, which resulted in pupal lethality (Fig 2A and C), and complete loss of GFP signal in muscle (Fig S2J). To reduce the strength of this knockdown, we combined our deGradFP flies with *tubulin-Gal80*$^{ts}$, and temperature shifted from 18°C to 29°C at late L3, but we still observed pupal lethality (Fig 2A). This result indicates that Rbfox1 is required in tubular muscle, as the IFMs are not required for eclosion or viability. To obtain viable adult flies that we could use for further experiments, we next tested three different UAS-RNAi reagents. We found that Mef2-Gal4 driven knockdown with *Rbfox1*-IR$^{KK110518}$ was pupal lethal, and larval lethal when driven with Act5c-Gal4, which expresses in all cells, or when combined with UAS-Dicer2 (Dcr2), which is reported to increase the efficiency of RNAi knockdown (Dietzl et al, 2007) (Fig 2A and B). *Rbfox1* mRNA levels were significantly reduced in *Rbfox1*-IR$^{KK110518}$ IFMs (Fig S1E). Although *Rbfox1* mRNA levels were also reduced significantly in *Rbfox1*-RNAi IFMs (Fig S1D), the phenotype of Mef2-Gal4 driven *Rbfox1*-RNAi was less severe, and around 70% of pupae were able to eclose (Fig 2A). *Rbfox1*-IR$^{27286}$ was the weakest hairpin, as more than 80% of flies eclosed when crossed to the universal Act5c-Gal4 driver or the constitutive muscle driver Mef2-Gal4 (Fig 2A). *Rbfox1* mRNA levels were not decreased significantly in *Rbfox1*-IR$^{27286}$ IFMs, but were significantly decreased when combined with Dcr2 (Fig S1E). Dcr2, *Rbfox1*-IR$^{27286}$ flies were pupal lethal at 25°C and 27°C, but eclosed at 22°C (Fig 2B). As RNAi efficiency increases with temperature, this result proves that phenotypic severity depends on the strength of *Rbfox1* knockdown. We thus are able to tune the expression level of Rbfox1 in muscle, and established a knockdown series ordered from the strongest to the weakest phenotype: deGradFP > *Rbfox1*-IR$^{KK110518}$ > *Rbfox1*-RNAi > *Rbfox1*-IR$^{27286}$. We conclude that Rbfox1 function in muscle is required for viability, as the strongest muscle-specific knockdown conditions resulted in early lethality. Rbfox1 is further required during adult muscle development, as weaker knockdown efficiencies resulted in pupal lethality, notably because of eclosion defects.

### Rbfox1 contributes to tubular muscle development and function

To determine if Rbfox1 is required in tubular muscles, as suggested by the eclosion defect, we investigated tubular muscle structure and function. We first assayed climbing ability by evaluating how many adult flies were able to climb 5 cm in 3 s. We tested *Rbfox1*-IR$^{27286}$ flies driven with Act5c-Gal4 and Mef2-Gal4 at 27°C, and with UAS-Dcr2, Mef2-Gal4 at 22°C, as we could obtain surviving adults from these conditions. Flies with reduced *Rbfox1* levels were poor climbers (Fig 2D), indicating impairment in tubular leg muscle function. We did not observe climbing defects when we performed knockdown with Act88F-Gal4 (Fig 2D), which is largely restricted to the fibrillar flight muscles. To assess functional defects in tubular

TDT muscle, we then assayed jumping ability by measuring the distance a startled fly can jump. Decreased levels of *Rbfox1* severely impaired jumping ability (Fig 2E), whereas control flies on average jumped a distance of around 2 cm, knockdown flies jumped under 1 cm. We also saw defective jumping in Act88F-Gal4–driven *Rbfox1* knockdown, and phenotypic severity was dependent on the strength of knockdown (Fig 2E). This reflects weak expression of the driver in jump muscle (Kao et al, 2021). Together, these data indicate that a decrease in *Rbfox1* levels results in behaviour defects associated with impaired tubular muscle function.

We next examined tubular muscle structure using confocal microscopy. We observed severe disruptions in sarcomere and myofibril structure in both TDT and Abd-M, depending on the strength of *Rbfox1* knockdown (Figs. 2F–O and S1F–O). TDT myofibrils were frayed and severely disorganized after knockdown with all three RNAi hairpins (Figs. 2F–J and S1H–J). Although nuclei were still organized in the center of the TDT myofibers, the cytoplasmic space between the nuclei was often invaded by myofibrils in knockdown conditions (compare Fig S1F to Fig S1H–J). In the most strongly affected samples, TDT fibers were atrophic and severely degraded (Fig S1P). The TDT sarcomeres were significantly shorter in 1 d adult flies with Mef2-Gal4 driven *Rbfox1*-IR$^{27286}$ (2.11 ± 0.21 µm versus 2.71 ± 0.19 µm in control, *P*-value < 0.001) and this was enhanced in the presence of Dcr2 (1.76 ± 0.31 µm versus 2.98 ± 0.26 µm in control, *P*-value < 0.001). However, sarcomeres were not significantly shorter at 90 h APF with Mef2-Gal4–driven *Rbfox1*-IR$^{KK110518}$ (2.43 ± 0.27 µm versus 2.52 ± 0.24 µm in control, *P*-value = 0.7413) (Fig 2P). Similar to this progressive shortening of TDT sarcomeres we observe between 90 h APF and 1 d adults, classic hypercontraction mutants in IFMs display a temporal phenotype where misregulated myosin activity leads to sarcomere shortening after eclosion (Nongthomba et al, 2003), suggesting that loss of *Rbfox1* leads to a hypercontraction phenotype in TDT.

We observed similar defects in Abd-M sarcomere and myofibril structure after *Rbfox1* knockdown (Figs. 2K–O and S1K–O). Knockdown with *Rbfox1*-RNAi during adult muscle development led to loss of sarcomere architecture (Fig 2L). In *Rbfox1*-IR$^{27286}$ and *Rbfox1*-IR$^{KK110518}$ knockdown animals, Abd-M myofibers were often torn (Fig 2M–O) or degraded (Fig S1Q). Myofibrils were disorganized, at times invading the center of the fiber (compare Fig S1K to Fig S1M–O), and laterally aligned Z-discs were poorly organized (Fig 2M–O). Abd-M sarcomeres in 1 d adults with Dcr2, Mef2-Gal4 driven *Rbfox1*-IR$^{27286}$ were significantly shorter than controls (2.99 ± 0.64 µm versus 3.70 ± 0.47 µm in control, *P*-value < 0.001), and were already significantly shorter at 90 h in Mef2-Gal4 driven *Rbfox1*-IR$^{KK110518}$ (2.71 ± 0.83 µm versus 3.74 ± 0.64 µm in control, *P*-value < 0.001) (Fig 2Q). Overall, the observed phenotypes in tubular TDT and Abd-M are consistent between independent RNAi hairpins, and phenotypic severity increases with increasing strength of *Rbfox1* knockdown. Taken together, the defects in eclosion, climbing, jumping, and tubular myofiber morphology indicate a general requirement for Rbfox1 in tubular muscle development.

### Knockdown of *Rbfox1* leads to hypercontraction in the IFMs

We next evaluated Rbfox1 function in fibrillar IFMs. Surviving *Rbfox1*-RNAi adults are completely flightless (Fig 3A), and surviving

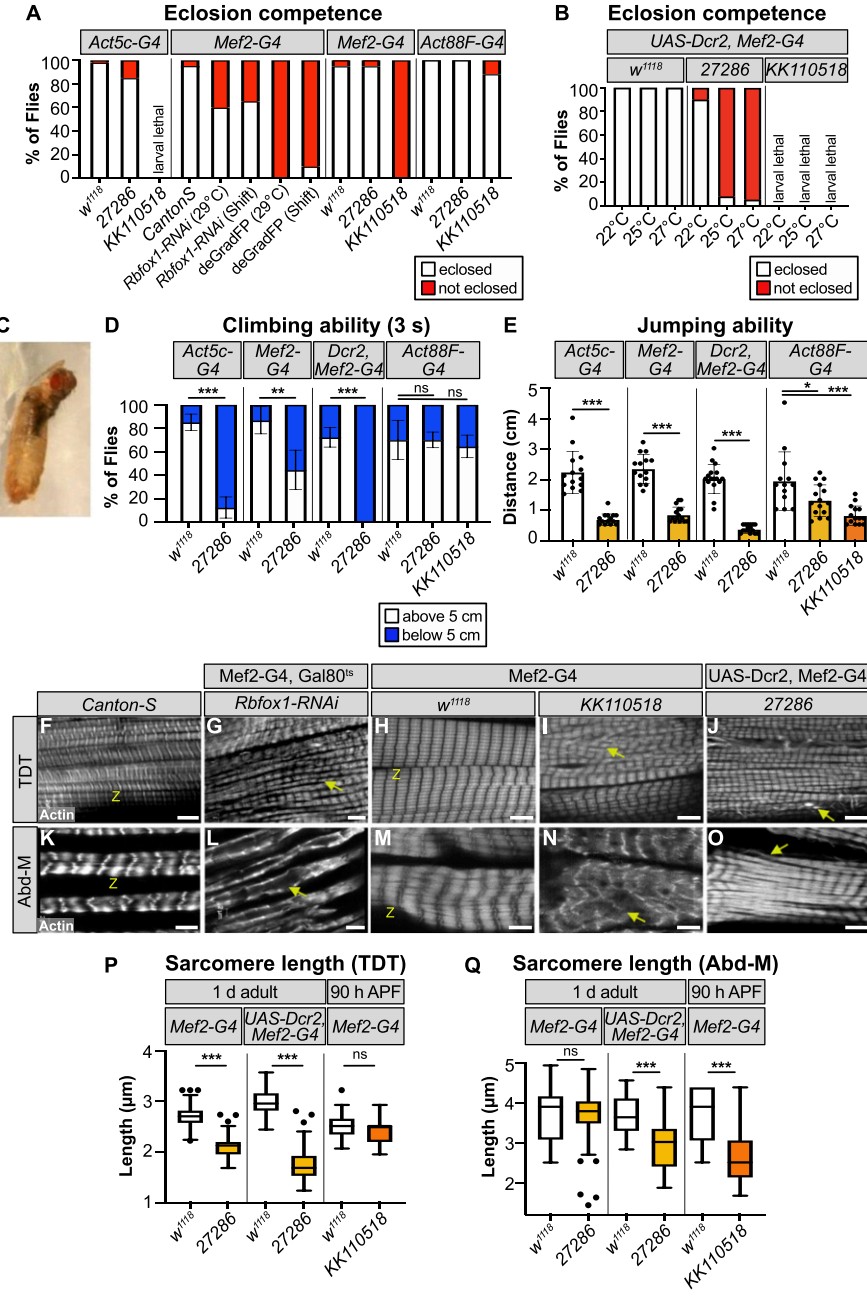

**Figure 2. Rbfox1 is necessary for tubular tergal depressor of the trochanter (TDT) and abdominal muscle (Abd-M) development.**
**(A)** Quantification of the percent of pupae that eclose for controls and *Rbfox1* knockdown flies. Genotypes as labeled. **(B)** Quantification of the percent of pupae that eclose for UAS-Dcr2, Mef2-Gal4–driven *Rbfox1-IR*[27286] and *Rbfox1-IR*[KK110518] knockdown at 22°C, 25°C, and 27°C. **(C)** Representative image of the eclosion defect in *Rbfox1*-RNAi. **(D)** Quantification of climbing ability measured by how many flies are able to climb 5 cm in 3 s. **(E)** Quantification of jumping ability measured as the distance in cm a startled fly can jump. **(D, E)** Error bars in (D, E) show SD. **(F, G, H, I, J, K, L, M, N, O)** Single-plane confocal images showing myofibril and sarcomere morphology of the TDT (F, G, H, I, J) and Abd-M (K, L, M, N, O). **(G, I, J, L, N, O)** Myofibril structure is altered in *Rbfox1* knockdown conditions, including disorganized myofibril structure (arrow in G, I), frayed myofibrils (arrow in J, O), and loss of sarcomere architecture (arrow in L, N). "Z" indicates z-discs. Scale bars = 5 μm. **(P, Q)** Quantification of sarcomere length in TDT (P) and Abd-M (Q). Boxplots are shown with Tukey whiskers, with outlier data points marked as dots. **(D, E, P, Q)** Significance in (D, E, P, Q) determined by ANOVA and post-hoc Tukey (ns, not significant; *P < 0.05; **P < 0.01; ***P < 0.001).
Source data are available online for this figure.

adults from all *Rbfox1*-IR[27286] crosses, as well as flies with IFM-restricted, Act88F-Gal4 driven *Rbfox1*-IR[KK110518] had impaired flight ability (Fig 3B), which is in agreement with our previous results (Nikonova et al, 2019). The strength of the flight defect increased with the strength of *Rbfox1* knockdown and was consistent across all three RNAi hairpins tested.

To determine if the impaired flight was the result of defective muscle structure or function, we examined IFMs using confocal microscopy. Myofibers in thoraxes of 1-d-old (1 d) adult *Rbfox1*-IR[27286] flies or 90 h APF *Rbfox1*-IR[KK110518] flies were frequently torn and detached (Fig 3C–F). Myofibrils in the remaining intact DLM myofibers were frayed and wavy (Fig 3C'–E'). Sarcomere length was

significantly shorter in 1-d adult flies with both Mef2 > *Rbfox1*-IR[27286] (2.90 ± 0.24 μm versus 3.34 ± 0.20 μm in control, *P*-value < 0.001) and with UAS-Dcr2, Mef2-Gal4 enhanced knockdown (2.98 ± 0.33 μm versus 3.43 ± 0.16 μm in control, *P*-value < 0.001) (Figs 3G and S2A). Myofibril width in Mef2 > *Rbfox1*-IR[27286] IFMs was significantly thicker than control (1.58 ± 0.25 μm versus 1.18 ± 0.11 μm in control, *P*-value < 0.001) (Figs 3H and S2B). Myofibril width was actually thinner with UAS-Dcr2, Mef2-Gal4 enhanced knockdown in 1-d adults (0.92 ± 0.22 μm versus 1.14 ± 0.12 μm in control, *P*-value < 0.001), reflecting the increased severity of myofibril fraying and loss. At 90 h APF, sarcomeres of *Rbfox1*-IR[27286] flies were not significantly shorter than the control, but myofibrils were significantly thicker (Figs 3G and S2A

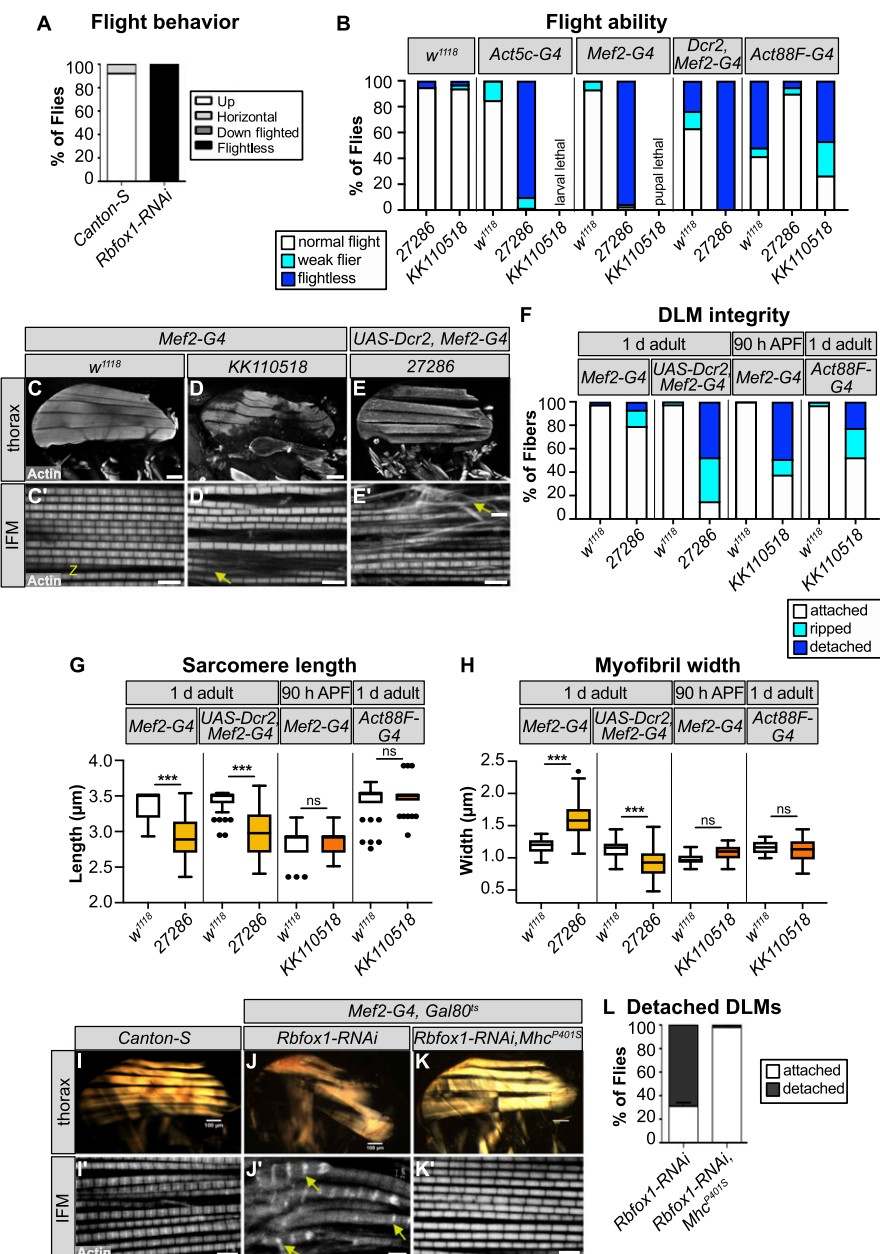

**Figure 3. *Rbfox1* knockdown results in indirect flight muscle (IFM) myofibril defects and hypercontraction-mediated myofiber loss.**
**(A, B)** Quantification of flight ability after *Rbfox1* knockdown. Genotypes as noted. **(C, C', D, D', E, E')** Confocal Z-stack images (C, D, E) of IFM myofiber structure (Scale bars = 5 μm) and single-plane images (C', D', E') of myofibril and sarcomere structure after *Rbfox1* knockdown. Arrows mark examples of frayed or torn myofibrils (arrow in D', E'). **(C, D, E, F)** Quantification of myofiber ripping and detachment phenotypes in (C, D, E). **(C', D', E', G, H)** Quantification of IFM sarcomere length and myofibril width in (C', D', E'). Boxplots are shown with Tukey whiskers, with outlier data points marked as dots. Significance determined by ANOVA and post hoc Tukey (ns, not significant; *$P < 0.05$; **$P < 0.01$; ***$P < 0.001$). **(I, I', J, J', K, K')** Polarized microscopy images (I, J, K) and single-plane confocal images (I', J', K') of hemithorax from wild-type (I, I'), *Rbfox1*-RNAi (J, J') and *Rbfox1*-RNAi, *Mhc^P401S* (K, K') flies. The *Mhc^P401S* allele suppresses myofiber loss and sarcomere phenotypes. **(J')** Arrows in (J') indicate zebra bodies. **(J, K, L)** Quantification of myofiber detachment in (J, K).
Source data are available online for this figure.

and B). Myofibrils in Act88F-Gal4–mediated knockdown only showed mild defects (Figs 3G and H and S2C and D), despite adult flies being flight impaired. Like we observed in TDT, this progressive shortening of IFM sarcomeres after eclosion is suggestive of a hypercontraction phenotype, which is supported by the Act88F-Gal4 result indicating that the regulation of actomyosin dynamics in *Rbfox1* knockdown muscle is disrupted.

We further confirmed the IFM defects with *Rbfox1*-RNAi and *Rbfox1^CC00511*-deGradFP. When we assessed DLMs of the few *Rbfox1^CC00511*-deGradFP escapers, we saw tearing or detachment of muscle fibers (Fig S2F–H) and defective patterning of the DLM myofibrils, including actin accumulations and sarcomeric defects (Fig S2I and J). We visualized DLM fibers from *Rbfox1*-RNAi adult flies

under polarized light and also observed tearing and loss of muscle fibers (Fig 3I, J, and L). Sarcomere cytoarchitecture was severely disrupted, accompanied by the appearance of actin accumulations at the Z-discs, also known as Zebra bodies (Fig 3I' and J'). Over-expression of *Rbfox1* with Mef2-Gal4 was lethal, but temporally and spatially restricted overexpression of *Rbfox1* from 40 h APF using the IFM-specific UH3-Gal4 (Singh et al, 2014) resulted in an IFM phenotype similar to the knockdown, including torn myofibers (Fig S2E) and thin, frayed, or torn myofibrils with short sarcomeres (Fig S2E'). The consistency in phenotype between all three RNAi hairpins and *Rbfox1^CC00511*-deGradFP, as well as the increased phenotypic severity with stronger knockdown, indicate that Rbfox1 is required for IFM development. Moreover, the decrease in sarcomere length

with a corresponding increase in myofibril width in 1 d old adults suggests that loss of Rbfox1 results in a hypercontraction phenotype. Interestingly, both *Rbfox1* knockdown and Rbfox1 overexpression produce similar hypercontraction defects.

Hypercontraction is caused by misregulated actomyosin interactions, which can result from many factors, including mutations in structural proteins, mechanical stress, stoichiometric imbalance, and mis-expression of structural protein isoforms (Nongthomba et al, 2003, 2004, 2007; Firdaus et al, 2015). These misregulated actomyosin interactions can be suppressed by a myosin heavy chain allele ($Mhc^{P401S}$) that minimizes the force produced by actomyosin interactions (Nongthomba et al, 2003). Including the $Mhc^{P401S}$ allele in the *Rbfox1*-RNAi knockdown background restored the structure of IFM myofibers (Fig 3K and L) and sarcomeric cytoarchitecture (Fig 3K'), confirming that the *Rbfox1* knockdown phenotype indeed resulted from muscle hypercontraction.

### Bioinformatic identification of Rbfox1 motif instances in muscle genes

To gain insight into the underlying cause of the sarcomere phenotype, we sought to identify Rbfox1 target genes in muscle. Both vertebrate and *Drosophila* Rbfox proteins are known to recognize (U)GCAUG motifs, and regulate alternative splicing and mRNA transcript stability (Carreira-Rosario et al, 2016), and in vertebrates the Rbfox1 motif is over-represented in introns flanking muscle-specific exons (Brudno et al, 2001). As there are no RNA CLIP data available from *Drosophila* muscle, we bioinformatically identified Rbfox1 motif instances in the transcriptome using oRNAment (Bouvrette et al, 2020), and genome-wide using PWMScan (Ambrosini et al, 2018). Many genes expressed in muscle, notably transcription factors and sarcomere proteins, contain Rbfox1 motifs (Fig S3A and Table S1). These motifs are distributed across intron and coding DNA sequence (CDS) regions (Fig S3C), signifying possible alternative splicing targets, as well as in 5'-UTR and 3'-UTR regions, which may indicate direct regulation of mRNA stability, trafficking or translation. Genes with Rbfox1 motif instances are enriched for gene ontology terms related to transcription, muscle development and cytoskeletal organization, for example, "transcription regulator activity," "motor activity," "developmental process," "muscle structure development," and "actin filament-based process" (Fig S3B and Table S2). We also see enrichment for terms such as "synapse organization," "behavior," and "locomotion," likely reflecting the characterized roles for Rbfox1 in neuronal development (Gehman et al, 2011). This indicates that genes important for muscle development are likely targets of Rbfox1 regulation, but genome-wide confirmation of bound motifs awaits future RNA CLIP studies. We next selected candidate Rbfox1 target genes to verify based on their direct or indirect involvement in muscle contraction, which could explain the sarcomere defects and misregulated actomyosin interactions in the *Rbfox1* knockdown condition.

### Rbfox1 regulates the expression of structural proteins TnI and Act88F

Among the structural proteins that contain Rbfox1 motif instances is TnI, the inhibitory subunit of the Troponin complex (Figs 4A and

S3C). TnI is encoded by the gene *wings up A* (*wupA*), and loss of an IFM-specific isoform of TnI was previously shown to result in hypercontraction (Barbas et al, 1993; Nongthomba et al, 2004). We checked the expression of TnI in *Rbfox1*-RNAi IFMs and found that TnI protein levels were significantly up-regulated in IFMs with *Rbfox1* knockdown (Fig 4B and C), and significantly reduced in IFMs with Rbfox1 overexpression (Fig 4E and F). Changes in corresponding levels of *wupA* mRNA were not significant (Fig S4A). By contrast, we did not observe significant changes in the protein or mRNA expression level of Act88F in *Rbfox1-RNAi* flies (Figs 4B and D and S4B). However, overexpression of Rbfox1 significantly decreased the expression level of Act88F protein and mRNA (Figs 4E and G and S4B), and $Rbfox1^{27286}$ and $Rbfox1^{KK110518}$ knockdown in TDT, but not IFMs, resulted in significantly decreased levels of Act88F mRNA (Fig S4C and D). These data demonstrate that the expression levels of structural proteins in both IFMs and TDT are altered after *Rbfox1* knockdown.

To determine whether Rbfox1 directly binds *wupA* and *Act88F* mRNAs, we performed RNA immunoprecipitation (RIP). We used the $Rbfox1^{CC00511}$ (Rbfox1-GFP) fly line to pull down target RNAs bound to Rbfox1. After confirming that anti-GFP antibodies could selectively immunoprecipitate Rbfox1-GFP (Fig 4H), we performed RIP and amplified RNA bound to Rbfox1 by RT–PCR with gene-specific primers (Table S3). The *Act88F* transcript, which lacks Rbfox1 binding sites and thus served as the negative control, could not be detected after RIP (Fig 4I). By contrast, *wupA* mRNA was enriched in the RIP with anti-GFP antibodies, but not in the IgG isotype control (Fig 4I'). Interestingly, the *wupA* transcript contains a single Rbfox1 motif in the 3'-UTR, suggesting this regulation is post-transcriptional. To support this interpretation, we performed co-immunoprecipitation from $Rbfox1^{CC00511}$ (Rbfox1-GFP) thoraxes followed by mass spectrometry to identify protein interactors of Rbfox1 (Fig S4E–G). We found that Rbfox1 interacted with the cellular translation machinery, including the eukaryotic translation initiation factor eIF4-A and nonsense-mediated decay regulator Rent1 (Fig S4G), motivating future experiments to determine if Rbfox1 regulates target mRNA stability or translation. These findings demonstrate that Rbfox1 directly binds the 3'-UTR of the *wupA* mRNA to regulate its expression, and physically interacts with other post-transcriptional regulatory factors.

### Misregulation of TnI contributes to hypercontraction in Rbfox1 knockdown IFMs

We wondered if the hypercontraction phenotype observed after *Rbfox1* knockdown and overexpression could be partially caused by misregulation of TnI expression. To test this possibility, we performed genetic interaction studies with TnI alleles $wupA^{hdp-3}$ and $wupA^{fliH}$ (Fig 4A). The $wupA^{hdp-3}$ mutant is caused by a mutation in the splice site preceding exon 6b1 (Barbas et al, 1993), and has a hypercontraction phenotype in IFMs in the heterozygous condition (Nongthomba et al, 2004). The $wupA^{fliH}$ mutant has a mutation in the Mef2 binding site located in an upstream response element and results in hypercontracted IFMs with reduced levels of TnI (Firdaus et al, 2015). Since *Rbfox1*-RNAi knockdown increases TnI levels (Fig 4A and B), we knocked down *Rbfox1* in each of the $wupA^{fliH}$ and $wupA^{hdp-3}$ mutant backgrounds to see if TnI levels were restored and hypercontraction was rescued. As $wupA^{fliH}$ is a recessive

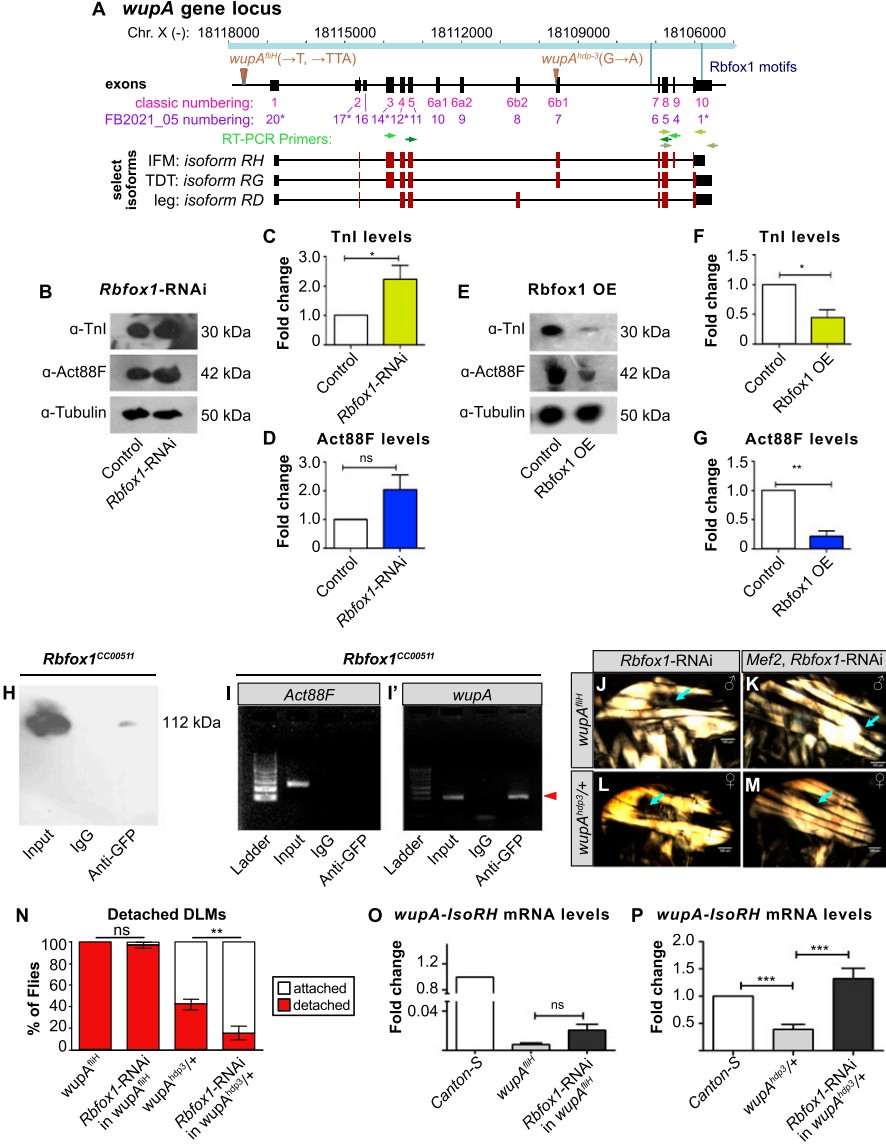

**Figure 4. Rbfox1 regulates expression of structural proteins in indirect flight muscles (IFMs).**
**(A)** Scheme of the *wupA* genomic locus. IFMs, tergal depressor of the trochanter and other tubular muscles express different *wupA* isoforms. The location of Rbfox1 motifs (light blue), RT–PCR primer pairs (greens) and lesions in the *wupA^fliH^* and *wupA^hdp-3^* mutants (brown) are noted. Both classic (magenta) (Barbas et al, 1993) and currently annotated (FB2021_05, purple) exon numbers are provided. Exons with an asterisk have multiple, consecutive numbers. **(B)** Western blot for TnI, Act88F, and Tubulin protein levels in *Rbfox1*-RNAi IFMs. **(B, C, D)** Quantification of TnI (C) and Act88F (D) expression levels from (B), normalized against Tubulin signal. **(E)** Western blot for TnI, Act88F, and Tubulin protein levels in IFMs with UH3-Gal4 driven Rbfox1 overexpression (Rbfox1 OE). **(E, F, G)** Quantification of TnI (F) and Act88F (G) expression levels from (E), normalized against Tubulin signal. **(C, D, F, G)** Error bars in (C, D, F, G) show SD; data from three biological replicates. Significance is from paired *t* test (ns, not significant; *P < 0.05; **P < 0.01). **(H)** Western blot confirming Rbfox1-GFP (*Rbfox1^CC00511^*) is selectively immunoprecipitated with anti-GFP antibody. **(I, I')** Gels showing RNA immunoprecipitation (RIP) followed by RT–PCR from *Rbfox1*-GFP thoraces. **(I, I')** mRNA from *Act88F* (I), which does not have an Rbfox1 motif in the UTR region, is not detected via RIP, whereas *wupA* (TnI) mRNA can be detected via RIP (red arrowhead, I'), indicating direct Rbfox1 binding. **(J, K, L, M)** Polarized microscopy images of hemithoraxes from *wupA^fliH^* hemizygous males (J), *wupA^fliH^*, *Rbfox1*-RNAi males (K), *wupA^hdp-3/+^* heterozygous females (L), and *wupA^hdp-3/+^*, *Rbfox1*-RNAi females (M) with detached IFM myofibers (cyan arrow). Scale bars = 100 µm.
**(I, J, K, L, N)** Quantification of myofiber attachment in (I, J, K, L) reveals a partial rescue in *wupA^hdp-3/+^*, *Rbfox1*-RNAi females. Significance is from paired *t* test, **P < 0.01. **(O)** RT-qPCR for *wupA* mRNA transcript levels in IFMs from *Canton-S*, *wupA^fliH^*, and *wupA^fliH^*, *Rbfox1*-RNAi males. **(P)** RT-qPCR for *wupA-6b1* mRNA transcript levels in IFMs from *Canton-S*, *wupA^hdp-3/+^*, and *wupA^hdp-3/+^*, *Rbfox1*-RNAi females. Significance is from paired *t* test (ns, not significant; ***P < 0.001).

mutation, we examined hemizygous males but did not observe a rescue of muscle hypercontraction with *Rbfox1*-RNAi (Fig 4J, K, N, and O). However, *Rbfox1*-RNAi in *wupA^hdp-3^* heterozygous mutant female flies partially rescued the IFM hypercontraction phenotype and significantly reduced myofiber loss (Fig 4L–N). This suggests that the transcriptional defect in *wupA^fliH^* cannot be rescued by knockdown of *Rbfox1*, but the splicing defect in *wupA^hdp-3^* flies may be at least partially compensated. To test this hypothesis, we performed quantitative RT–PCR and found that the 60–64% reduction of *wupA* mRNA expression in *wupA^fliH^* heterozygous mutants cannot be rescued in *Rbfox1-RNAi*, *wupA^fliH^* IFMs (Fig 4O). By contrast, whereas expression of the IFM-specific *wupA* isoform is significantly reduced in *wupA^hdp-3^* mutants, this isoform is rescued in *Rbfox1-RNAi*, *wupA^hdp-3^* IFMs (Fig 4P). These results demonstrate that Rbfox1 regulation of TnI expression contributes to the muscle hypercontraction phenotype, and further lead us to hypothesize

that Rbfox1 regulates the muscle-specific splicing of structural genes, which we explore below.

## Rbfox1 and the RBP Bruno1 regulate each other's expression

In addition to structural proteins, our bioinformatic analysis revealed Rbfox1 motif instances in RBPs such as *bruno1* (*bru1*) (Figs 5A and S3C). Bru1 was previously shown to be necessary and sufficient for IFM-specific alternative splicing of structural protein genes, including *wupA* (Oas et al, 2014; Spletter et al, 2015). To determine if Rbfox1 regulates Bru1, which could contribute to misregulation of alternative splicing in the Rbfox knockdown background, we evaluated Bru1 protein expression after *Rbfox1* knockdown using immunostaining and Western blot. In immunostainings of wild-type (*w^1118^*) adult IFMs, Bru1 is strongly expressed and localized to the nucleus (Fig 5B). We found that Bru1

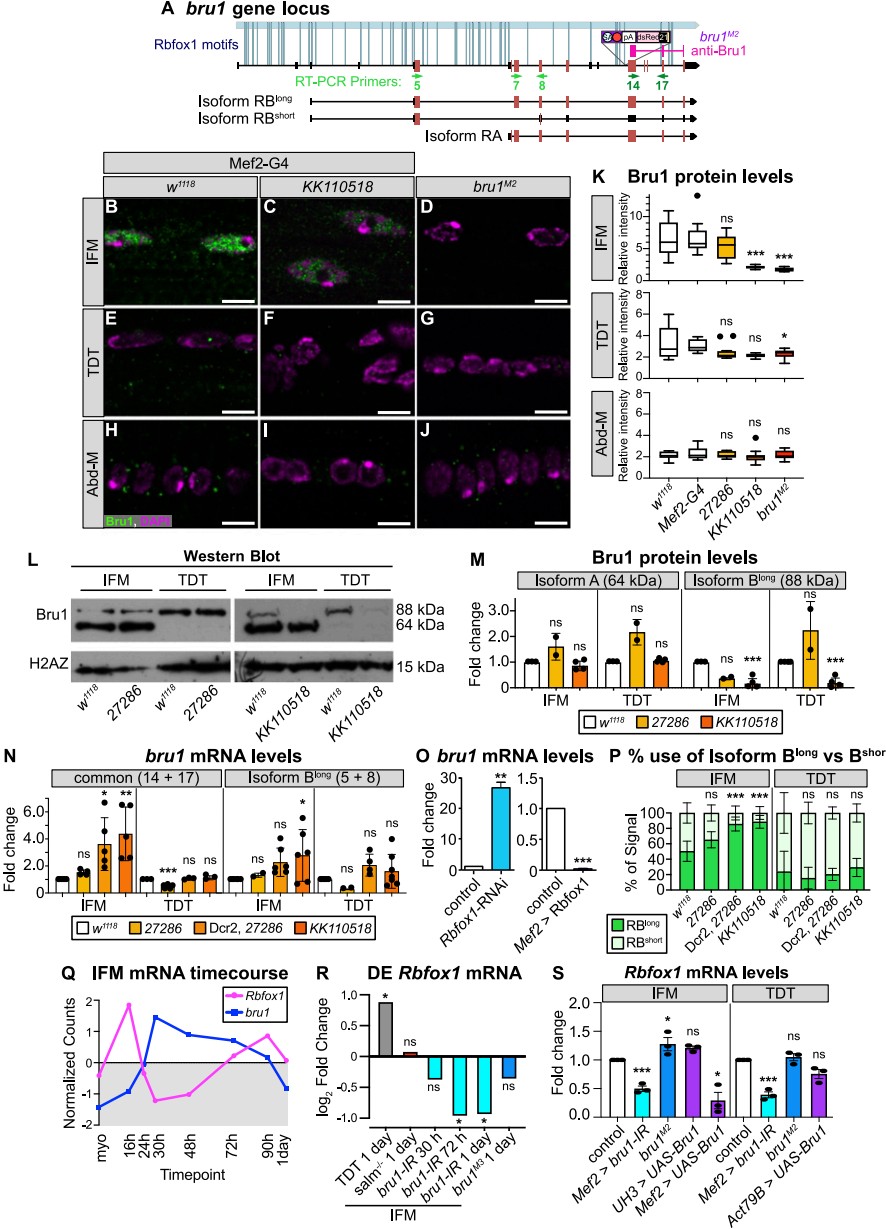

**Figure 5. A cross-regulatory interaction exists between Rbfox1 and Bru1.**

**(A)** Diagram of the *bruno1* (*bru1*) locus. Representative isoforms including *bru1-RA* and *bru1-RB* (*bru1-RB^long*, annotated full length), as well as a novel *bru1-RB^short* isoform which splices over exon 7 resulting in a frame shift and early truncation (see also Fig S5J), are illustrated. Exons, red; UTR, black. In the *bru1^M2* allele (purple), the modification cassette containing a strong splice acceptor followed by a triple frame stop inserted upstream of exon 12, resulting in a strong hypomorphic allele (see also Fig S5A–E). Rbfox1 binding motif instances (light blue lines) and the target region of the rabbit anti-Bru1 antibody (magenta) are indicated. RT–PCR primers, green. Not drawn to scale. **(B, C, D, E, F, G, H, I, J)** Confocal images of immunostaining with rabbit anti-Bru1 in indirect flight muscles (IFMs) (B, C, D), tergal depressor of the trochanter (TDT) (E, F, G), and abdominal muscle (H, I, J). **(C, D, F, G, I, J)** Bru1 signal is reduced in IFMs with *Rbfox1-IR^KK110518* (C, F, I) and undetectable via immunofluorescence in *bru1^M2* mutant muscle (D, G, J). Bru1, green; DAPI, magenta; Scale bars = 5 μm. **(B, C, D, E, F, G, H, I, J, K)** Quantification of Bru1 fluorescence levels in (B, C, D, E, F, G, H, I, J). Boxplots are shown with Tukey whiskers. Significance determined by ANOVA and post hoc Tukey in comparison to both wild-type (*w^1118*) and Gal4 alone (*Mef2-Gal4 x w^1118*) controls (ns, not significant; *P < 0.05; ***P < 0.001). **(L)** Western blot of Bru1 protein levels in IFMs and TDT from *Rbfox1-IR^27286* (left) and *Rbfox1-IR^KK110518* (right) knockdown flies. Levels of isoform Bru1-PA (at 64 kD) do not change, whereas levels of the Bru1-PB isoform (at 88 kD) decrease in *Rbfox1-IR^KK110518* muscle. H2AZ was used as a loading control. **(L, M)** Quantification of fold change in band intensity in (L), normalized to H2AZ and control IFM expression levels. *w^1118*, white; *Rbfox1-IR^27286*, light orange; *Rbfox1-IR^KK110518*, dark orange. **(N)** Quantification of fold change in band intensity from semi-quantitative RT–PCR with primers specific to *bru1-RB* (primers 5 + 8) or common to all *bru1* isoforms (primers 14 + 17) (representative gel images in Fig S5F–H). Intensity was normalized to *RpL32* (*RP49*) and control IFM expression levels. Error bars represent SD. Significance determined by ANOVA and post-hoc Tukey (ns, not significant; *P < 0.05; **P < 0.01, ***P < 0.001). **(O)** Quantification of RT-qPCR data for *bru1* transcript levels in IFMs from *Rbfox1*-RNAi (left) or Rbfox1 OE (right). Significance is from paired *t* test (**P < 0.01; ***P < 0.001). **(N, P)** Quantification of relative expression level of *bru1-RB^long* versus *bru1-RB^short* in the indicated genotypes. **(N)** Significance as in (N). **(Q)** Standard normal count values for *Rbfox1* (magenta) and *bru1* (blue) from an mRNA-Seq developmental timecourse of wildtype IFMs (Spletter et al, 2018). *Rbfox1* and *bru1* have opposite temporal expression profiles until 72 h after puparium formation (APF). **(R)** Differential expression of *Rbfox1* in mRNA-Seq data based on DESeq2 comparison of IFMs versus TDT (1 d adult), IFMs versus *salm^-/-* IFMs (1 d adult), IFMs versus *bru1-IR* IFMs (30 h APF, 72 h APF, 1 d adult), and IFMs versus *bru1^M3* IFMs (ns, not significant; *P < 0.05). **(S)** RT–PCR quantification of fold change in *Rbfox1* transcript level from IFMs and TDT with altered levels of Bru1 expression including *bru1-IR* (light blue), *bru1^M2* (dark blue) and UAS-Bru1 overexpression (purple) with UH3-Gal4, Mef2-Gal4 and Act79B-Gal4 (representative gel images Fig S5I). **(N)** Errors bars represent SD, significance as in (N). Source data are available online for this figure.

staining is significantly reduced in *Rbfox1*-IR^KK110518 IFMs (Fig 5C and K). Bru1 staining is undetectable in IFMs from the hypomorphic *bru1^M2* allele (Figs 5A, D, and K and S5A–E), indicating that our antibody is specific. We were only able to detect extremely low levels of mostly cytoplasmic Bru1 in wild-type TDT in immunostainings, and the Bru1 signal was undetectable in Abd-M (Fig 5E–K). In Western blot of dissected wild-type IFMs, we consistently observed Bru1 bands at 64 and 88 kD (Fig 5L), corresponding to the Bru1-PA and Bru1-PB protein isoforms, respectively. Expression of

these bands was decreased in IFMs from *bru1-IR* and *bru1^M2* flies, and the antibody recognized purified Bru1-PA at 64 kD (Fig S5D), demonstrating specificity. TDT predominantly expresses Bru1-PB, whereas the Bru1-PA band is observed in dissected ovaries and testis (Figs 5L and S5D). Bru1-PB was significantly reduced in IFMs and TDT from *Rbfox1*-IR^KK110518 flies, whereas the Bru1-PA isoform was largely unaffected (Fig 5L and M). Bru1 protein levels were not significantly changed with weaker knockdown in *Rbfox1*-IR^27286 flies (Fig 5L and M). These data demonstrate that knockdown of *Rbfox1*

alters the protein expression level of Bru1, and notably the Bru1-PB isoform, in muscle.

We next evaluated *bru1* expression at the mRNA level, to gain insight into whether the observed change in Bru1 protein levels reflects an RNA or protein level regulatory mechanism. We focused on IFMs and TDT, where we had detected Bru1 antibody staining. Using semi-quantitative RT–PCR with primers targeting a C-terminal region common to all *bru1* isoforms, we observed a significant increase in overall *bru1* transcript levels in IFMs from strong knockdown conditions, including Dcr2-enhanced *Rbfox1*-IR$^{27286}$, *Rbfox1*-RNAi and *Rbfox1*-IR$^{KK110518}$ (Figs 5N and O and S5F–H). Correspondingly, overexpression of Rbfox1 significantly reduced *bru1* transcript levels in IFMs (Fig 5O). When we used RT–PCR primers that selectively amplified *bru1-RB*, we unexpectedly observed two isoforms: the annotated isoform (*bru1-RB$^{long}$*) as well as a novel event we refer to as *bru1-RB$^{short}$* that skips *bru1* exon 7 resulting in a frame shift and stop in exon 8 (Figs 5A and S5J). We saw a significant increase in *bru1-RB$^{long}$* expression in *Rbfox1*-IR$^{KK110518}$ IFMs (Figs 5N and S5F–H), as well as a significant switch in isoform use, selectively in IFMs, from both Dcr2-enhanced *Rbfox1*-IR$^{27286}$ and *Rbfox1*-IR$^{KK110518}$ flies (Fig 5P). mRNA levels of *bru1* were not significantly regulated in TDT (Figs 5N and P and S5F–H). We additionally performed RIP to determine if Rbfox1 regulation of *bru1* mRNA is direct and indeed could detect *bru1* RNA bound to Rbfox1-GFP (Fig S4I), but we are unable to resolve the specific transcript or distinguish between mature mRNA or partially spliced pre-mRNA in the bound fraction. We conclude that Rbfox1 regulates the expression level of *bru1* mRNA and protein in fibrillar IFMs, which motivates future experiments to decipher the detailed biochemical regulatory mechanism.

We next evaluated if Bru1 regulates the expression of *Rbfox1*. We observed that endogenous *Rbfox1* and *bru1* transcripts have opposite temporal mRNA expression profiles across IFM development in mRNA-Seq data (Fig 5Q), and the observed dip in *Rbfox1* transcript levels corresponds with a decrease in Rbfox1-GFP expression at mid-points of development (Fig 1B–F). We then examined what happens to *Rbfox1* expression if we genetically alter Bru1 levels. *Rbfox1* is significantly down-regulated in mRNA-Seq data from *bru1-IR* IFMs at 72 h APF and in 1 d adults (Fig 5R). We confirmed this result in 1 d adults via semi-quantitative RT–PCR on both *bru1-IR* IFMs and TDT (Figs 5S and S5I). Early and strong Bru1 overexpression with the Mef2 driver significantly decreases *Rbfox1* mRNA levels in IFMs, but overexpression in IFMs from 34 h APF with UH3-Gal4 does not (Fig 5S). Overexpression of Bru1 in TDT with Act79B-Gal4 did not significantly reduce *Rbfox1* mRNA levels (Fig 5S). *Rbfox1* mRNA levels in *bru1$^{M2}$* IFMs are weakly increased, but *Rbfox1* expression is not significantly altered in *bru1$^{M3}$* IFMs or *bru1$^{M2}$* TDT (Figs 5S and S5I), suggesting this regulation depends on how much Bru1 protein is present in the muscle. These data indicate that Bru1 can regulate Rbfox1 levels in *Drosophila* muscle, although further experiments will be necessary to establish if this regulation is direct or indirect.

### Rbfox1 and Bru1 genetically interact during IFM development

Having established that Rbfox1 and Bru1 regulate each other's expression, we next explored if they might cooperatively regulate muscle development. Similar to the *Rbfox1* knockdown phenotype

in IFMs (Figs 3, 6C and G, and S6B and F) and as compared to the control (Figs 6A and E and S6A and E), *bru1$^{M2}$* and *bru1*-IR flies display IFM-specific loss of myofibers and a hypercontraction phenotype characterized by short, thick sarcomeres (Figs 6B, F, Q, and R and S6C and G), as has been previously reported (Oas et al, 2014; Spletter et al, 2015). Unlike *Rbfox1* knockdown which causes a phenotype in both TDT and Abd-M (Figs. 2, 6K and O, and S6J and N), decreased Bru1 levels in *bru1$^{M2}$* and *bru1*-IR flies does not produce a phenotype in either TDT or Abd-M (Figs 6I, J, M, N, S, and T and S6I, K, M, and O). To test if overexpression of Bru1 can also induce a hypercontraction phenotype like we observed with overexpression of Rbfox1 (Fig S2E), we drove UAS-Bru1 using Mhc-Gal4 (which expresses from 40 h APF onwards). Indeed, as compared with the control (Fig S6Q and Q'), overexpression of Bru1 leads to an IFM hypercontraction phenotype, including myofiber loss and torn myofibrils with short sarcomeres (Fig S6R and R'). This phenotype could be partially rescued by the *Mhc$^{P401S}$* allele of myosin heavy chain (Fig S6S and S'), confirming that myofiber detachment is indeed due to hypercontraction. Thus, loss as well as gain of both Bru1 and Rbfox1 in IFMs results in similar phenotypes, including hypercontraction.

This led us to test what happens to muscles lacking both Rbfox1 and Bru1. Knockdown with *Rbfox1*-IR$^{27286}$ in the *bru1$^{M2}$* background reveals a strong genetic interaction. IFM myofibers were still present but severely disorganized and displayed an unusual banded actin pattern (Fig 6D). Myofibril and sarcomere structures were completely compromised, and F-actin formed into disarrayed clumps, as well as spine and star-like structures (Fig 6H). We obtained an identical IFM phenotype with double knockdown in *bru-IR, Rbfox1*-RNAi flies (Fig S6D and H). This genetic interaction is restricted to IFMs, as the phenotype in TDT and Abd-M was not enhanced and appeared consistent with the phenotype observed in *Rbfox1*-IR$^{27286}$ (compare Fig 6K and O to Fig 6L and P) or *Rbfox1*-RNAi (compare Fig S6J and N to Fig S6L and P) alone. TDT myofibrils were disorganized and frayed with short sarcomeres (Figs 6L and S and S6L), whereas Abd-M myofibrils were discontinuous and sarcomere structure was irregular (Figs 6P and T and S6P). This result indicates that Rbfox1 and Bru1 genetically interact in fibrillar IFMs, but not in tubular TDT and Abd-M where primarily Rbfox1 seems to function.

### Rbfox1 and Bruno1 co-regulate alternative splice events in IFMs

Considering their strong genetic interaction in IFMs, we next checked if Rbfox1 and Bru1 co-regulate alternative splicing in *Drosophila* muscle. When examining transcriptome-wide motif instances of Rbfox1 and Bru1 in the oRNAment database (Bouvrette et al, 2020), we found that of 64 sarcomere proteins with at least one motif, 45% have motif instances of both Rbfox1 and Bru1 (compared with 31% of all genes) (Fig S4H and K). Rbfox1 and Bru1 motifs are also closer together than is expected if the motifs were located randomly in the transcriptome (Fig S4J), as anticipated if they co-regulate specific targets. We therefore selected a panel of alternative splice events, many of which produce fibrillar and tubular specific isoforms, in 12 structural proteins, including *Formin homology 2 domain containing* (*Fhos*), *Myosin heavy chain* (*Mhc*), *rhea* (Talin), *sarcomere length short* (*sals*), *sallimus* (*sls*), *Stretchin-Mlck*

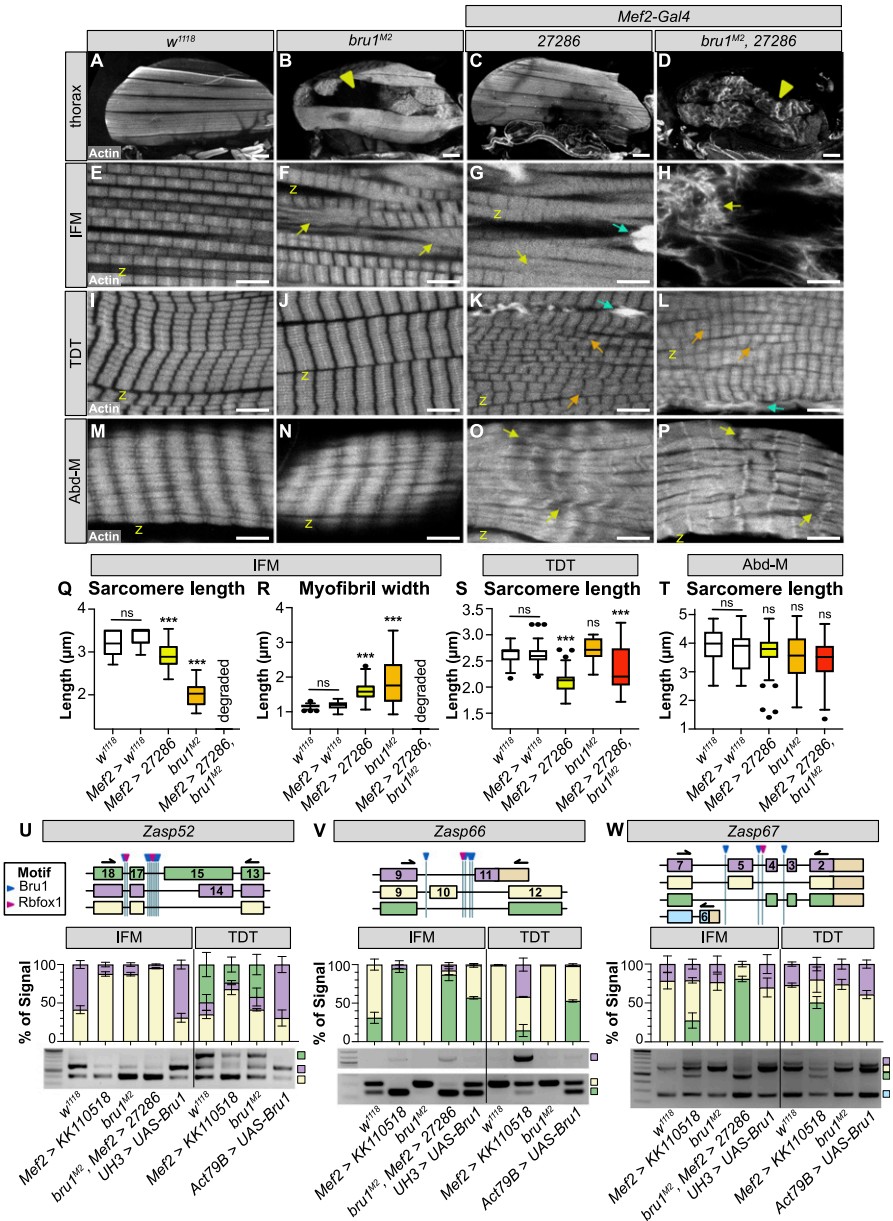

**Figure 6. Rbfox1 and Bru1 genetically interact in indirect flight muscle (IFM) myogenesis and regulate the alternative splicing of sarcomere genes.**
**(A, B, C, D)** Confocal projections of hemithoraces showing IFMs (A, B, C, D) from $w^{1118}$, $bru1^{M2}$, Rbfox1-$IR^{27286}$ and $bru1^{M2}$, Rbfox1-$IR^{27286}$ flies. Arrowheads indicate aberrant, torn myofibers. Scale bars = 100 μm. **(E, F, G, H)** Single-plane confocal images from IFMs, showing torn myofibrils (yellow arrows) with short sarcomeres and actin inclusions (cyan arrows) in $bru1^{M2}$ (F) and Rbfox1-$IR^{27286}$ (G). **(H)** $bru1^{M2}$, Rbfox1-$IR^{27286}$ demonstrates genetic interaction and loss of myofibril structure (H). **(I, J, K, L, M, N, O, P)** Single-plane confocal images from tergal depressor of the trochanter (I, J, K, L) and abdominal muscle (M, N, O, P) from $w^{1118}$, $bru1^{M2}$, Rbfox1-$IR^{27286}$ and $bru1^{M2}$, Rbfox1-$IR^{27286}$ flies. Myofibrils in Rbfox1 knockdown muscles are disorganized (orange arrows), have actin inclusions (cyan arrows) and are often torn (yellow arrows). Scale bars = 5 μm. **(Q, R)** Quantification of sarcomere length (Q) and myofibril width (R) in IFMs. **(S, T)** Quantification of sarcomere length in tergal depressor of the trochanter (S) and abdominal muscle (T). Boxplots are shown with Tukey whiskers, with outliers denoted by dots. Significance determined in comparison to $w^{1118}$ by ANOVA and post hoc Tukey (ns, not significant; *$P$ < 0.05; ***$P$ < 0.001). **(U, V, W)** RT–PCR for select alternative splice events in Zasp52 (U), Zasp66 (V) and Zasp67 (W). Top: Diagram of alternative isoforms and primer locations. The location of predicted motifs for Rbfox1 (magenta) and Bru1 (blue) are indicated. Diagrams are oriented according to transcript 5′ to 3′. Exon numbers according to annotation FB2021_05. 3′-UTR regions, tan; color coding of select isoforms consistent across top, middle and bottom panels. Middle: Quantification of relative expression level of detectable events. Bottom: RT–PCR gel image. Genotypes as labeled. Ladder in the far-left lane. Source data are available online for this figure.

(Strn-Mlck), Tropomyosin 1 (Tm1), thin (tn), wupA (TnI), Z band alternatively spliced PDZ-motif protein 52 (Zasp52), Zasp66, and Zasp67. We then assayed splicing changes in IFMs and TDT from Rbfox1$^{KK110518}$ and $bru1^{M2}$ flies using RT–PCR. As described below, we identified four classes of alternative splicing regulatory events, including events misregulated (1) in the same direction in both Rbfox1$^{KK110518}$ and $bru1^{M2}$, (2) in opposite directions in Rbfox1$^{KK110518}$ and $bru1^{M2}$, (3) in either Rbfox1$^{KK110518}$ or $bru1^{M2}$, and (4) not misregulated in either genotype.

We identified alternative splice events in Zasp52, Mhc, and Tm1 that change in a similar direction in Rbfox1$^{KK110518}$ and in $bru1^{M2}$ flies. Fibrillar and tubular specific splice events in Zasp52 have been reported to be regulated by Bru1 (Spletter et al, 2015), and we confirmed that Zasp52 exon 15 is preferentially included in TDT,

whereas the shorter version of this exon, exon 14, is preferentially included in IFMs (Fig 6U). After Rbfox1 knockdown, Zasp52 isoforms containing exon 14 or exon 15 are both decreased in IFMs and TDT. In $bru1^{M2}$ mutants, Zasp52 isoforms containing exon 14 are decreased selectively in IFMs, and overexpression of Bru1 is sufficient to increase the inclusion of exon 14 in both IFMs and TDT (Fig 6U). This suggests that Rbfox1 promotes use of Zasp52 exon 15, whereas Bru1 promotes use of the alternative 3′ splice site generating exon 14. Mhc has three alternative C-terminal exons, and as previously reported (Clyne et al, 2003; Orfanos & Sparrow, 2013; Kao et al, 2019), we found that in adult flies, Mhc exon 35 is preferentially used in IFMs, whereas exon 37 is predominant in TDT (Fig S6U). In both IFMs and TDT from Rbfox$^{KK110501}$ as well as $bru1^{M2}$ flies, use of Mhc exon 35 and 36 is greatly reduced, whereas use of exon 37 is increased (Fig

S6U). This may be an indirect regulatory event, as we found a single possible Bru1 motif in the 3′-UTR region of *Mhc*. We additionally tested the use of *Tm1* alternative C-terminal exons that have been reported to be fiber type–specific (Basi et al, 1984; Nikonova et al, 2020). In whole thorax from *Rbfox* $^{KK110501}$ or from *bru1*$^{M2}$ flies, although splicing from *Tm1* exon 27 to 31 is not altered, the splice event from exon 27 to 30 is lost, whereas exon 27–28/29 is gained (Fig S6V). In these three examples, we found that exon use changes in the same direction when levels of either Rbfox1 or Bru1 are decreased. This indicates that Rbfox1 and Bru1 can cooperatively regulate alternative splicing, or alternatively that misregulation of Bru1 in the Rbfox1 background might be responsible for the observed change in splicing.

An event from our panel in *Zasp66* revealed opposing regulatory effects in *Rbfox1*$^{KK110518}$ as compared to *bru1*$^{M2}$ flies. In TDT, all expressed *Zasp66* isoforms include alternative cassette exon 10, whereas in IFMs, about 60% of isoforms include exon 10, whereas the other 40% skip this exon (Fig 6V). In *Rbfox1*$^{KK110518}$ and *bru1*$^{M2}$, *Rbfox1*$^{27286}$ IFMs, predominantly the skip event is detected, whereas in *bru1*$^{M2}$ IFMs, only the inclusion event is detected (Fig 6V). Overexpression of Bru1 promotes skipping of *Zasp66* exon 10 in both IFMs and TDT. Strikingly, in *Rbfox1*$^{KK110518}$ TDT, there is a strong increase in use of alternative C-terminal exon 11, which is normally not used in either IFMs or TDT (Fig 6V). The splicing pattern in *bru1*$^{M2}$ TDT is unchanged. This result demonstrates that Rbfox1 promotes inclusion, whereas Bru1 promotes skipping of exon 10, and illustrates the difference in splicing outcome between muscle fiber types.

The remaining events we tested in our splicing panel were misregulated selectively in the *Rbfox1*$^{KK110518}$ (*Fhos* and *Zasp67*) or *bru1*$^{M2}$ (*sls*, *Strn-Mlck*, *wupA*) background, or were unchanged (*rhea*, *sals*, and *tn*). In *Zasp67*, alternative splicing of cassette exons 3, 4, and 5 produces alternative C-terminal domains. *Rbfox1*$^{KK110518}$ knockdown in both IFMs and TDT results in skipping of exon 5, whereas exon use is not altered in *bru1*$^{M2}$ mutants (Fig 6W). Alternative use of *Fhos* exon 8 generates a short isoform with an alternative C-terminus. In both IFMs and TDT, *Rbfox1*$^{KK110518}$ knockdown but not decreased Bru1 expression in *bru1*$^{M2}$ results in increased use of *Fhos* exon 8 (Fig S6X). This indicates these events are Rbfox1 dependent. In contrast to events in *Zasp67* and *Fhos*, the fiber type–specific events we tested in *wupA* and *sls* were altered in IFMs but not TDT from a *bru1*$^{M2}$ background, consistent with previous results in *bru1-IR* IFMs (Spletter et al, 2015), but are unchanged in *Rbfox1*$^{KK110518}$ IFMs and TDT (Fig S6T and X). Overexpression of Bru1 in IFMs and TDT is sufficient to force a switch to the IFM-specific event in *wupA* and *sls* (Fig S6T and X), indicating that these events are Bru1 dependent. An alternative splice event in *Strn-Mlck* that promotes use of an alternative 3′-UTR in exon 25 is also lost specifically in *bru1*$^{M2}$ but not *Rbfox1*$^{KK110518}$ tissues (Fig S6W). Alternative events we tested in *tn*, *rhea* and *sals* were not altered in whole thorax samples from *Rbfox1*$^{KK110518}$ or *bru1*$^{M2}$ flies, and were not pursued further (Fig S6V). Biochemical confirmation of direct binding of Rbfox1 and Bru1 to motifs near regulated exons for all tested events awaits future RNA CLIP studies. Taken together, our data suggest a complex regulatory dynamic where Rbfox1 and Bru1 co-regulate some alternative splice events and independently regulate other events in a muscle type–specific manner.

## Rbfox1 regulates the expression of Mef2, a key transcriptional regulator of muscle genes

Beyond sarcomere proteins and RBPs, we observed Rbfox1 motif instances in transcription factor genes such as *Mef2*, *extradenticle* (*exd*), *spalt major* (*salm*), and others (Fig S3C and Table S1), which have been shown to regulate adult muscle identity or myofiber gene expression (Schönbauer et al, 2011; Dobi et al, 2015). Misregulation of transcription factor expression or function in an *Rbfox1* knockdown background could plausibly provide an indirect mechanism for changes in expression of structural genes, such as Act88F (Figs 4A–F and S4A–D), that lack Rbfox1-binding motifs. Thus, we next tested if Rbfox1 regulates transcriptional activators including *Mef2*, *salm*, and *exd*, which could in turn regulate muscle gene expression.

Mef2 is a well-characterized MADS-box transcription factor that regulates and maintains structural protein expression in muscle (Molkentin et al, 1995; Tanaka et al, 2008). *Mef2* mRNA levels in IFMs were significantly up-regulated with *Rbfox1*-RNAi, and significantly down-regulated with Rbfox1 overexpression (Fig 7A). This regulation is direct, as *Mef2* contains two Rbfox1 motifs in the 3′-UTR, and we were able to detect Rbfox1 binding to *Mef2* mRNA in RIP from adult thoraces of *Rbfox1*$^{CC00511}$ flies (Figs 7B and S3C). Mef2 expression level is known to affect muscle morphogenesis (Gunthorpe et al, 1999), so we next examined whether increased Mef2 levels can induce a phenotype similar to that observed in *Rbfox1* knockdown flies. Mef2-Gal4 driven overexpression of UAS-*Mef2* caused lethality after 48 h, but flies with Mhc-Gal4–driven overexpression had significantly increased *Mef2* levels (Fig S7A) and survived to adulthood. These flies had increased protein levels of TnI and Act88F in IFMs (Fig 7C), in agreement with our observations in *Rbfox1* knockdown flies. However, flies overexpressing Mef2 notably did not display a hypercontraction defect, even though they were flightless and displayed sarcomeric defects (Fig 7D and D′). We conclude that increased levels of Mef2 can lead to an overall increase in many structural proteins, but hypercontraction observed upon changes in Rbfox1 and Bru1 levels likely results from alternative splicing defects and a possible isoform-imbalance amongst structural proteins.

While examining Rbfox1 motif instances, we also noticed that many potential Rbfox1-binding sites (8 of 12) are concentrated in the upstream gene region of *Mef2* (Figs 7E and S3C). Although adult muscle-specific use of these regions has not yet been described, based on the annotation, three distinct promoters, combined with alternative splicing of seven exons, generates five different *Mef2* 5′-UTR regions (Fig 7E). In our mRNA-Seq data, although overall differences in *Mef2* mRNA expression were not significant among adult fiber types from 1 d adult flies (Fig S7B), we did observe significant changes in both temporal and fiber type–specific use of *Mef2* 5′-UTR exons (Fig S7C). The short 5′-UTR encoded by *Mef2* exon 17 is preferentially used in IFMs, which we could confirm using qPCR (Figs 7F and S7C). The longer 5′-UTR encoded by *Mef2* exon 20 is used in all muscles as they mature, whereas a second long 5′-UTR encoded by *Mef2* exon 21 is predominantly used in developing tubular muscle and myoblasts (Fig S7C). Interestingly, using RT–PCR, we could detect increased use of *Mef2* exon 17 in IFMs and Abd-M from adult *Rbfox1*-RNAi flies (Fig 7G), and we observed altered dynamics

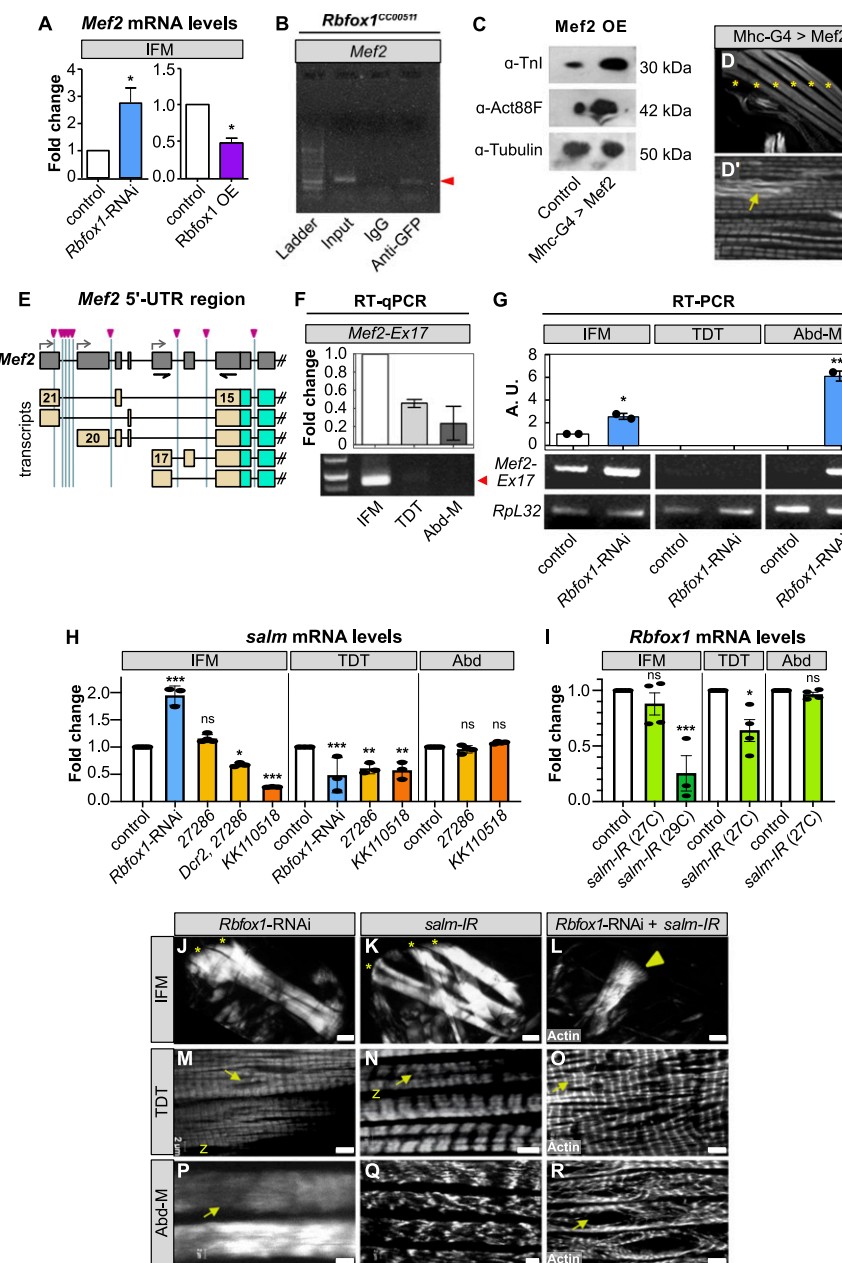

**Figure 7. Rbfox1 regulates expression of myogenic transcription factors and genetically interacts with *salm* in indirect flight muscle (IFM) development.**
**(A)** RT-qPCR quantification of the fold change in *Mef2* mRNA expression in IFMs with *Rbfox1*-RNAi (blue) or Rbfox1 OE (purple). Significance is from paired *t* test (*P < 0.05). **(B)** RNA immunoprecipitation using the *Rbfox1^CC00511^* line followed by RT–PCR indicates Rbfox1 binds to *Mef2* mRNA (red arrowhead). **(C)** Western blot demonstrating increased expression levels of Actin88F and TnI in IFMs with Mef2 OE. **(C', D, D')** Confocal images of thorax hemisection (D) and IFM myofibrils (C') with overexpression of Mef2 driven by Mhc-Gal4. Myofibrils show actin accumulations (yellow arrow), but no hypercontraction. "*" indicates IFM myofibers. **(E)** Diagram of *Mef2* 5'-UTR region and annotated isoforms. Predicted Rbfox1 motifs marked by magenta arrowheads. UTR regions, tan; primers, black. **(F)** Semi-quantitative RT–PCR demonstrating that *Mef2* isoforms containing exon 17 and thus a short 5'-UTR (see also Fig S7C) are preferentially expressed in wildtype IFMs. Red arrow marks PCR band at 885 base pairs. **(G)** RT–PCR detects increased use of *Mef2-Ex17* in *Rbfox1*-RNAi IFMs and abdominal muscle. Quantification, top; RT–PCR gel, bottom; arbitrary units (A.U.). **(H)** Fold change in *salm* transcript levels in IFMs, tergal depressor of the trochanter (TDT) and Abd after Rbfox1 knockdown as determined by RT-qPCR (*Rbfox1*-RNAi) and semi-quantitative RT–PCR (*Rbfox1-IR^27286^*, *Rbfox1-IR^KK110518^*). Data were normalized by *RpL32* levels. **(I)** Fold change in *Rbfox1* transcript levels in IFMs, TDT and Abd normalized to *RpL32* after *salm-IR* at 27°C or 29°C, as determined by RT-qPCR (29°C) and semi-quantitative RT–PCR (27°C). Significance determined by ANOVA and post hoc Tukey (ns, not significant; *P < 0.05, **P < 0.01, ***P < 0.001), error bars indicate SD. **(J, K, L, M, N, O, P, Q, R)** Polarized microscopy images of hemithoraces showing a reduction in myofiber number (stars) with *Rbfox1*-RNAi (J) and *salm-IR* (K), and a complete loss of IFMs with double *Rbfox1*-RNAi, *salm-IR* knockdown (L). TDT, yellow arrowhead. Scale bars = 100 μm. **(M, N, O, P, Q, R)** Single-plane confocal images of tubular TDT (M, N, O) and abdominal muscle (P, Q, R) showing abnormal myofibril structure and tearing (arrows) in *Rbfox1*-RNAi, *salm-IR*, and *Rbfox1*-RNAi, *salm-IR* knockdown. Scale bars = 5 μm. **(J, K, L, M, N, O, P, Q, R)** Phenotypes from (J, K, L, M, N, O, P, Q, R) are quantified in Fig S7K.
Source data are available online for this figure.

of *Mef2* exon use in mRNA-Seq data from *bru-IR* IFMs (Fig S7C), suggesting that Rbfox1 and Bru1 influence use of these variable *Mef2* 5'-UTR regions. This regulation may be direct, through regulation of alternative splicing, or indirect, by influencing use of alternative promoter regions.

## Rbfox1 regulates fibrillar fate determining transcriptional activator Salm

In addition to *Mef2*, we also tested if transcription factors *salm* and *exd* are misregulated after *Rbfox1* knockdown. Salm is a C2H2-type zinc finger transcription factor that serves as a master regulator of the fibrillar muscle fate, and is expressed downstream of the

homeodomain protein Exd (Schönbauer et al, 2011; Bryantsev et al, 2012b). Salm is speculated to influence muscle diversification by modification of Mef2 expression level (Spletter & Schnorrer, 2014) and is known to regulate expression of *bru1*, *wupA*, and *Act88F* (Schönbauer et al, 2011; Spletter et al, 2015, 2018). Thus, we wanted to determine if Salm interacts with the Rbfox1 regulatory hierarchy, as they share multiple regulatory targets. In TDT, *salm* levels were significantly decreased in all *Rbfox1* knockdown conditions, and in IFMs, *salm* levels were significantly decreased with both Dcr2-enhanced *Rbfox1-IR^27286^* and *Rbfox1-IR^KK110518^* knockdown (Figs 7H and S7E). *salm* mRNA levels were, however, significantly increased in *Rbfox1-RNAi* IFMs (Fig 7H). The molecular mechanism by which Rbfox1 regulates Salm remains to be determined, but we did test

*exd* expression and found that *exd* levels were significantly decreased in *Rbfox1-IR$^{KK110518}$* IFMs, but unchanged in TDT or in *Rbfox1-IR$^{27286}$* or *Rbfox1-RNAi* genotypes (Fig S7D). Thus, a change in *exd* expression in IFMs might indirectly affect *salm* expression after knockdown of *Rbfox1*. We further tested if *Rbfox1* expression levels are regulated by Salm, and after confirming that *salm-IR* is efficient (Fig S7H), we indeed saw a significant decrease in *Rbfox1* mRNA levels in IFMs and TDT after *salm-IR* (Figs 7I and S7F). These data suggest that Rbfox1 and Salm cross-regulate each other's expression in IFMs and TDT.

To examine the physiological relevance of this regulatory interaction between Rbfox1 and Salm, we knocked down both factors in all muscle fiber types using Mef2-Gal4. We verified previous findings that *salm-IR* results in a tubular muscle fate conversion of the IFMs and loss of *bru1* expression (Figs 7K and S7I and K) (Schönbauer et al, 2011; Spletter et al, 2015). Strikingly, the muscle phenotype was even more pronounced in double knockdown flies, and IFMs were completely missing from adult hemithoraces (Fig 7L, quantification in Fig S7L). Like we observed above for Rbfox1 and Bru1, Rbfox1 and Salm have a strong genetic interaction in IFMs. We then examined tubular TDT, and surprisingly, as Salm has only been reported to be necessary for IFM development (Schönbauer et al, 2011), we observed mild defects in myofibrillar patterning in the TDT with *salm-IR* (Fig 7N) and in *FRT-salm-FRT* mutants (Fig S7G–H'). We confirmed that *salm* mRNA is expressed in TDT (Fig S7E), suggesting that this low level of expression contributes to proper myofibrillogenesis. Double knockdown with *Rbfox1*-RNAi and *salm-IR* resulted in lethality and severe locomotion defects due to structural abnormalities in tubular muscles. Although TDT was present, both myofibril structure and organization were aberrant, and Abd-M displayed a loss of cytoarchitecture similar to that observed with *Rbfox1*-RNAi (Figs 7O and R and S7J). Strikingly, these data demonstrate a genetic interaction between Salm and Rbfox1 in the development of both IFMs and TDT and suggests there is cross-regulation between identity transcription factors and fiber type–specific splicing networks that is necessary for proper fiber type-specific gene expression and alternative splicing. Altogether, our results suggest that Rbfox1 is involved in the regulation of fiber = specific isoforms of structural proteins, particularly TnI, not only through directly regulating RNA and the splicing process, but also through hierarchical regulation of the fiber diversity pathway.

# Discussion

Here, we report the first detailed characterization of Rbfox1 function in *Drosophila* muscle. We show that Rbfox1 functions in a fiber type–specific manner to modulate both fibrillar and tubular muscle development. Collectively, our data demonstrate that Rbfox1 operates in a complex regulatory network to fine-tune the transcript expression levels and alternative splicing pattern of fiber type–specific genes and structural proteins, such as Act88F, TnI, Zasp52, Zasp66, Zasp67, Tm1, Fhos, and Mhc (Fig 8A). It does this directly, by binding to 3'-UTR regions to regulate transcript levels, for example, of *wupA* and *Mef2*, as well as by promoting or inhibiting alternative splice events (Fig 8B). *Rbfox1* expression is higher in

tubular than fibrillar muscle, and the robustness of behavioral, cellular and molecular phenotypes depends on the strength of *Rbfox1* knockdown, indicating that the Rbfox1 regulatory network is carefully balanced and sensitive to alterations in Rbfox1 expression level. In addition, Rbfox1 regulates transcriptional activators, such as Mef2 and Salm, and other splicing factors, such as Bru1, which further contributes to misregulation of transcript levels and alternative splicing in *Rbfox1* knockdown muscle (Fig 8A). Our data demonstrate cross-regulatory and genetic interactions between Rbfox1 and Bru1 in IFM development, including fiber type–specific co-regulation of alternative splice events in structural genes (Fig 8B), reflecting the interdependence of RBP function in myogenesis. Interestingly, cross-regulatory and genetic interactions extend to Salm, which suggests that RBPs such as Rbfox1 actively regulate both transcriptional and splicing networks to guide and refine the acquisition of fiber type–specific properties during muscle differentiation.

## Rbfox1 function in muscle development is evolutionarily conserved

Although *Drosophila* Rbfox1 was previously reported to promote differentiation and survival of ovarian germline cysts (Tastan et al, 2010; Kucherenko & Shcherbata, 2018), and to regulate neuronal differentiation and excitability (Guven-Ozkan et al, 2016; Shukla et al, 2017), the data presented here is the first comprehensive demonstration of Rbfox1 function in fly muscle. Both vertebrate and *Drosophila* Rbfox1 proteins were previously shown to recognize a conserved 5'-UGCAUG-3' motif (Ray et al, 2013; Pedrotti et al, 2015; Nazario-Toole et al, 2018) that is enriched in introns flanking skeletal and cardiac muscle–specific exons in humans and mice (Castle et al, 2008; Kalsotra et al, 2008). We find that Rbfox1-binding motif instances in flies are also found in muscle genes and near fiber type–specific exons (Figs 6, S3, and S6), although future RNA CLIP studies will be necessary to confirm the genome-wide distribution of Rbfox1-bound motifs. Transcripts of hundreds of structural genes are mis-spliced in *Rbfox1* and *Rbfox2* knockout mice, which display developmental defects in muscle structure and function and fail to maintain skeletal muscle mass as adults (Pedrotti et al, 2015; Singh et al, 2018). Similarly, knockdown of Rbfox1 and Rbfox2 in zebrafish leads to defects in alternative splicing, myofiber morphology, and function of both heart and skeletal muscle (Gallagher et al, 2011), and *fox-1* mutants in *C. elegans* display aberrant myoblast migration and impaired egg-laying (Kuroyanagi et al, 2006; Mackereth, 2014). We now show that knockdown of *Rbfox1* in fly muscle causes behavioral deficits and impaired muscle function (Figs 2 and 3). Defects in alternative splicing and isoform expression levels lead to aberrant myofibril and sarcomere structure in both fibrillar and tubular fiber types (Figs 2, 3, 6, and 8). Our results are thus consistent with published functions of Rbfox1 in other organisms, demonstrating a conserved role for Rbfox1 in muscle development. Interestingly, multiple other RBPs also have conserved functions in myogenesis, for example, Mbl (MBNL family), Bru1 and Bru3 (CELF family), How (Quaking family), TBPH (TDP-43), and others (reviewed in Jagla et al [2017] and Nikonova et al [2019]), illustrating that *Drosophila* can be a powerful model to explore conserved myogenic RNA regulatory networks.

## A. Rbfox1 function in muscle development

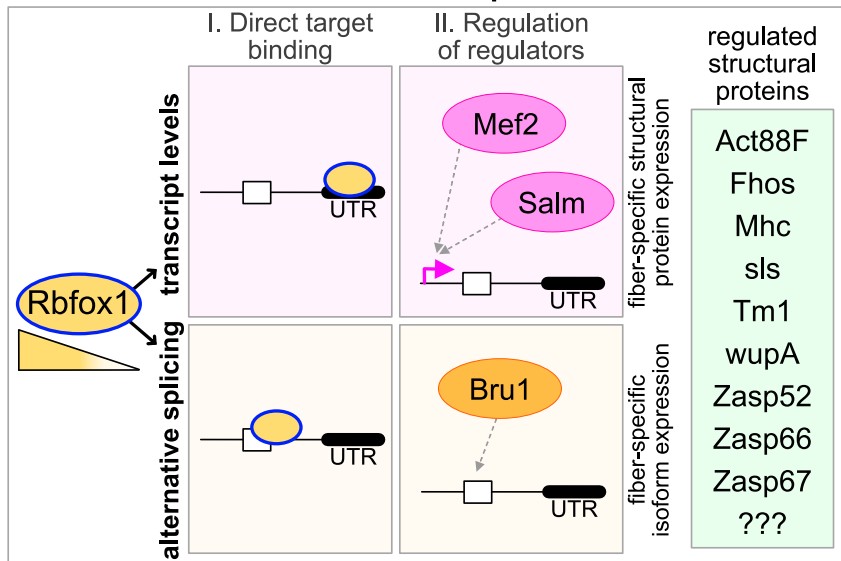

**Figure 8. Model of Rbfox1 function in *Drosophila* muscle development and alternative splicing.**
**(A)** Rbfox1 regulates transcript levels and alternative splicing of muscle genes. Some transcripts are regulated directly through intronic or UTR binding. Other transcripts are regulated indirectly, as Rbfox1 regulates expression of transcriptional activators Mef2 and Salm as well as the RNA-binding protein Bru1. Ultimately, this defines muscle fiber type–specific expression levels and splice isoform usage of sarcomeric genes. RNA-binding proteins, orange; Rbfox1, blue outline; transcription factors, magenta; structural proteins, green. **(B)** Rbfox1 regulates alternative splicing of sarcomere genes. All events tested by RT–PCR in this study (see Figs 6 and S6) and their muscle type specificity are summarized in heatmap form (B1). Events are classified as increased (yellow), decreased (blue) or unchanged (grey) after knockdown of Rbfox1 or Bru1. Exons are numbered according to the FB2021_05 annotation. Schematics in B2 illustrate four types of identified events: single factor events regulated by either Rbfox1 or Bru1, cooperative events regulated by both Rbfox1 and Bru1 (or indirect events affected by changes in Bru1 expression in the Rbfox1 knockdown background), opposing events where Rbfox1 and Bru1 have an opposite regulatory effect, and events that are not regulated by either Rbfox1 or Bru1.

## B. Rbfox1 regulation of alternative splicing

### B1. Summary of tested events

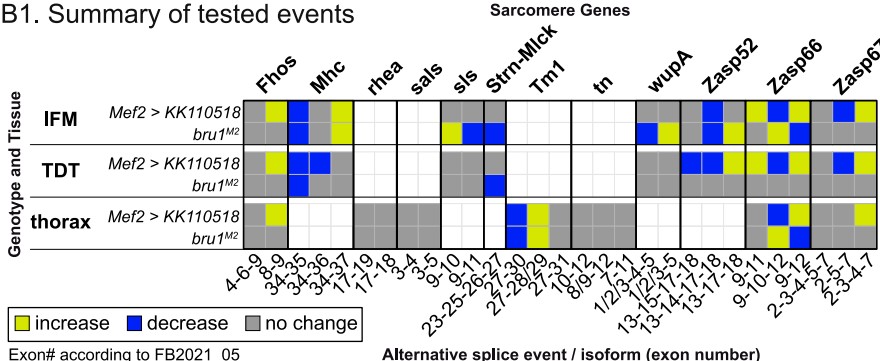

### B2. Types of events

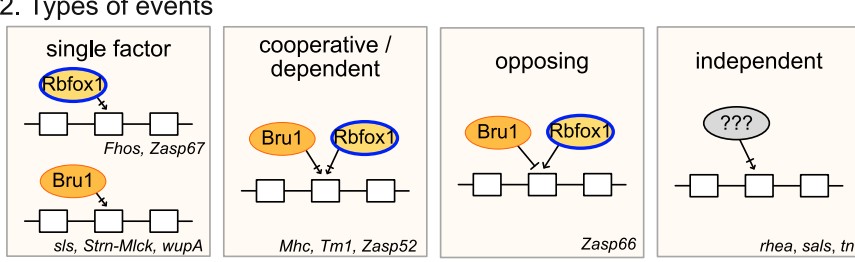

**Cross-regulatory interaction with Bru1 refines temporal and fiber type–specific Rbfox1 expression levels**

Although Rbfox1 is expressed in all muscles we tested, our data show that Rbfox1 expression level is regulated temporally during IFM development and between muscle fiber types (Fig 1). Rbfox1 is significantly up-regulated from 30 to 72 h APF as IFM matures (Fig 1G), which is a similar dynamic to the up-regulation of Rbfox1 observed during vertebrate cardiac cell differentiation (Gao et al, 2016). In mice, this temporal regulation is physiologically important, as *Rbfox1* knockdown results in cardiac hypertrophy and splicing defects, which is moreover consistent with the reduction in Rbfox1 expression

found in human patients with dilated cardiomyopathy and in hypertrophic heart tissue from mice and zebrafish (Gao et al, 2016). In flies, our data demonstrate that *Rbfox1* knockdown results in alternative splicing defects and hypercontraction, but also reveals a continuous requirement for Rbfox1 function during muscle development, as the sarcomere phenotype observed in temporal-restricted Act88F-Gal4 knockdown is less severe than with constitutive Mef2-Gal4 (Figs 2 and 3). It will be interesting to determine if the requirement for Rbfox1 early in fiber differentiation is fly-specific and to identify temporal-specific regulatory signatures of Rbfox1.

The temporal dynamics of Rbfox1 regulation in vertebrate muscle correlate with a transition from embryonic to mature splicing patterns

of muscle genes which is regulated by CELF, MBNL, and RBFOX factors (Blech-Hermoni et al, 2016). Interestingly, RBFOX and CELF family RBPs are suggested to regulate each other's expression. In vertebrates, CELF2 represses Rbfox2 expression in the heart, and overexpression of CELF1/2 or depletion of Rbfox2 leads to the same changes in splicing direction and magnitude (Gazzara et al, 2017). Here we demonstrate for the first time that a cross-regulatory interaction exists between Rbfox1 and Bru1 in fly muscle. Knockdown and overexpression of Rbfox1 result in increased and decreased expression of *bru1*, respectively (Fig 5). Changes in Bru1 expression level also result in corresponding changes in *Rbfox1* expression (Figs 5 and S5). Furthermore, mRNA expression levels of *bru1* and *Rbfox1* are inversely correlated across IFM development (Fig 5), and decreased expression of both RBPs leads to complete loss of myofibril structure (Fig 6). Our data thus show that cross-regulatory interactions are conserved, and therefore may be a common feature of RBFOX and CELF regulatory networks, motivating future studies to clarify the detailed molecular mechanisms.

## Rbfox1 regulation of target genes is expression level-dependent

An intriguing observation from our results with both Rbfox1 and Bru1 is that phenotypic severity as well as the change in the expression level of target genes depends on knockdown strength. We see stronger behavioral defects, increased lethality, and more severe myofibril phenotypes as *Rbfox1* knockdown efficiency increases (Figs 2 and 3). We observe this effect on *bru1*, *exd*, and *salm* expression levels in *Rbfox1* knockdown flies (Figs 5, 7, and S7), and on *Rbfox1* expression in *bru1-IR* or *bru1* mutant flies (Fig 5). We have previously reported a similar variation in phenotypic strength after knockdown with multiple RNAi hairpins targeting other RBPs, notably *SF1*, *Hrb87F*, *snf*, *Prp19*, and others (Kao et al, 2021). These results illustrate the experimental importance of testing multiple, independent RNAi constructs to distinguish hypomorphic from null phenotypes, beyond merely addressing off-target effects (Mohr & Perrimon, 2012; Kaya-Çopur & Schnorrer, 2019; Neumeier & Meister, 2020). Moreover, tuning the level of knockdown can offer insight into regulatory events that are sensitive to RBP activity level. Such level-dependent regulation of alternative splice events is described in human patients and animal models for spinal muscular atrophy, a neuromuscular disorder caused by reduction of the survival motor neuron 1/2 (SMN 1/2) protein (Edens et al, 2015). RNAi as well as hypomorphic *Smn* alleles in *Drosophila* reveal that some splice events are more sensitive than others to a reduced snRNP pool and decreased expression of Smn (Aquilina & Cauchi, 2018; Yeo & Darras, 2020). Such a mechanism could apply more broadly to RNA isoform expression, as unlike Bru1, which has fiber type–restricted expression, most RBPs, like Rbfox1, are widely expressed but show fiber type–specific patterns in expression level, alternative splicing or post-translational modification (Kino et al, 2015; Nakka et al, 2018; Hinkle et al, 2019). However, further studies are needed to understand in greater detail how expression-level sensitivity tunes RNA regulatory outcomes.

## Rbfox1 and Bru1 regulate muscle fiber type–specific isoform expression

The fibrillar muscle fate is initially specified through the activity of transcriptional regulators including Vestigial (Vg), Extradenticle (Exd), Homothorax (Hth), and Spalt-major (Salm) (Schönbauer et al,

2011; Bryantsev et al, 2012b). In addition to transitions in fiber type–specific gene expression (Zappia & Frolov, 2016; Spletter et al, 2018), this muscle identity program dictates the stoichiometry of structural gene isoform expression, which is essential for myofibril assembly and proper regulation of actomyosin contractility (Bottinelli, 2001; Singh et al, 2014; Firdaus et al, 2015; Zhao & Swank, 2017). An integral part of this regulatory program is muscle type–specific alternative splicing, and notably, the splicing regulator Bru1 is up-regulated downstream of Salm (Oas et al, 2014; Spletter et al, 2015). We have previously shown that Bru1 promotes many, but not all, fiber type–specific splice events in IFMs, whereas in parallel factors such SF1 and Hrb87F regulate splicing independently of the fiber identity pathway (Spletter et al, 2015; Kao et al, 2021). The identity and function of other RNA regulatory components and their interaction with the fiber differentiation pathway is still under active investigation.

Here, we report that many muscle genes contain both Rbfox1- and Bru1-binding motifs, and these motifs are closer together than expected by chance alone (Fig S5). We provide evidence that Rbfox1 and Bru1 regulate fiber type–specific splice events in *Fhos*, *Zasp67*, *Zasp66*, *Mhc*, *Tm1*, *Zasp52*, *sls*, *Strn-Mlck*, and *wupA* (Figs 6, S6, and 8), revealing instances of cooperative, opposing, and independent regulation. Moreover, Salm positively regulates *Rbfox1* levels in both IFMs and TDT (Fig 7). Interestingly, Rbfox1 regulation feeds back onto this regulatory hierarchy. Rbfox1 regulates a developmental switch in the use of *Mef2* 5′-UTR exons, as well as fiber type–specific expression levels of myogenic regulators, including Mef2, Salm, and Bru1 (Figs 5, 7, 8, and S7). Rbfox1 further directly binds 3′-UTR regions of *wupA* and *Mef2,* and interacts with RNA processing factors, including eIF4a and Rent1 (Figs 4I′, 7B, and S4G), indicating that it regulates transcript stability or translation in addition to its role in fiber type–specific splicing. We therefore conclude that Rbfox1 is a component of the fiber identity pathway in *Drosophila* adult muscle that interacts with Bru1 to regulate muscle type–specific isoform expression.

Although such a role for Rbfox1 in fiber type–specification has not been identified in other organisms, there are parallels with the reported functions of RBFOX proteins in vertebrate myofiber differentiation. RBFOX binding motifs are found to be co-enriched with MBNL and CELF motifs around the same groups of exons in humans, mice and chickens (Kalsotra et al, 2008; Bland et al, 2010; Merkin et al, 2012), and CELF2 and Rbfox2 co-regulate exons in cardiac tissue that show temporal use preferences, or are altered in hearts of a Type I diabetes mouse model (Gazzara et al, 2017). In mice, Rbfox1 and Rbfox2 regulate splicing of *Mef2D exon α2* during myotube differentiation, allowing Mef2D to escape inhibitory PKA signaling, and activate the late muscle gene expression program (Runfola et al, 2015). Rbfox1 also regulates splicing of a MEF2A exon in mouse and zebrafish heart that is mis-spliced in cells from human patients with dilated cardiomyopathy (Gao et al, 2016), and Rbfox1 and Rbfox2 cooperatively regulate splicing of Mef2D during C2C12 differentiation (Runfola et al, 2015). In *C. elegans*, FOX-1/ASD-1 and SUP-12 regulate a developmental switch in the expression of the fibroblast growth factor receptor *egl-15* that is necessary for myoblast migration and vulval muscle formation (Kuroyanagi et al, 2007; Mackereth, 2014). Thus, RBFOX proteins may generally promote developmental switches in myogenic regulatory and

structural genes, and it would be informative to investigate differences in Rbfox1/2 function between fiber types in vertebrate models.

### Rbfox1 modulates fiber type–specific transcriptional networks

Fiber type–specific isoform expression is modulated through RNA regulatory mechanisms, such as alternative splicing or 3′-UTR binding, as discussed above, but is also impacted by transcription. The conserved regulation of Mef2 by Rbfox proteins is particularly intriguing, as Mef2 is a key regulator of the expression level of most structural proteins during the assembly of the sarcomere (Molkentin et al, 1995; Gunthorpe et al, 1999; Tanaka et al, 2008; Taylor & Hughes, 2017). *Drosophila* Mef2 is not only expressed at different levels between fiber types, but some of its targets only respond to higher levels of Mef2 expression, providing a mechanism to modulate temporal and fiber type–specific expression dynamics (Elgar et al, 2008; Bryantsev et al, 2012a). Given the thin to thick filament ratio is 6:1 in fibrillar muscles, and 8–12:1 in the tubular muscles (Bernstein et al, 1993), misregulation of fiber type–specific Mef2 activity level in *Rbfox1* knockdown flies might contribute to the expression changes we observe in sarcomeric proteins, such as TnI and Act88F (Fig 4). Furthermore, knockdown of *Rbfox1* is able to partially rescue the hypercontraction phenotype in *wupA^{hdp-3}* splicing mutants, but not in the *wupA^{fliI}* mutant, where indels in the Mef2 binding sites likely desensitize *wupA* to Mef2 regulation (Fig 4). Misregulated expression of Mef2, Bru1 and Salm combined with isoform transitions in TnI, all favoured by low levels of Rbfox1, could generate a stoichiometric imbalance resulting in hypercontraction in the *Rbfox1* knockdown condition. It is also possible that Rbfox1 more directly regulates transcription. Although it is an RBP, Rbfox2 can interact with the Polycomb repressive complex 2 (PRC2) through a unique C-terminal domain and regulate transcription in mice (Wei et al, 2016). In *Drosophila*, Rbfox1 can interact with Cubitus interruptus (Ci) and Suppressor of Hairless (Su(H)), transcription factors in the Hedgehog (Hh) and Notch (N) signaling pathways, respectively, to regulate vein-intervein and sensory organ specification in the wing disc (Usha & Shashidhara, 2010; Shukla et al, 2017). Taken together, the multi-layered function of Rbfox1 in the fiber diversification network in flies suggests that combinatorial integration of RNA regulation and transcriptional feedback ultimately fine-tunes the expression level and ratio of structural protein isoforms. Such a mechanism may be broadly applicable to allow muscle fibers to flexibly adjust regulator levels during development, or to promote plasticity in response to exercise, aging, injury, or disease.

## Materials and Methods

A table of key resources is available as Table S4.

### Fly stocks and crosses

Approval for work with *Drosophila* under §15 GenTSV in Germany was granted under license number 55.1-8791-14.1099. Fly stocks were maintained using standard culture conditions. Wild-type controls include either *w^{1118}* or Canton-S. Rbfox1-GFP (*Rbfox1^{CC00511}*) was generated as part of a protein enhancer trap library (Kelso et al, 2004), and does not alter protein function or localization. Fly stocks of UAS-*Rbfox1*-RNAi and UAS-*Rbfox1* (Usha & Shashidhara, 2010) were kind gifts from L. Shashidhara, IISER. UAS-*Rbfox1*-RNAi, also called UAS-dA2BP1^{RNAi}, targets a 350 bp sequence in *Rbfox1* distinct from sequences targeted by lines in the Vienna and Bloomington/Harvard collections (Fig S1A). The deGrad-FP fly line *pUASP1-deGradFP/CyO; MKRS/TM6,Tb* (Caussinus et al, 2012) was a kind gift of Sonal Jaishwal, CCMB. deGradFP knockdown was carried-out during adult IFM development by temperature shifts of late third instar larvae (L3). *Mhc^{P401S}* (Nongthomba et al, 2003) is a myosin mutant that minimizes actomyosin force in IFMs, whereas *wupA^{fliH}* (Firdaus et al, 2015) and *wupA^{hdp3}* (Barbas et al, 1993) are known hypercontraction mutants in *wupA* (TnI). RNAi lines were obtained from the Vienna Drosophila Resource Center (VDRC) including UAS-*Arrest*-RNAi (*Bru1-IR*) (41547, 48237, and 41568) (Dietzl et al, 2007; Oas et al, 2014; Spletter et al, 2015), UAS-Salm-RNAi (*salm-IR*) (3029, 101052) (Schönbauer et al, 2011), UAS-*Rbfox1*-IR^{KK110518} (110518) or from the Bloomington Drosophila Stock Center (BDSC) UAS-*Rbfox1*-IR^{27286} (TRiP27286, JF02600). *Rbfox1*-IR^{KK110518} has one predicted off-target (1 of 305 total 19mers is found in CG11966, *ichor*), but *ichor* is not expressed in IFMs, TDT, or leg samples based on mRNA-Seq data. *Rbfox1*-IR^{27286} has 0 predicted off-targets. UAS-Mef2 lines were provided by Alberto Ferrus (Gunthorpe et al, 1999). UAS-Bru1-PA (also called UAS-Arrest) was kindly provided by Richard Cripps (Oas et al, 2014) and expresses the full-length *bru1-RA* mRNA from DGRC clone LD29068. A second UAS-Bru1-PA line was generated by cloning the full-length *bru1-RA* cDNA (obtained by RT–PCR from *w^{1118}*) into the *pUAS-TattB* transformation vector (Bischof et al, 2007) and integrating into the attP-86Fb landing site. The *bru1^{M2}* and *bru1^{M3}* alleles were generated using a CRISPR approach (Zhang et al, 2014). In the *bru1^{M2}* mutant, a selectable 3xP3-DsRed cassette was inserted upstream of *bru1* exon 12. This results in a strong hypomorphic allele because of redirection of *bru1* splicing into the splice acceptor contained in the cassette, which is followed by a triple frame stop and polyadenylation sequence (Fig S5A–E). In the *bru1^{M3}* mutant, the same selectable 3xP3-DsRed cassette is inserted upstream of *bru1* exon 18. sgRNA sequences, homology arm primers, and primers for testing transcripts produced in *bru1^{M2}* mutants are listed in Table S3. Gal4 drivers used were: Mef2-Gal4 (Ranganayakulu et al, 1996), which drives in all muscle (maintained at 27°C or 29°C); UAS-Dcr2, Mef2-Gal4 which enhances RNAi efficiency (maintained at 22°C); Act5c-Gal4, which drives in all cells (maintained at 27°C and 25°C); Mhc-Gal4 (Davis et al, 1996), which drives in muscle from 40 h APF; UH3-Gal4 (Singh et al, 2014) is a driver with IFM specific expression after 36–40 h APF (maintained at 27°C); Act88F-Gal4 (Bryantsev et al, 2012a) is a driver with IFM specific expression after 24 h APF (maintained at 25°C) and Act79B-Gal4 (Dohn & Cripps, 2018) is a driver with TDT specific expression (maintained at 27°C). Temperature sensitive *Tubulin-Gal80^{ts}*, as noted in figure panels and legends, was used to restrict some knockdown experiments to adult muscle development by a temperature shift of late third instar larvae from 18°C to 29°C. Rbfox1 overexpression with UH3-Gal4 was induced 40 h APF onwards to avoid lethality at earlier stages.

### Behavioral assays

Flight behavior was tested as described previously (Drummond et al, 1991), or by introducing 30 adult males flies into a 1-m long

cylinder divided into five zones (Schnorrer et al, 2010). Flies landing in the top two zones are "normal fliers," in the middle two zones are "weak fliers," and at the bottom are "flightless." Pupal eclosion (survival) was determined by counting the number of flies that eclose from at least 50 pupae of the appropriate genotype. Climbing ability was assayed using a modified rapid iterative negative geotaxis approach (Nichols et al, 2012). Adult males were collected on $CO_2$ and recovered at least 24 h before testing three times with a 1-min recovery period for their ability to climb 5 cm in a 3 or 5 s timeframe. Jumping ability was assayed as described previously (Chechenova et al, 2017). After clipping the wings and 24-h recovery, 10–15 males were individually placed on A4 paper and gently pushed with a brush to stimulate the jump response. The start and the landing points were marked and the distance was calculated in centimetres.

### Rabbit anti-Bruno1 antibody generation

The divergent domain (DIV) region of Bru1 or the complete *bru1-RA* cDNA sequence was cloned using SLIC into pCOOFY4 to generate His6-MBP-DIV or His6-MBP-RA, respectively. Primer sequences are listed in Table S3. Fusion to MBP was necessary to maintain solubility. The protein was expressed in *Escherichia coli* BL21-RIL cells and induced with 0.2 mM IPTG at 60°C overnight. Expressed protein was purified over Ni-NTA beads and then cleaved with HRV3C-protease. MBP was depleted by incubation with Amylose beads. Protein was then dialyzed in buffer (200 mM NaCl, 50 mM Tris, and 20 mM Imidazole) and sent as purified protein for antibody production (Pineda). Rabbit polyclonal antibodies against the DIV domain were generated by Pineda according to a standard 120-d protocol. Resulting serum was affinity purified over an MBP column (to remove background antibodies generated against the MBP protein) followed by a column with beads coupled to Bru1-PA. Antibody bound to the column was eluted in citric acid and buffered to pH 7. Antibody was directly frozen in small aliquots in liquid nitrogen and stored at −80°C until use.

### Immunofluorescence and microscopy

Fly hemithoraces were prepared for polarized microscopy as described previously (Nongthomba & Ramachandra, 1999). The hemithoraces were observed in an Olympus SZX12 microscope and photographed using Olympus C-5060 camera under polarized light optics. For confocal microscopy, flies were bisected, fixed in 4% paraformaldehyde for 1 h, washed with 0.3% PBTx (0.3% Triton X in PBS) for 15 min, and stained with 1:250 phalloidin-TRITC for 20 min. Sections were mounted on slides after washes with PBTx. Images were obtained using a Carl Ziess LSM 510 META confocal microscope.

Alternatively, IFMs and Abd-M were dissected and stained as previously described (Weitkunat & Schnorrer, 2014). All tissues were fixed for at least 30 min in 4% PFA in 0.5% PBS-T (1× PBS + Triton X-100). For visualization of IFMs, thoraces were cut longitudinally with a microtome blade. Abd-M was fixed on a black silicon dissection dish, after the ventral part of the abdomen was carefully removed together with fat, gut and other non-muscle tissues. TDT (jump) muscle was exposed by opening the cuticle sagittally using

fine biological forceps. One tip of the forceps was kept parallel to the fly thorax and gently inserted into the wing socket, allowing the initial split of the cuticle without damaging underlying tissues. The remaining cuticle covering the T2 mesothorax region, ventrally from the leg socket up to the dorsal bristles, was carefully removed to expose the underlying TDT muscle. Samples were blocked for 90 min at room temperature in 5% normal goat serum in PBS-T and stained with primary antibodies overnight at 4°C. Rabbit anti-Bru1 (1:500) and mouse anti-Lamin (ADL67.10, 1:100; DSHB) were used for staining. Samples were washed three times in 0.5% PBS-T for 10 min and incubated for 2 h at room temperature with secondary conjugated antibodies (1:500) from Invitrogen (Molecular Probes), including Alexa 488 goat anti-rabbit IgG, Alexa 647 goat anti-mouse IgG, and rhodamine-phalloidin. Samples were washed three times in 0.5% PBS-T and mounted in Vectashield containing DAPI.

Confocal images were acquired on a Leica SP8X WLL upright using Leica LAS X software in the Core Facility Bioimaging at the Biomedical Center of the Ludwig-Maximilians-Universität München. Whole fly thorax images were taken with a HCPL FLUOTAR 10×/0.30 objective and detailed sarcomere structure was imaged with a HCPL APO 63×/1.4 OIL CS2 objective. Bru1 signal intensity was recorded at the same laser gain settings adjusted on the brightest control sample for each muscle type. All samples of same replicate were stained with the same antibody mix on the same day and imaged in the same imaging session.

### RNA isolation and RT–PCR

For *Rbfox1*-RNAi experiments, 30 flies were bisected and dehydrated in 70% ethyl alcohol overnight. IFMs or TDT was dissected, homogenised and RNA isolated using TRI Reagent (Sigma-Aldrich) following the manufacturer's instructions. RNA was confirmed using readings from NanoDrop software, and was converted to cDNA using a first-strand cDNA synthesis kit (Fermentas). Primers and PCR conditions are listed in Table S3.

For *Rbfox1*-IR[KK110518] and *Rbfox1*-IR[27286] experiments, IFMs (from 30 flies) or TDT (from 60 flies) were dissected as previously described (Kao et al, 2019). For Abd-M, abdominal carcass was prepared from 15 flies in pre-cooled 1× PBS using fine biological forceps to remove fat, gut, trachea, and other non-muscle tissues through a posterior cut in the abdomen. The abdomen was then removed from the thorax using fine scissors and snap-frozen in 50 µl of TRIzol (TRIzol Reagent; Ambion) on dry ice and immediately stored at −80°C. For whole thorax preparations, 15 flies were placed in pre-cooled 1× PBS and then the head, wings and abdomen were removed using fine scissors. The thorax sample was further processed as described for Abd-M. Dissection times were limited to a maximum of 30 min. RNA was isolated using TRIzol according to the manufacturer's protocol. Total RNA samples were treated with DNaseI (New England Biolabs) and measured on a Qubit 2.0 Fluorometer (Invitrogen). Comparable total RNA quantities were used for reverse transcription with LunaScript RT SuperMix Kit (New England Biolabs). cDNA was amplified with Phusion polymerase for 30–36 cycles and resulting PCR products were separated on a standard 1% agarose gel next to a 100 bp or 1 kb ladder (New England Biolabs). *Ribosomal protein L32* (*RpL32*, also called *RP49*) served as an internal control for normalization in all reactions.

Where applicable, the target band was carefully cut out of the agarose gel on a table UV-lamp, isolated from the gel using the mi-Gel Extraction kit (Metabion), and sent for sequencing with an appropriate primer (GATC-Biotech; Eurofins). All PCR primers are listed in Table S3.

### RIP followed by cDNA synthesis

The RIP protocol was modified from Carreira-Rosario et al (2016). Approximately 500 mg of thoraces (from *Rbfox1^CC00511* cultured flies) were lysed in 1 ml of RIPA buffer (50 mM Tris–HCl, 200 mM NaCl, 0.4% NP-40, 0.5% sodium deoxycholate, 0.1% SDS, 2 mM EDTA, and 200 mM NaCl) with Sigma-Aldrich RNAse inhibitor, pre-cleared with Protein-G magnetic Dynabeads, and incubated with mouse anti-GFP (12A6; Developmental Studies Hybridoma Bank [DSHB]) or IgG isotype (purified from normal mouse serum). The beads with immunoprecipitated RNA bound to Rbfox1-GFP were washed and treated with Proteinase K (25 min in 37° C), followed by a TRI reagent–based RNA extraction, cDNA synthesis, and PCR as described above.

### Protein extraction and Western blotting

For *Rbfox1*-RNAi experiments, IFMs from 20 flies were dissected, "skinned," and thin filaments extracted as previously described (Vikhorev et al, 2010). These samples were run on SDS–PAGE and transferred onto a nitrocellulose membrane (product no. IPVH00010; Milipore), using a semi-dry transfer apparatus. Blots were stained with rabbit anti-Actin or rabbit anti-TnI (1:1,000; a gift from A Ferrus) or mouse anti-Tubulin (1:1,000; Sigma-Aldrich) and washed with TBS-Triton X (0.1%). Blots were incubated with HRP-conjugated secondary anti-rabbit or anti-mouse antibodies (1:5,000 in TBS-Triton X), washed and developed on an X-ray film in the dark.

For *Rbfox1*-IR^KK110518 and *Rbfox1*-IR^27286 experiments, IFMs from 8 flies, TDT from 20 flies or Abd from 6 flies was dissected as described above. Samples were homogenised in 20 $\mu$l of freshly made SDS-buffer (2% SDS, 240 mM Tris, pH 6.8, 0.005% bromophenol blue, 40% glycerol, and 5% $\beta$-mercaptoethanol), incubated at 95°C for 3 min and stored at –20°C. Samples were run on 10% SDS–PAGE for separation and then transferred onto nitrocellulose membranes (Amersham Protran 0.2 $\mu$m NC) for 2 h at 120 V. Membranes were stained with Ponceau S (Sigma-Aldrich) to access the quality of the blotting. Membranes were de-stained and blocked with 5% non-fat milk solution in 0.5% Tween-TBS buffer (T-TBS) for 1 h, washed and incubated for 1 h at room temperature with primary antibodies (rabbit anti-Bru1, 1:500; rabbit anti-H2AZ, 1:2,000). Membranes were washed three times with T-TBS for 15 min and incubated with goat anti-rabbit HRP-conjugated secondary antibodies (Bio-Rad) for 1 h at room temperature. After three rounds of washes, the membranes were developed using Immobilion Western chemiluminescent (Milipore) substrate and exposed to X-ray films (Fuji medical X-ray, Super RX-N) or imaged on a ChemiDoc MP (Bio-Rad).

### Co-immunoprecipitation and mass spectrometry

Approximately 500 mg of thoraces (from *Rbfox1^CC00511* cultured flies) were lysed in 1 ml of RIPA buffer with Sigma-Aldrich protease inhibitor mix, pre-cleared with Protein-G magnetic Dynabeads (10030D; Thermo Fisher Scientific), and incubated with mouse anti-

GFP (12A6; DSHB) or IgG isotype (purified from normal mouse serum). The beads with immunoprecipitated proteins bound to Rbfox1-GFP were washed in RIPA buffer, followed by protein elution and denaturation, as described previously (Carreira-Rosario et al, 2016). Proteins were analysed by SDS–PAGE and unique bands were cut and processed for mass spectrometric analysis following the protocol provided by the Proteomics facility, Molecular Biophysics Unit, Indian Institute of Science.

### Image analysis

Confocal image analysis was performed with Image J/Fiji (Schindelin et al, 2012). For every experiment, 10–15 images were acquired from at least 10 individual flies. Fiber detachment was scored from Z-stacks of whole thorax images. Sarcomere length and width were measured using MyofibrilJ ([Spletter et al, 2018], https://imagej.net/MyofibrilJ) based on rhodamine-phalloidin staining. Sarcomere length and width plots generated in GraphPad Prism are shown with Tukey whiskers, where whiskers are drawn to the 25th and 75th percentile plus 1.5 times the interquartile range. Dots above or below the whiskers represent outlying data points outside of this range. Analysis of Bru1 intensity was performed manually in Fiji from at least three nuclei per image. Analysis of semi-quantitative RT–PCR gels and Western blots was performed using the "gel analysis" feature in Fiji. *RpL32* and H2AZ were used as internal normalization controls for RT–PCR and Western analysis, respectively. Fold change was calculated by dividing the normalized intensity in a knockdown sample by the normalized intensity of the control run in the same PCR replicate and on the same gel. The percentage of exon use (% of signal) for alternative splice events assayed by RT–PCR was calculated as: $100 \times (individual\ band\ intensity)/\sum(intensity\ of\ all\ bands)$ generated by the same primer pair. Data were aggregated in Microsoft Excel. Plotting and statistical analysis were performed in GraphPad Prism 9.

### Bioinformatics

Rbfox1 has been identified to bind (U)GCAUG motifs in both vertebrates and *Drosophila* (Pedrotti et al, 2015; Nazario-Toole et al, 2018), and RBP binding specificity in the form of a position weight matrix (PWM) has been determined in vitro for both Rbfox1 and Bru1 using RNACompete (Ray et al, 2013; Bouvrette et al, 2020). There are no published RNA CLIP data available from *Drosophila* muscle, so we do not know genome-wide which motif sequences are physically bound by Rbfox1 or Bru1. As a viable proxy to identify possible Rbfox1 targets in muscle, we downloaded all motif instances for the Rbfox1 and Bru1 PWMs in the transcriptome from the oRNAment database (http://rnabiology.ircm.qc.ca/oRNAment), and searched for and downloaded genome-wide instances of the PWMs using PWMScan (https://ccg.epfl.ch/pwmtools/). PWM graphic summaries generated by PWMScan are presented in Fig S4H. For PWMScan data, the BED output was converted to a GRanges object in R, and sequence locations mapping to intron, exon, CDS, 5′-UTR or 3′-UTR regions (based on Flybase dmel_r6.38 annotation files) were isolated. Gene identifiers were assigned based on genomic coordinates, and sequences were filtered to match gene orientation (i.e., to retain sequences present in the transcribed pre-mRNA). oRNAment data were imported into R in the form of a GRanges object.

All analysis and plotting were performed in R using packages listed in Table S4.

Lists of genes from oRNAment or PWMScan with Rbfox1 motif instances in specific genomic locations (introns, CDS, 5′-UTR, or 3′-UTR regions) were subjected to enrichment analysis using GOrilla (Eden et al, 2009) or with custom gene sets (Spletter et al, 2018). GOrilla term lists were reduced using the rrvgo package. Expected numbers of genes with motifs in Fig S3A were calculated by averaging 150 simulations of how many genes in a random set of 100, 500, 100, or 3,000 genes contained an Rbfox1 motif. Plots of Rbfox1 and Bru1 binding motif locations in selected genes were generated based on the UCSC visualization option in PWMScan and the IGV option in oRNAment. The distance from an Rbfox1 motif to the nearest Bru1 motif in Fig S4J was calculated using the distanceToNearest algorithm in the GenomicRanges package in R. The expected distributions for all genes or genes with a muscle phenotype were generated by averaging the results from 50 simulations of the distance from Rbfox1 to the nearest Bru1 motif if Bru1 motifs were randomly distributed across the transcriptome (random sets of motif coordinates in exons were generated using the random.intervals algorithm from the seqbias package).

mRNA-Seq data used in this study has been published previously (Spletter et al, 2015, 2018) and is available from the Gene Expression Omnibus (GEO) under accession numbers GSE63707, GSE107247, and GSE143430. Data were mapped with STAR to ENSEMBL genome assembly BDGP6.22 (annotation dmel_r6.32 [FB2020_01]), indexed with SAMtools, and features counted with featureCounts. Downstream analysis and visualization were performed in R using the packages listed in Table S4. Differential expression was analysed with DESeq2 and DEXSeq, which additionally generated normalized counts values. Read-tracks were visualized on the UCSC Genome Browser. Splice junction reads were exported from STAR, and junction use for hand-selected events was calculated as: (number of reads for select junction $D^1A^x$)/(total number of reads $D^1A^1 + D^1A^2 ... + D^1A^n$) × 100, where D stands for donor and A for acceptor. In this way we could determine the percent of junction reads from a given donor that use acceptor "x," or swap A and D to determine the percent of junction reads from a given acceptor coming from donor "x."

## Data Availability

Raw numbers used to generate plots are available in the source data files accompanying each figure. mRNA-Seq data are publicly available from GEO with accession numbers GSE63707, GSE107247, and GSE143430.

## Supplementary Information

## Acknowledgements

We sincerely thank LS Shashidhara, R Cripps, A Ferrus, Sonal Jaishwal, Frank Schnorrer, the BDSC, the VDRC, and the Drosophila Stock Facility at the National Centre for Biological Sciences (NCBS), Bangalore, India, for providing flies. ML Spletter is grateful to Andreas Ladurner for generous support. We thank John Sparrow and Shao-Yen Kao for inputs on manuscript preparation and critical comments. We thank Sandra Esser for excellent technical assistance. We acknowledge the Core Facility Bioimaging at the LMU Biomedical Center (Martinsried, DE) for confocal imaging support. We acknowledge the Indian Institute of Science (IISc), the Department of Science and Technology (DST) (DST FIST, 2008–2013 Ref. No. SR/FST/LSII-018/2007), the University Grant Commission (UGC-SAP to MRDG: Ref. No. F.3-47/2009 [SAP-II]) and the Department of Biotechnology (DBT), Govt. of India (DBT-IISC Partnership Program for Advanced Research in Biological Sciences & Bioengineering Sanction Order No: DBT/BF/PRIns/2011-12/IISc/28.9.2012), the Deutsche Forschungsgemeinschaft (ML Spletter, 417912216), the Center for Integrated Protein Science Munich at the Ludwig-Maximilians-University München (ML Spletter), the Deutsche Gesellschaft für Muskelkranke e.V. (ML Spletter), and the International Max Planck Research School (E Nikonova) for financial assistance.

## Author Contributions

E Nikonova: formal analysis, validation, investigation, visualization, and writing—review and editing.
A Mukherjee: formal analysis, validation, investigation, visualization, and writing—original draft, review, and editing.
K Kamble: formal analysis, validation, investigation, visualization, and writing—original draft, review, and editing.
C Barz: investigation and writing—review and editing.
U Nongthomba: conceptualization, supervision, funding acquisition, and writing—original draft, review, and editing.
ML Spletter: conceptualization, formal analysis, supervision, funding acquisition, validation, and writing—original draft, review, and editing.

## Conflict of Interest Statement

The authors declare that they have no conflict of interest.

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
