## [Reviewer comments · Life Science Alliance]

Life Science Alliance

Rbfox1 is required for myofibril development and maintaining fiber-type specific isoform expression in *Drosophila* muscles

Elena Nikonova, Amartya Mukherjee, Ketaki Kamble, Christiane Barz, Upendra Nongthomba, and Maria Spletter

DOI: <https://doi.org/10.26508/lsa.202101342>

Corresponding author(s): Maria Spletter, Ludwig-Maximilians-Universität München and Upendra Nongthomba, Indian Institute of Science

Review Timeline:

Submission Date:	2021-12-16
Editorial Decision:	2021-12-20
Revision Received:	2021-12-22
Accepted:	2021-12-23

Transaction Report:

Please note that the manuscript was reviewed at *Review Commons* and these reports were taken into account in the decision-making process at *Life Science Alliance*.

1st Authors' Response to Reviewers

1. General Statements

We thank the reviewers for their helpful, detailed and insightful comments. We have modified the figures and rewritten large sections of the manuscript following the reviewers' suggestions. In addition, we have incorporated new data throughout the manuscript and figures to clarify and better support our conclusions. All of these changes have significantly improved the coherence, consistency and clarity of our data, and have allowed us to better communicate the advance our findings represent for the fields of splicing and muscle development.

Please find a point-by-point response to the reviewers' comments below. Our response is in blue, while the reviewers' comments are in black and italics.

Response to Reviewer 1

Reviewer #1 (Evidence, reproducibility and clarity (Required)):

Rbfox proteins regulate skeletal muscle splicing and function and in this manuscript, Nikonova et.al. sought to investigate the mechanisms by which Rbfox1 promotes muscle function in Drosophila.

Using a GFP-tagged Rbfox1 line, the authors showed that Rbfox1 is expressed in all muscles examined but differentially expressed in tubular and fibrillar (IFM) muscle types, and expression is developmentally regulated. Based on RNA-seq data from isolated muscle groups, the authors showed that Rbfox1 expression is much higher in TDT (jump muscle) than IFM.

*Using fly genetics authors developed tools to reduce expression of Rbfox1 at different levels and the highest levels of muscle-specific Rbfox1 knockdown was lethal and displayed eclosion defects (*deGradFP > Rbfox1-IRKK110518 > Rbfox1-RNAi > Rbfox1-IR27286*). Consistently, Rbfox1 knockdown flies have reduced jumping and climbing phenotypes, due to tubular muscle defect where Rbfox1 is expressed at higher levels. Rbfox1 knockdown in IFM caused flight defects which have been shown previously. Further characterization of IFM and tubular muscles demonstrated a requirement of Rbfox1 for the development of myofibrillar structures in both fibrillar (IFM) and tubular fiber-types in Drosophila. Interestingly, knockdown or overexpression of Rbfox1 displayed hypercontraction phenotypes in IFMs which is often an end result of misregulation of acto-myosin interactions which was rescued by expression of force-reduction myosin heavy chain (*Mhc, P401S*), in the context of Rbfox1 knockdown (the rescue experiment could not be performed with Rbfox1 overexpression due to complex genetics).*

*Authors also performed computation analyses of the Rbfox binding motifs in the fly genome and identified GCAUG motif in 3,312, 683, and 1184 genes in the intronic, 5'UTR, and 3'UTR, respectively. These genes are enriched for factors that play important roles in muscle function including transcription factors (*exd, Mef2, Salm*), RNA-binding proteins (*Bru1*), and structural proteins (*TnI, encoded by wupA*). Many of these gene transcripts and proteins are affected in flies with reduction or overexpression of Rbfox1. Using fly genetics, authors propose and test different mechanisms (co-regulation of gene targets by Rbfox1 and Bru1), and regulators of muscle function (*exd, Me2, Salm*) and structural proteins (*TnI, Mhc, Zasp52, Strn-Mlck, Sls*) by which these changes could affect the muscle function.*

Overall, the characterization of Rbfox1 phenotypes and myofibrillar structure is very well elucidated, mechanisms by which Rbfox1 affects muscle function are not clear and remain largely speculative.

We thank the reviewer for the positive evaluation of our phenotypic analysis of Rbfox1 knockdown in multiple muscle fiber types. This manuscript is the first detailed characterization of Rbfox1 in Drosophila muscle, extending far beyond our previous finding that Rbfox1-IR flies are flightless. Beyond behavioral and cellular phenotypes, we report that there are regulatory interactions between Rbfox1, Bruno1 and Salm and identify other Rbfox1 targets in flies. We acknowledge that there are molecular and biochemical details of specific regulatory mechanisms that remain to be elucidated, but this paper provides many foundational observations to guide future biochemical experiments and is thus important to the muscle field.

****Major comments****

1. The varying level of *Rbfox1* knockdown (*deGradFP* > *Rbfox1-IR^{KK110518}* > *Rbfox1-RNAi* > *Rbfox1-IR²⁷²⁸⁶*) was achieved by different strategies without validation at the protein level (likely due to lack of a *Rbfox1* antibody). It is important to show different *Rbfox1* protein level (at least with different RNAi), especially when authors propose that autoregulation of *Rbfox1* causes increased level *Rbfox1* transcript in case of *Rbfox1-RNAi* (mild knockdown). Autoregulation of *Rbfox1* in mammalian cells may not be similar in flies.

To address this comment, we have toned-down the discussion of level-dependent regulation throughout the manuscript, and have removed claims of *Rbfox1* autoregulation. We appreciate the reviewer's point that it would be ideal to be able to determine the protein levels of *Rbfox1* in the different knockdown conditions. We have tested the published

antibody against Dm*Rbfox1*, but it is very dirty and we see multiple bands in Western Blot. This background partially obscures the bands from 80-90 kDa at the molecular weight where we expect *Rbfox1*, and prevents accurate quantification (see Reviewer Figure 1). Verification of protein levels of *Rbfox1* will require generation of a new antibody which is beyond the scope of this study. As we do not have a good antibody, we performed two experiments to demonstrate our ability to tune knockdown efficiency. First, we crossed *Rbfox1-IR^{KK110518}* and *Rbfox1-IR²⁷²⁸⁶* to UAS-Dcr2, Mef2-Gal4 and demonstrated we could enhance the phenotype (Figure 2A, B). Second, we performed knockdown with the same hairpins at different temperatures and demonstrate that stronger knockdown at higher temperature leads to stronger phenotypes with the same hairpin (Figure 2B). This data supports our knockdown series interpretation.

Reviewer Figure 1. Western Blot of whole fly with anti-*Rbfox1* (A2BP1) (Shukla et al., 2017). Tubulin was blotted as a loading control.

2. *Tnl* and *Act88F* protein levels are inversely correlated with *Rbfox1* level in IFM but did not correlate with the RNA level. Using RIP authors showed that *Rbfox1* was shown to bound to *wupA* transcripts (has *Rbfox* binding sites) but not *Act88F* transcripts (does not have *Rbfox* binding sites). Authors performed *Rbfox1* IP and identified co-IP of components of cellular translational machinery and propose that *wupA* (*Tnl*) levels are regulated by translation or NMD (non-sense mediated decay). A follow up experiment was not performed to identify the mechanism by which *Tnl* level is regulated by *Rbfox1*.

Further biochemical and genetic verification of the underlying mechanisms of *Rbfox1* regulation in *Drosophila* muscle will be addressed in a future manuscript, as *in vivo* modulation of translation or NMD in an *Rbfox1* knockdown background involves recombination to coordinate multiple genetic elements. We have modified the text to reflect this hypothesis remains to be explored in future experiments (Line 473-474).

We have further added RT-PCR data for *wupA* transcript levels in IFM and TDT with *Rbfox1-IR^{KK110518}* knockdown (Figure S4 A), but as in *Rbfox1-RNAi* flies, there is not a significant change in expression. We do see significant downregulation of *Act88F* when we overexpress *Rbfox1* in IFM (Figure S4 B), as well as in TDT when we knockdown *Rbfox1* with either *Rbfox1-IR^{KK110518}* or *Rbfox1-IR²⁷²⁸⁶*.

3. It was known that *Tnl* mutations (affects splice site, *fliH* or Mef2 binding site, *Hdp-3*) led to a reduction in *Tnl* level and hypercontraction. Authors showed rescue of hypercontraction phenotype in *hdp-3* background by knocking down *Rbfox1*, likely due to increase in *wupA* transcription (Mef2-dependent or independent manner). However, no rescue was observed in the *fliH* background. Reduced level of *Rbfox1* in *fliH* background would be expected to cause worsening of phenotype

as splicing of remaining *wupA* transcripts would be affected with reduced *Rbfox1* level. The splicing of *wupA* of exon 4 is not affected in *Rbfox1* knockdown (fig. 6U), it's not clear if the splicing of exon 6b1 is affected in *Rbfox1* knockdown.

We thank the reviewer for pointing out our lack of clarity regarding exon 6b1 and IFM-specific isoform 6b1. To address this comment and validate our previous data, we performed additional Sanger sequencing on RT-PCR products, added a diagram of the *wupA* gene region in Figure 4 A and improved the clarity of our discussion of the *fliH* and *hdp3* alleles and our results in the text.

To directly respond to the reviewer, first, it is unclear if the reduced level of *Rbfox1* in a *fliH* background should actually cause a more severe phenotype. Our data suggests that *Rbfox1* represses *TnI* expression through binding the 3'-UTR, and can likely indirectly regulate *wupA* expression level via *Mef2*. Thus, arguably, the reduced level of *Rbfox1* in the *fliH* background might not affect splicing, as the mutations in the regulatory element should rather make *wupA* insensitive to increased *Mef2* expression in the *Rbfox-RNAi* background.

Second, we confirmed via Sanger sequencing of RT-PCR products that both IFM and TDT in control and *Rbfox1-IR* flies use exon 6b1 (current exon 7). The IFM isoform contains exon 3, 6b1 and 9, while the TDT isoform contains exon 3 and 6b1, but skips exon 9 (see Figure 4 A). In other tubular muscles, *wupA* isoforms skip exons 3 and 9, and use exon 6b2 instead of 6b1. Thus, to directly answer the reviewer's question, no, splicing of exon 6b1 itself is not affected by *Rbfox1*. However, *Rbfox1* does influence expression of the "6b1 isoform", or the *wupA* isoforms in IFM and TDT containing exon 6b1 and exon 3. Additionally, our data shows that *Bru1*, not *Rbfox1*, regulates alternative splicing of *wupA* exon 9 (Fig. S6 T).

What the reviewer has correctly identified with this comment is that the effect on splicing in the *hdp-3* allele also appears to be complex and to have not been fully clarified. Although *hdp-3* results from mutation of a splice site in exon 6b1 (which based on (Barbas et al., 1993) results in aberrant use of 6b2 in IFM), it also results in a near complete absence of the longer isoform containing exon 3 in adult flies. *hdp-3* is reported in the same paper to affect both IFM and TDT, which both express isoforms containing exon 3 and 6b1. It is not known how mis-splicing of exon 6b1 leads to loss of isoforms containing exon 3, but our data indicate that *Rbfox1* is somehow involved. It is purely speculation and beyond the scope of this manuscript, but perhaps selection of alternative exons in *wupA* are not independent events (ie that the splicing of exon 3 depends on correct splicing of exon 6b1). This could be mediated with interactions with chromatin, the PolII complex or through a larger splicing factor complex (something like LASR, for example (Damianov et al., 2016)), that restricts choice in alternative events through higher-order interactions. Another possible mechanism is that a second mutation exists in the *hdp-3* allele that affects splicing of exon 3, although this was not indicated in the extensive sequencing data in (Barbas et al., 1993).

4. *Bruno1* was identified as a co-regulator of *Rbfox1* in different IFM and tubular muscle types. However, except *Mhc*, other *Rbfox1* targets seem to be regulated by either *Rbfox1* or *Bruno1*, not both. Analyses of RNA-seq datasets from single and double knockouts should identify additional targets to support the claim that - *Rbfox1* and *Bruno1* co-regulate alternative splice events in IFMs. Phenotypic changes with reduced *Rbfox1* and *Bruno1* double knockdowns are very severe, but the mechanistic basis of such genetic interaction resulting in synergistic phenotypes in IFMs is lacking as splicing changes in single vs double knockout is similar.

We agree with the reviewer that RNA-seq data would be useful to obtain a genome-wide perspective on the regulatory interactions between *Rbfox1* and *Bru1*, and we plan to generate this data as part of a future manuscript. However, the tissue-specific dissections to isolate enough material from all of the necessary genotypes will take months to complete, and are not realistic to wait to include in this manuscript. Instead, to address the reviewer's question, we have expanded our RT-PCR experiments to cover a wider panel of events in 12 sarcomere genes (see new data in Figures 6 and S6 and summary in Figure 8). We now can show that splice events in *Fhos* and *Zasp67* are *Rbfox1* dependent, while events in *sIs*, *Strn-Mlck* and *wupA* are *Bru1* dependent. An event in *Zasp66* responds to both *Rbfox1* and *Bru1*, but in opposite directions. Events in *Mhc*, *Tm1* and *Zasp52* are regulated by both *Rbfox1* and *Bru1* (or are sensitive to changes in *Bru1* expression in the *Rbfox1* background), and change in the same direction. This data provides a clearer mechanistic basis for the synergistic phenotype observed between *Rbfox1* and *Bru1* in IFM.

5. *Rbfox1* is expressed at a high level in tubular muscle whereas *Bruno1* is expressed at a high level in IFM. *Rbfox1* binds to *Bruno1* transcript and inversely regulates *Bru1-RB* level but knockdown of *Bru1* does not affect *Rbfox1* level (Fig. S5 G,I,J). Overexpression of *Bruno1* decreased the *Rbfox1* level, however, it's difficult to interpret these results as overexpression of *Bruno1* may have other effects on IFM gene expression.

The reviewer correctly pointed out that we did not observe significant changes in *Rbfox1* mRNA levels in the mutant *bru1^{M3}* background, however, in the original version of this manuscript, we also showed a significant decrease in

Rbfox1 expression in IFM from the *bru1-IR* background at both 72 h APF and 1 d adult in mRNA-Seq data. To clarify differences in *Rbfox1* levels between *bru1-IR* and our *bru1* mutant backgrounds, we have performed additional RT-PCR experiments. We examined *Rbfox1* levels after knockdown of *bru1* (*bru1-IR*), and we now show that *Rbfox1* levels are significantly decreased in IFM and TDT after *bru1-IR* (Fig. 5S, Fig S5 I). We see a weaker effect in the *bru1^{M2}* hypomorphic mutant, which likely reflects differences in Bru1 expression levels in *bru1-IR* and the *bru1^{M2}* allele. These results are consistent with the mRNA-seq data we presented previously (now in Fig. 5R). These additional data suggest that loss as well as gain of Bru1 affects *Rbfox1* expression levels.

6. A dose-dependent effect of *Rbfox1* knockdown was shown to regulate the expression of transcription factors that are important for muscle type specification and function including *exd*, *Mef2*, and *Salm*. However, it is not clear how *Rbfox1* mechanistically regulates the expression of these transcription factors.

We present two pieces of data suggesting possible regulatory mechanisms for *Mef2*. First, RIP data suggest *Rbfox1* can directly bind the 3'-UTR region of *Mef2*, and this region contains two binding motifs identified in both the oRNAmnt database and in our PWMScan dataset. Second, we show that use of the 5'-UTR regions of *Mef2* is altered in *Rbfox1-IR* muscle. Although not definitive, this suggests that regulation of alternative 5'-UTR use may influence transcript stability or translation efficiency. We feel the many experiments to elucidate the detailed mechanism of regulation (and indeed to determine the likely contribution of multiple, layered regulatory processes) are beyond the scope of this paper, and are better left for future studies. This manuscript is the first in-depth characterization of *Rbfox1* function in *Drosophila* muscle, and we provide multiple lines of evidence suggesting that different regulatory mechanisms exist as a basis for future experiments to explore these interesting and important regulatory interactions.

****Minor comments****

1. It is not described if the rescue of *Rbfox1* knockout by expression of force-reduction myosin heavy chain (*Mhc*, *P401S*) led to rescue of phenotypes (jumping, climbing, flight).

Force-reduction myosin heavy chain *Mhc^{P401S}* is a mutation at the endogenous *Mhc* locus that results in a headless myosin and was previously characterized to be flightless (Nongthomba et al., 2003). It is however able to rescue jumping and walking defects observed with the *hdp²* Tnl allele, and supports largely normal myofibril assembly (Nongthomba et al., 2003). It is also important to note that fibrillar muscle function is very finely tuned, such that alterations that result in flightlessness in many cases do not alter myofibril structure as detected by confocal microscopy (Schnorrer et al., 2010). We therefore looked at myofiber and sarcomere structure as a more sensitive read-out of the rescue ability in the *Rbfox1* knockdown, to be able to detect a partial-rescue of myofibrillar structure that may not be evident in a behavioral assay.

2. Immunofluorescence (IF) and Western blotting are different techniques, and Bruno1 antibody was validated for specificity in IF but not in Western blots. Figure 5L and S5 E should include muscle samples from *Bru1M2*.

We have added a Western Blot panel in Figure S5 D including *bru1-IR*, *bru1^{M2}* and samples of different wild-type tissues including abdomen, ovaries, testis and IFM.

3. To quantify alternative splicing or percent spliced in (PSI), primers are typically designed in the exons flanking the alternative exons. A better primer design along with PSI calculation by RT-PCR will robustly validate alternative splicing changes in different genetic background (Fig 6U and S6 U).

We do not yet have RNA-Seq data from these *Rbfox1* knockdown samples to facilitate calculation of transcriptome-wide PSI values; thus, we rely on the results from our RT-PCR experiments. Our primers used to detect alternative splice events are indeed located within flanking exons or as close to the alternative exons as possible based on sequence design limitations (see schemes in Figure 6 and Figure S6). Many of the events we are detecting are complex, and not a simple "included" or "excluded" determination, and are therefore not amenable to RT-qPCR. To increase the robustness of our validation, we now provide RT-PCR gel-based quantification of exon use for the events we tested in *Zasp52*, *Zasp66*, *Zasp67*, *wupA* and *Mhc* (Figure 6 U-W and Figure S6 T-U).

Reviewer #1 (Significance (Required)):

Describe the nature and significance of the advance (e.g. conceptual, technical, clinical) for the field.

Understanding how muscle fiber type splicing and gene expression is regulated will conceptually move the field forward. How transcriptional and posttranscriptional programs coordinate to specify muscle fiber type gene expression is still lacking.

Place the work in the context of the existing literature (provide references, where appropriate).

Multiple RNA binding proteins and splicing factors have been shown to affect muscle function along with hundreds of gene expression and splicing changes in a complex fashion. Linking phenotypes with gene expression changes is still challenging as RNA binding proteins or RBPs are multifunctional and affect the function of other regulators that are important for muscle biology.

We thank the reviewer for recognizing the conceptual advance our findings represent, as well as the complexity in the regulatory network we are seeking to understand. A detailed understanding of the coordination of transcriptional and posttranscriptional programs is enabled by our work and will be the subject of future investigation.

State what audience might be interested in and influenced by the reported findings.

Fly genetics, alternative splicing regulation, muscle specification and function.

Define your field of expertise with a few keywords to help the authors contextualize your point of view. Indicate if there are any parts of the paper that you do not have sufficient expertise to evaluate.

Regulation and function of alternative splicing in muscle. I do not have a thorough knowledge of Drosophila genetics.

Response to Reviewer 2

Reviewer #2 (Evidence, reproducibility and clarity (Required)):

****Summary****

This paper reports analysis of the function of RbFox1, an RNA-binding protein, best known for roles in the regulation of alternative splicing. It uses Drosophila as its in vivo model system, one that is highly suited to the analysis in vivo of complex biological events. In general, the authors present a very thorough approach with an impressive range of molecular analysis, genetic experiments and phenotypic assays.

We thank the reviewer for recognizing the suitability of our model organism as well as the time investment and diversity of experiments that were performed in this work. We have added and revised multiple experiments during this revision, which has greatly improved the manuscript.

The authors report that Rbfox1 is expressed in all Drosophila muscle types, and regulated in both a temporal and muscle type specific manner. Using inhibitory RNA to knock down gene function, they show that Rbfox1 is required in muscle for both viability and pupal eclosion, and contributes to both muscle development and function. A Bioinformatic approach then identifies muscle genes with Rbfox1-binding motifs. They show Rbfox1 regulates expression of both muscle structural proteins and the splicing factor Bruno1, interestingly preferentially targeting the Bruno1-RB isoform. They report functional interaction between Rbfox1 and Bruno1 and that this is expression level-dependent. Lastly, they report that Rbfox1 regulates transcription factors that control muscle gene expression.

They conclude that the effect on muscle function of RbFox1 knock down is through mis-regulation of fibre type specific gene and splice isoform expression. Moreover, "Rbfox1 functions in a fibre-type and level-dependent manner to modulate both fibrillar and tubular muscle development". They propose that it does this by "binding to 5'-UTR and 3'-UTR regions to regulate transcript levels and binding to intronic regions to promote or inhibit alternative splice events." They also suggest that Rbfox1 acts "also through hierarchical regulation of the fibre diversity pathway." They provide further evidence to the field that Rbfox1's role in muscle development is conserved.

****MAJOR COMMENTS****

Are key conclusions convincing?

In terms of presentation, I suggest ensuring a clear demarcation throughout of the evidence behind the main conclusions. This can get somewhat lost as a great deal of information is presented, including all the parallels with prior findings in other systems. I am not saying this is a major problem, just highlighting the importance of clarity. Conclusions to clearly evidence include:

Rbfox1 functions in a fibre-type manner to modulate both fibrillar and tubular muscle development (e.g. L664);

Rbfox1 functions in a level-dependent manner (e.g. L664);

Rbfox1 functions by binding to 5'-UTR and 3'-UTR regions to regulate transcript levels (e.g. L670);

Rbfox1 functions by binding to intronic regions to promote or inhibit alternative splice events" (e.g. L670);

"Bru1 can regulate Rbfox1 levels in Drosophila muscle, and likely in a level-dependent manner" (L488) - Clearly evidence the level effect;

"first evidence for negative regulation for fine tuning acquisition of muscle-type specific properties. Depending on its expression level, Rbfox1 can either promote or inhibit expression of" muscle regulators (L797).

Lastly, the controlled stoichiometry of muscle structural proteins is known to be important, but all mechanisms are not known, so again make the supporting evidence as clear as possible for the interesting point of a role for Rbfox1 in this (e.g. L787).

Using the above comments from the reviewer as a guide, we have rewritten the manuscript, including large portions of the discussion, introduction and results. We thank the reviewer for pointing out where we could more effectively communicate our results, support our conclusions and highlight the significance of our findings.

Should some claims be qualified as preliminary or removed?

P301 "complicated genetic recombination" - seems a bit weak to include. Either do it or don't include?

We have removed this statement from the text.

Also, see section below on "adequate replication of experiments"

Are additional expts essential? (if so realistic in terms of time and cost)

None essential in my view. It depends on the authors' goals, but for the most impact of the project then following up these suggestions are possible.

L369-372: mutate putative Rbfox1 binding site and ask does binding still occur or not. If it doesn't, then ask if this mutation affects the expression of the putative target gene.

L775-777 "Our data thus support findings that Rbfox1 modulates transcription, but introduce a novel method of regulation, via regulating transcription factor transcript stability." It would be good to demonstrate this.

We thank the reviewer for these suggestions, and agree they are indeed interesting experiments, but beyond the scope of this manuscript. We plan to pursue the detailed molecular and biochemical mechanisms of regulation in a future project including exploring Rbfox1 binding through use of reporters, identification of direct targets via CLIP and investigation of post-transcriptional regulation of translation or NMD.

Presented in such a way as to be reproduced

Yes

Are expts adequately replicated?

A main area I would address is the authors frequent use of "may", "tend", "trend". This is confusing the picture they present. What is statistically significant and what is not? Only the former can be used as evidence.

Examples include:

L170: "may display preferential exon use" - does it or doesn't it?

L272: "myofibrils tended to be thicker" - were they or weren't they?

L350 "wupA mRNA levels tend towards upregulation in Rbfox1-RNAi".

L353 "but tended towards upregulation (Fig. S4A)"

L466 "Correspondingly, we see a trend towards increased protein-level expression of Bru1-PA"

L474 "both Bru1-PA and Bru1-PB tend to increase"

L485 "Overexpression of Bru1 in TDT with Act79B-Gal4 also tends to reduce Rbfox1"

L595 "Rbfox1-IR27286 tended towards increased exd levels in IFM (Fig. 7A)"

L614 "and a trend towards increased use of Mef2-Ex20 "

Also, L487 "suggesting that Bru1 can also negatively regulate Rbfox1" - one cannot use a non-significant observation to suggest something.

We have modified the text to limit use of "may", "tend" and "trend", and have removed discussion of non-significant results. We thank the reviewer for the very helpful and detailed list of sentences to modify.

****MINOR COMMENTS****

Although individual samples are not significant, in aggregate there is a trend....

Specific exp issues that are easily addressable

L162: "dip in Rbfox1 expression levels around 50h APF". The Fig indicates as early as 30h. Is this significantly less than the 24h data point?

Comparisons in Figure 1G that are significant based on DESeq2 differential expression analysis with an adjusted p-value < 0.05 are marked with asterisks. Rbfox1 mRNA expression at 24 h APF is significantly less than at 16 h APF, but the mRNA expression at 30 h APF and 48 h APF is not significantly less than at 24 h APF. Between 48 h APF and 72 h APF there is a significant increase in Rbfox1 mRNA levels. This may reflect a decrease on the mRNA level by 30 h that is followed a bit later by a decrease in protein levels by 50 h APF, but we have not performed additional experiments to confirm the relationship between the mRNA and protein levels.

L427 "this staining was lost after Rbfox1 knockdown". This conflicts with Fig 5K which says no significant difference.

Again in L429 "Rbfox1 knockdown leads to a reduction of Bru1 protein levels in IFMs and TDT." Fig says no significant difference in TDT.

We thank the reviewer for pointing out this inconsistency. We have revised the text accordingly. Our Western Blot (Figure 5L, M) and RT-PCR (Figure 5N, O) do show changes of Bru1 protein and mRNA expression levels after knockdown of Rbfox1^{KK110518}.

Are prior studies referenced appropriately?

This m/s is an authoritative presentation of the field as a whole with a comprehensive, impressive reference list. However, a point related to this area is one of the main things I would consider tackling. This is to have more clarity in the demarcation of what this study has found that adds to prior knowledge. It is worthwhile in itself to demonstrate the many similarities with previous work in other systems, as part of establishing the Drosophila system with all its analytical advantages for in vivo molecular genetics as an excellent model for future study in this area of research. However, the impact/strength of this m/s would be enhanced by clarity in presenting what is new to the field in all organisms.

We thank the reviewer for this suggestion. We have rewritten large portions of the manuscript, including the introduction and discussion, to improve the clarity of our findings and their importance to the field.

Are the text and Figs clear and accurate?

TEXT

L156: more precise language than "in a pattern consistent with the myoblasts" - maybe a simple co-expression with a myoblast marker?

We have revised this phrasing in the text. Rbfox1 expression in myoblasts was previously reported by (Usha and Shashidhara, 2010).

L181: at first use define difference between RNAi and IR

We use IR as an abbreviation for RNAi. In particular, we are trying to distinguish the two hairpins obtained from stock centers (27286 and KK110518) from the third, homemade RNAi hairpin, originally named UAS-*dA2BP1^{RNAi}*, that was generated by Usha and Shashidhara (Usha and Shashidhara, 2010). We have better defined this in the text and methods.

L205: maybe clearly explain the link between eclosion and tubular muscle??

We have added a sentence explaining the link between eclosion and tubular muscle (see Line 331).

L231: "Sarcomeres were not significantly shorter at 90h APF with the stronger Mef2-Gal4" - not clear why this is the case when the less strong knockdown conditions have shorter sarcomeres.

We have modified the text as well as the figure labeling to clarify that the other samples were tested in 1 d adult, while the KK110518 hairpin was tested at 90 h APF. This likely indicates that the short sarcomeres observed in 1 d adults reflect hypercontraction, which in IFM is classically first apparent after eclosion when the flies actively try to use the flight muscles. The difference in timing is due to pupal lethality of the KK110518 hairpin line, so we could not evaluate adult flies.

L234: "classic hypercontraction mutants in IFMs display a similar phenotype" - presumably not similar to the not significantly shorter sarcomeres of the previous sentence.

We have modified the text to clarify this statement. The change in sarcomere length from 90 h APF to 1 d adult is actually the relevant observation, as this reflects the progressive shortening of sarcomeres observed in classic hypercontraction mutants.

L244: "90h", should be "90h APF"?

Yes, we have modified the text.

L273: "Myofibrils in Act88F-Gal4 mediated knockdown only showed mild defects (Fig. 3 G, H, Fig. S2 C, D) despite adult flies being flight impaired". This seems worthy of discussion - the functional defect is not due to overt structure change?

In our own experience as well as observations included in a genome-wide RNAi screen in muscle (Schnorrer et al., 2010), there are a rather large number of knockdown conditions where few if any structural defects are observed at the level of light microscopy, but flies are completely flightless. We interpret this to reflect the narrow tuning of IFM function, where slight alterations in calcium regulation or sarcomere gene isoform expression result in dysfunction and a lack of flight. Ultrastructural evaluation might reveal defects in these cases, but the defect could also be with the dynamics of tropomyosin complex function, calcium regulation, mitochondrial function or even neuro-muscular junction structure. We have added a sentence to the text to discuss and clarify the Act88F result.

L281 "also known as Zebra bodies" - helpful to indicate these on the Fig, they are not.

We have added arrows to the figure to mark the Zebra bodies, and updated the figure legend.

L282: "we were unable to attempt a rescue of these defects" - I may have missed something, but what about rescue undertaken of the defects on previous pages?

This is the first point in the text where we introduced overexpression of *Rbfox1*, as preceding experiments where knockdown or using a GFP-tagged protein trap line at the endogenous locus. We have revised the sentence to focus on the overexpression phenotype with UH3-Gal4.

L283: "Over-expression of Rbfox1 from 40h APF" - this is the first over-expression experiment, so introduce why done now (and perhaps not earlier), and also explain the use of a different Gal4 driver.

We have reworded this section of the text. The UH3-Gal4 driver is restricted to expressing in IFM from 40h APF, so is first expressed after myofibrils have been generated and selectively in IFM. This avoids lethality observed from pan-muscle expression with *Mef2-Gal4* (presumably due to severe defects in tubular muscles), and also allows us to image IFM tissue from adult flies. Later experiments with *Mef2-Gal4* were performed with a later temperature shift to avoid this early lethality.

L290 "Interestingly, both Rbfox1 knockdown and Rbfox1 over-expression produce similar hypercontraction defects" - this could be interesting, worthy of discussion/explanation.

The most logical explanation is that Rbfox1 regulates the balance in fiber-type specific isoform expression. Loss of Rbfox1 would cause a shift in the relative ratio of the isoforms of structural genes, and overexpression of Rbfox1 would likely cause a similar shift in the opposite direction. This is supported by our RT-PCR panel, where we see co-regulation of different events with Bru1, and we see fiber-type specific difference in regulation of alternative splicing (Figure 8). Overexpression of Rbfox1 would be expected to make IFM look more like TDT, which would result in an isoform imbalance and lead to the observed hypercontraction phenotype. Interestingly, loss and overexpression of Bru1 also result in the same hypercontraction phenotype, similar to what we observe with Rbfox1. We have added a paragraph in the discussion about level-dependent regulation, to address this reviewer comment.

P305: Bioinformatic analysis. It is not clear what is taken as a potentially interesting result. On average a specific 5 base motif is found every 1000bps - so what is being looked for? How many sites in what length or position? A range of examples are described in the next pages of the m/s. For example: L337 "Bruno1.... contains 42 intronic and 2 5'-UTR Rbfox1 binding motifs" and L591 "exd contains three Rbfox1 binding sites,"

We have redone the bioinformatic analysis completely, relying on data from oRNAmotif and the *in-vitro* determined PWM. We have also rewritten all portions of the text related to this analysis and no longer focus on the number of observed motifs in a given gene. As we unfortunately do not have RNA CLIP data, we do not know genome-wide which motifs are bound in muscle. Clustering of motifs may reflect binding, but a single, strong motif can also be bound, as we demonstrate via RIP of the *wupA* transcript. Thus, we identified interesting targets to test based on 1) a previously described role in the literature in myofibril assembly or contractility and 2) the presence of any Rbfox1 motif in that gene. A more elegant selection method of direct and indirect target exons will be designed for a future manuscript after integrating CLIP and mRNA-Seq data that have not yet been collected.

L315: "many of these genes have binding or catalytic activity". "catalytic activity" seems very vague.

For the original supplemental figure panel, we relied on Panther high-level ontology terms, which can unfortunately be rather vague, ie "catalytic activity" or "binding activity". We have redone this analysis and rely rather on GO terms in the biological process and molecular function categories (Figure S3 B).

L317 "When we look in previously annotated gene lists" - be more specific. What are they?

This section of the text has been rewritten, and the "previously annotated gene lists" are described in greater detail in the Methods.

L327 "may also affect the neuro-muscular junction" - maybe better left for the Discussion?

We have removed this sentence from the Results.

L333 "extradenticle (exd) and Myocyte enhancer factor 2 (Mef2) contain 3 and 7 Rbfox1 motifs," Discuss the number and position of multiple motifs found in known targets?

We have removed the discussion of the number of binding sites for different target genes, instead incorporating this information graphically in Figure S3 C. It is not clear that the number of binding sites per gene has any influence on whether it is regulated in *Rbfox1* knockdown. Thus, we have de-emphasized discussion of the number of binding sites throughout the text.

L350 "wupA mRNA levels " - clearer to stick to using TroponinI or WupA?

We have updated instances throughout the text to consistently refer to the protein as Troponin-I (TnI) and the gene or mRNA as *wupA*.

L376 "To check whether Rbfox1 regulates some target mRNAs such as wupA....." The suggestion here is more of a further indication than a "check".

We have reworded this section of the results to make the link between post-transcriptional regulation and our mass spectrometry results more salient.

L544 "In IFMs, knockdown of Rbfox1 and loss of Bru1 results in...." clarify if this is the two genes separately or the two genes together?

We have rewritten this entire section and present an expanded list of tested alternative events. We have taken care in this revision to clearly denote if the genotype is *Rbfox1-IR* or *bru1^{M2}* or a double knockdown background.

L580 "Our bioinformatic analysis identified Rbfox1 binding motifs in more than 40% of transcription factors genes" - is this all TFs or just "muscle" TF genes?

We have redone this analysis and changed this sentence in the text.

L598, what would be the mechanism of some decrease in Rbfox1 increasing mRNA levels and more of a decrease resulting in a decrease of the mRNA? The authors say "the nature of this regulation requires further investigation".

We have added more data to this section of the manuscript and repeated several of these experiments. After adding more biological replicates and additional data points, we have more consistent results that also demonstrate the variability in *bru1* expression levels after *Rbfox1* knockdown. Overall levels of *bru1* assayed with a primer set in exons 14 and 17 now consistently show an increase in *bru1* expression after *Rbfox1* knockdown between all three hairpins (*Rbfox1-RNAi*, *Rbfox1-IR^{KK110518}* and *Dcr2, Rbfox1-IR²⁷²⁸⁶*) (Figure 5 N).

The relationship between expression level of Rbfox1 and expression level of *bru1* and Bru1 protein isoforms is more complex. We now report a novel splice event in the annotated isoform *bru1-RB* that skips exon 7, resulting in a frame shift and generation of a protein that lacks all RRM domains, which we call *bru1-RB^{short}* (Figure S5). This short isoform is preferentially used in TDT, while the long isoform encoding the full-length protein is preferentially used in IFM (Figure 5 P). Presumably, this provides a mechanism, in addition to the use of different promoters, for muscle cells to regulate expression levels of different Bru1 isoforms. Knockdown of Rbfox1 in IFM results in a significant increase in the use of the long mRNA isoform, but paradoxically a decrease in the corresponding protein isoform (Figure 5, S5). We interpret this to mean that Rbfox1 regulates alternative splicing of Bru1, and likely independently a translational/post-translational mechanism regulates the expression level of Bru1-RB. This in theory could be mediated by interaction with translational machinery, post-translational modification, increased P-granule association, etc., and given the depth and breadth of experiments (as well as the multitude of isoform-specific expression reagents) required to isolate the responsible pathway, we deem it beyond the scope of this manuscript to biochemically demonstrate this specific regulatory mechanism.

L609 "The short 5'-UTR encoded by Mef2-Ex17". Ensure all abbreviations are defined. What does "Ex" mean here? Not straightforward to relate to the diagram in the Supplemental material that indicates the Mef2 gene has many fewer than 17 exons. In Fig7 legend too.

We have changed "Ex" to "exon" in the text. We apologize for the confusion. We have also added a diagram to Figure 7 E of the 5'-UTR region of Mef2, and a complete diagram of the locus in Figure S3 C. Based on the current annotation, *Mef2* exons are numbered 1 to 21, corresponding to at least 16 distinct regions of the genome (18 if you include the variable 3'-UTR lengths). Exons sometimes will have more than one number in the annotation if a particular splice event causes a shift in the ORF, or if alternative splice sites or poly-adenylation sites are used. *Mef2* is also on the minus strand, so as exons are numbered based on the genome scaffold, the exon numbering goes in reverse (ie exon 1 is the 3'-UTR).

We strongly believe in following the numbering provided in the annotation, to increase reproducibility and transparency in working with complex gene loci for many different genes. Another researcher can go to Flybase, look-up the exon number from a given gene from a specific annotation, and get the exact location and sequence of the exons we name. It is incredibly challenging and time intensive to go through older papers and figure out which exon or splice event corresponds to those in the current annotation, and we aim to alleviate this difficulty (we illustrate this in Figure 4 A for the *wupA* locus, where we verified exon numbers in annotation FB2021_05 by BLASTing each individual sequence and primer provided in (Barbas et al., 1993).

L617 "Levels of Mef2 are known to affect muscle morphogenesis but not production of different isoforms" - clarify what is meant here by "different isoforms".

We have revised this section of the text. This statement was meant to reflect that Mef2 affects muscle morphogenesis through regulation of transcription levels, but not at the level of alternative splicing.

L638 "Salm levels were significantly increased in IFM from Rbfox1-RNAi animals, but significantly decreased in IFMs from flies with Dcr2 enhanced Rbfox1-IR27286 or Rbfox1-IRKK110518". This is worth discussion or further analysis. Normally would expect an allelic series, with an effect becoming more apparent with increased loss-of-function.

Dcr2, Rbfox1-IR27286 and *Rbfox1-IRKK110518* produce a stronger knockdown than *Rbfox1-RNAi*, and indeed produce significantly decreased levels of *salm*, thus following the allelic series. We repeated this experiment, but obtained the same results.

L641 "This suggests that Rbfox1 can regulated Salm". How, if there are no Rbfox1 binding sites? Deserves further analysis?

Our new bioinformatic analysis suggests a possible answer, in that it identified possible Rbfox1 motifs in a *salm* exon and a site in an intron. Previously, we had focused on introns and UTR regions. In addition, using the PWM we now recover Rbfox1 binding sites of the canonical TGCATGA as well as AGCATGA sites. The intron site in *salm* is an AGCATGA site. Further experiments will be required to determine if Rbfox1 directly binds to *salm* mRNA, if it interacts with the transcriptional machinery to regulate *salm* expression, or if this regulation occurs through yet a different mechanism, and are beyond the scope of this manuscript.

L674: "We found the valence of several regulatory interactions..." I'm not sure the meaning of "valence" here and elsewhere will be readily understood.

Thank you for pointing this out. We have used a different phrasing throughout the text.

FIGURES

Fig 1 it is difficult to see the green in A-F. Can this be improved? It is clearer in I-L.

We have replaced the images with better examples and increased the levels to make the green channel better visible.

Fig 2 legend (others too), say what the clusters of small black ellipses in P and Q are.

Thank you for pointing out this oversight. All boxplots are plotted with Tukey whiskers, such that they are drawn to the 25th and 75th percentile plus 1.5 the interquartile range. Dots represent outlying datapoints outside of this range. We have added statements in the relevant figure legends, as well as a more detailed explanation in the Methods.

Fig 3 it is not easy to see a shorter sarcomere in D, as the arrow partially obscures what is being indicated. Also, the data in G indicates that sarcomeres are not shorter in Mef2 GAL4 > KK110518, although the legend says this is shown in D. We have rephrased the statement in the legend. The arrows are pointing to frayed or torn myofibrils.

Fig 5 legend "-J). Bru1 signal is reduced with Rbfox1-IRKK110518 (C, F, I)". Clarify that this is only in IFM. It is not significant in TDT or Abd-M.

Done.

Fig 7 legend "quantification of the fold change in *exd* transcript levels" - only KK110518 in IFM is significant.

This panel was moved to Figure S7. The relevant regions of the text and figure legend were modified to reflect that only Rbfox1-IR^{KK110518} results in a significant change in *exd* levels.

C - "indicates Rbfox1 binds to Mef2 mRNA" - it is not easy to see the band.

We replaced the image and adjusted the levels to make the band more visible.

D - what do the different lanes on the gel below the histogram in D correspond to?

We adjusted the labeling on the figure panel. The gel is a representative image of RT-PCR results that are quantified above in the histogram.

Suggestions that would help the presentation of their data and conclusion

There is a lot of good, thorough work here, but overall there is the impression that some of the presentation/writing could be improved (also see the above lists on clarity and accuracy).

I admire the authors for their comprehensive presentation of what has already been found out in this field. As the authors summarise, a lot is already known in many other species, so (as also indicated above) it is crucial to emphasise what new is found in this work that advances overall knowledge in this field. This can be obscured in many places where they say because of what was found in vertebrate systems we looked in *Drosophila*. These include:

L417: "This led us to investigate if Rbfox1 might regulate Bru1 in *Drosophila*."

L452: "and we were curious if these interactions are evolutionarily conserved in flies."

L528 "Thus, we next checked if Rbfox1 and Bru1 co-regulate alternative splicing in *Drosophila* muscle."

L677 "Moreover, as in vertebrates, Rbfox1 and Bru1 exhibit cross-regulatory interactions"

L683 "Rbfox1 function in muscle development is evolutionarily conserved"

L697 "Here we extend those findings and show that as in vertebrates....."

L702 "our observations are consistent with observations in vertebrates"

L707 "Studies from both vertebrates and *C. elegans* suggest that *Rbfox1* modulates developmental isoform switches."

L746 "We see evidence for similar regulatory interactions between *Rbfox1* and the *CELF1/2* homolog *Bru1* in our data from *Drosophila*."

We thank the reviewer for this honest and helpful assessment of the manuscript. Upon rereading the original text and with the guidance of the list of sentences above, we agreed with the reviewer and we have rewritten large segments of the manuscript. In particular in the introduction and discussion, we now better emphasize what is new in our findings and how they advance overall knowledge in this field.

L185 paragraph. The knockdown series is important for the study. A lot is presented in this paragraph, especially for a non-specialist and it could be easier to follow. Perhaps present the four genetic conditions in the order of the severity of their phenotype on viability. Also, clearly state what each *Gal4* driver is used for. What is the nature of the RNAi/IR lines such that *Dcr2* could enhance their action? Also comment on off targets - are any predicted?

We have rewritten this paragraph as the reviewer requested. The hairpins are ordered by decreasing phenotypic severity, and we have more clearly described each *Gal4* driver as well as *Dicer2*. This information is also available in the Methods, along with the off targets for the hairpins. KK110518 has one predicted off-target *ichor*, but this gene is not expressed in IFM, TDT or leg based on mRNA-Seq data. 27286 has no predicted off-targets.

L227: "In severe examples". Be as clear as possible. Are the "severe examples" using the stronger RNAi line or are they the most severe examples with a single line? I'd suggest including the result in the main Fig rather than in the Supplemental. However, as I read more of the m/s I realise there is a great deal of important information in the Supplemental Figs, and so the case is not much stronger for this example than many others. The balance of what is included where could be looked at, because it is not straightforward for the reader to read the paper and quickly flick between the main and supplemental Figs. Later in the m/s is a substantial section that starts L450 (finishes L489) and which only refers to Supplemental Figs. L503 is another area where it is necessary, and difficult, for the reader to move between main Figs and supplemental Figs.

We have reorganized the figure panels in several figures, notably Figures 4, 5, 6, 7 and 8 and the corresponding supplementary figures, including moving panels from the supplemental figures to the main figures and generating more comprehensive quantification panels. In the specific case referenced here for Fig. S1 P and Q, we chose to keep the most representative images of the phenotype in the main figure (Fig. 2 I, N), and have reworded the text to reflect that the most severe phenotypic instances are in the supplement. As we do not have CLIP data, we chose to keep the bioinformatics analysis in the supplement and have shortened the paragraph in the results devoted to Figure S3. We hope our reorganization and rewriting have better streamlined the text and figures.

L258: - perhaps a Table summarising this and other phenotype trends with the different RNA conditions might be helpful. It gets quite difficult to follow.

We have revised the text and several figure panels to make the phenotypic trends with the different RNAi conditions easier to follow.

Reviewer #2 (Significance (Required)):

The advance reported is mechanistic.

The authors already do a very good job of placing their work in the context of prior research (see comment in Section A).

Muscle biologists interested in its development and function will be interested in this work. More broadly, those intrigued by alternative splicing will be interested. Despite its very widespread occurrence, much about alternative splicing is still poorly understood in terms of regulation and significance. This is especially the case in vivo, and this paper uses an excellent in vivo model system (*Drosophila*) for the genetic and mechanistic analysis of complex biological problems. My field of expertise: cell differentiation, gene expression, muscle development, *Drosophila*.

Response to Reviewer 3

Reviewer #3 (Evidence, reproducibility and clarity (Required)):

****SUMMARY****

This manuscript characterizes the role of splicing factor Rbfox1 in Drosophila muscle and explores its ability to modulate expression of genes important for fibrillar and tubular muscle development. The authors hypothesize that Rbfox1 binds directly to 5'-UTR and 3'-UTR regions to regulate transcript levels, and to intronic regions to promote or inhibit alternative splicing events. Because some of the regulated genes encode transcriptional activators and other splicing factors such as Bru1, the effects of Rbfox1 may encompass a complex regulatory network that fine-tunes transcript levels and alternative splicing patterns that shape developing muscle. Most likely the authors' hypothesis is correct that Rbfox1 is critical for muscle development in Drosophila, but overall the interesting ideas presented here are too often based only on correlations without further experimental validation.

We respectfully disagree with the reviewer that our hypothesis that Rbfox1 is critical for muscle development in *Drosophila* is based only on correlation without further experimental validation. In this manuscript we extensively characterize the knockdown phenotype of 3 RNAi hairpins against Rbfox1 as well as a GFP-tagged Rbfox1 protein in both fibrillar flight muscle and tubular abdominal and jump muscle. All hairpins produce similar phenotypes with defects in myofiber and myofibril structure and result in behavioral defects in climbing, flight and jumping, confirming this phenotype is due to loss of Rbfox1 and not a random off-target gene. We also convincingly demonstrate that Rbfox1 regulates Bru1, another splicing factor known to be critical for fibrillar specific splice events in IFM. Moreover, Rbfox1 and Bru1 genetically interact selectively in IFM and our RT-PCR data for 12 select structural genes reveals fiber-type specific alternative splicing defects regulated by Rbfox1 selectively, by Bru1 selectively, or by both Rbfox1 and Bru1. Thus, we conclude that Rbfox1 is indeed critical for muscle development, and this is the first report to demonstrate this requirement in *Drosophila*.

****MAJOR COMMENTS****

The hypothesis that Rbfox1 plays an important role in regulating muscle development is based on previous studies in other species and supported by much new data in this manuscript. Initial bioinformatic analysis showed that many Drosophila genes, including 20% of all RNA-binding proteins, 40% of transcription factors, etc. have the motifs in introns or UTR regions. However, I think a deeper analysis is required. Any hexamer might be present about once every 4kb, and we do not expect all UGCAUG motifs are necessarily functional, so one might ask whether the association of Rbfox motifs with muscle development genes is statistically significant? Are the motifs conserved in other Drosophila species, which might support a functional role in muscle? Are the intronic motifs located as expected for regulatory effects, that is, proximal to alternative exons that exhibit changes in splicing when Rbfox1 expression is decreased or increased?

We appreciate the point of the reviewer that it would be ideal to distinguish genome-wide motifs that are actually bound directly by Rbfox1 from those that are unused, but our behavioral and phenotypic characterization of the knockdown phenotype in this manuscript is also valid without this data. The most effective approach to identify direct targets is to perform cross-linking immunoprecipitation, or CLIP, but we unfortunately do not have CLIP data from *Drosophila* muscle and it is beyond the scope of the current study to generate this data. It is not trivial to obtain the amount of material necessary to identify tissue-specific binding sites, as we would also likely expect differences in targeting specificity between tubular and fibrillar muscle. Genome-wide analysis of the evolutionary conservation of binding site motifs is also not trivial and is beyond the scope of this paper.

Despite these limitations and to address the reviewer's comment, we have done the following:

1. We have completely redone our bioinformatic analysis using transcriptome data from the oRNAment database (Benoit Bouvrette et al., 2020), as well as searching genome-wide for instances of the *in vitro* determined PWM using PWMScan, to capture possible sites in introns (Figure S3). The oRNAment database was shown to reasonably predict peaks identified in eCLIP from human cell lines, which we assume would translate to a similar predictive capacity in the *Drosophila* transcriptome.
2. We have calculated the expected distribution of Rbfox1 sites in a random gene list for Figure S3, and indeed the number of Rbfox1 sites in sarcomere genes is significantly enriched.
3. We have looked more carefully at the distribution of Rbfox1 and Bru1 motifs in the transcriptome (in the oRNAment data), and find not only that these motifs frequently occur in the same muscle phenotype genes, but also that they are closer together than is expected by chance (Fig. S4 J).
4. We marked the location of Rbfox1 and Bru1 motifs in the vicinity of select alternative splice events we tested via RT-PCR on the provided summary diagrams (Fig. 6, Fig. S6).

5. We have tested additional alternative splice events in total from 12 structural genes, and of the 9 events misregulated after Rbfox1 or Bru1 knockdown, all but 1 are flanked by Rbfox1 or Bru1 binding motifs. This indicates that the motifs are indeed located as expected for a regulatory effect.

Is it possible to knock out an Rbfox motif and show that splicing of the alternative exon is altered, or regulation of transcript levels is abrogated?^{F17}_{SEP}

The construction and mutation of reporter constructs is possible, but would take longer than the recommended revision time-frame, in particular to generate reporters that can be evaluated *in vivo*. We intend to address the biochemical mechanism(s) of Rbfox1 regulation with future experiments in a separate manuscript.

Also, what was the background set of genes used for the GO enrichment analysis? Genes expressed in muscle or all genes?

The background set of genes for GO enrichment (now Figure S3 B) was all annotated genes for the “all genes” label and all muscle phenotype genes for the “Muscle phenotype” label.

2. The data on cross regulation between Rbfox1 and Bru1 are confusing and inconsistent, since mild knockdown and stronger knockdown of Rbfox1 seem to have different effects on Bru1 expression. New data suggest that Rbfox1 can positively regulate Bru1 protein levels (Fig.5), but this seems inconsistent with the lab's earlier studies indicating opposite temporal mRNA expression profiles for Rbfox1 and Bru1 across IFM development.

We apologize for the confusion, but the relationship between Rbfox1 and bru1 levels across IFM development has not been published previously. We previously generated that mRNA-Seq data, but presented here (now in Figure 5Q) is a new analysis of that data, specifically focused on *Rbfox1* and *bru1* expression. We have corrected the phrasing in the text.

To address this comment, along with points raised above by Reviewer 2, we have revised this part of the manuscript, added more data to this section of the manuscript and repeated several of these experiments. After adding more biological replicates and additional data points, we have more consistent results that also demonstrate the variability in *bru1* expression levels after *Rbfox1* knockdown. Overall levels of *bru1* assayed with a primer set in exons 14 and 17 now consistently show an increase in *bru1* expression after *Rbfox1* knockdown between all three hairpins (*Rbfox1-RNAi*, *Rbfox1-IR^{KK110518}* and *Dcr2, Rbfox1-IR²⁷²⁸⁶*) (Figure 5 N). This is consistent with our observations of inversely correlated mRNA levels during IFM development, as when Rbfox1 levels decrease, *bru1* transcripts increase.

We agree with the reviewer that the relationship between the expression level of Rbfox1 and expression level of *bru1* mRNA and Bru1 protein isoforms is more complex. We now report a novel splice event in the annotated isoform *bru1-RB* that skips exon 7, resulting in a frame shift and generation of a protein that lacks all RRM domains, which we call *bru1-RB^{short}* (Figure S5). Unknowingly, we had previously used a primer set from exon 7 to exon 8 as “common”, which lead to some confusion. This short isoform is preferentially used in TDT, while the long isoform encoding the full-length protein is preferentially used in IFM (Figure 5 P). Presumably, this provides a mechanism, in addition to the use of different promoters, for muscle cells to regulate expression levels of different Bru1 isoforms. Knockdown of Rbfox1 in IFM results in a significant increase in the use of the long mRNA isoform, but paradoxically a decrease in the corresponding protein isoform (Figure 5, S5). We interpret this to mean that Rbfox1 regulates alternative splicing of Bru1, and likely independently a translational/post-translational mechanism regulates the expression level of Bru1-RB. This in theory could be mediated by interaction with translational machinery, post-translational modification, increased P-granule association, etc., and given the depth and breadth of experiments (as well as the multitude of isoform-specific expression reagents) required to isolate the responsible pathway, we deem it beyond the scope of this manuscript to biochemically demonstrate this specific regulatory mechanism.

Both Rbfox1 and Bru1 gene have many Rbfox motifs, but they are both large genes (>100kb) and would be expected to have many copies of all hexamers. How do we know whether any of them are functional?

We do not know if all of the Rbfox1 binding sites in the Bru1 and Rbfox1 loci are bound, but the CLIP data required to assess this is beyond the scope of this manuscript, as discussed above. We do show, however, that changes in the expression level of Rbfox1 affect the expression of Bru1 on both the mRNA transcript and protein level, and changes in the expression level of Bru1 also can affect the expression level of Rbfox1. The direct or indirect nature of this regulation remains to be fully elucidated, although we do provide RIP data showing we can detect *bru1* transcript bound to Rbfox1-GFP (Figure S4 I). We have modified the text to address this comment.

3. Figure S4, section I, J: if changes in Bru1-RB isoform expression are correlated with Rbfox1 knockdown, it seems

reasonable to test whether the *Bru1-RB* promoter can drive expression of GFP in an *Rbfox1*-dependent manner. But if I understand correctly, the assay as described on p. 19 uses the promoter region upstream of *Bru1-RA*. What is the logic for this experiment? It is not surprising that no effect was observed. The end result is that we have no idea whether *Rbfox1* directly regulates *bru1-RB*. Even if it does, *bru-Rb* appears to be a minor component of *Bru* expression in IFM.

Upon reevaluating this experiment and with respect to the reviewer's comment, we have removed it from the manuscript to avoid confusion. Our new data indicate a switch in use of the *bru1-RB^{long}* and *bru1-RB^{short}* isoforms (Figure 5 N-P), suggesting that *Rbfox1* regulation is on the level of splicing.

Further experiments will be necessary to refine the indirect versus direct regulatory effects of *Rbfox1* on *Bru1*, but our data do demonstrate that *Bru1* levels are regulated in *Rbfox1* knockdown conditions. We also provide a RIP experiment (Figure S4 I) showing that *Rbfox1*-GFP does directly bind *bru1* mRNA, but we did not determine if this was isoform-specific. Multiple additional experiments would be necessary to distinguish between regulation of alternative splicing, direct binding to regulate transcript translation or stability, or transcriptional regulation via regulation of Salm, or some combination of these possible mechanisms. The data presented here are important to the field as they are the first report of isoform-specific regulation of *Bru1* in muscle, even if we do not conclusively show if this regulation by *Rbfox1* is direct or indirect.

4. In the section "*Rbfox1* and *Bruno1* co-regulate alternative splice events in IFMs", the data show that splicing of several genes is altered by knockdown or over-expression of *Rbfox1* and *Bru1*. The interesting conclusion is for a complex regulatory dynamic where *Rbfox1* and *Bru1* co-regulate some alternative splice events and independently regulate other events in a muscle-type specific manner. However, if we are to conclude that these activities are due to direct binding of *Rbfox1* and *Bru1* to the adjacent introns, we need information about the location of flanking *Rbfox* and/or *Bru1* motifs. Do upstream or downstream binding sites correlate with enhancer or silencer activity, as reported in previous studies of these splicing factors in other species? For *wupA*, Figure S3 shows an intronic *Rbfox* site, but exon 4 is not labeled so the reader cannot correlate this information with the diagram in Figure 6U.

As mentioned above, we have marked the location of *Rbfox1* as well as *Bru1* binding motifs in the diagrams in Figure 6 and Figure S6. We have tested additional alternative splice events, and can now show events regulated only in the *Rbfox1* knockdown, only after *bru1* knockdown, or in double knockdown flies (Figure 8). 8 out of 9 events where we see clear changes in splicing are flanked by potential *Rbfox1* or *Bru1* motifs. Demonstration of direct binding and assay of genome-wide binding sites through CLIP studies is beyond the scope of this manuscript and will be pursued in the future.

5. The evidence that *Rbfox1* directly affects expression of transcription factor *Exd* seems to be based only a correlation between *Rbfox1* knockdown and decreased expression of *Exd*. The observation that binding of *Rbfox1* to the *Exd* 3'UTR in RIP experiments further weakens the case.

We agree with the reviewer and have moved the data related to *exd* to the supplement (Figure 7 and S7). We still mention *exd* in the text as it is significantly decreased after knockdown with *Rbfox1-IR^{KK110518}*, but we have removed it from larger claims of transcriptional regulation as well as from the summary in Figure 8. Also, just to note that although we failed to detect *Rbfox1*-GFP bound to *exd*, this experiment was performed with adult flies. Since *Exd* is functionally important early in pupal development during fate specification of the IFMs, it is possible we might detect binding to *exd* mRNA at a different developmental timepoint.

6. Similarly, there is a correlation of *Rbfox1* knockdown with expression of alternative 5'UTRs in the *Mef2* gene. However, the changes in UTR expression appear mostly not statistically significant. Do the authors have a model to explain what mechanism might allow *Rbfox* to regulate expression of alternative 5'UTRs, which would seem to be a transcriptional process?

Mef2 transcript levels are significantly increased after knockdown with *Rbfox1-RNAi* and decreased after overexpression of *Rbfox1*, and we can detect direct binding of *Rbfox1*-GFP to *Mef2* RNA via RIP. This establishes *Mef2* as a likely direct target of *Rbfox1* regulation, likely through the two *Rbfox1* motifs in the 3'-UTR (Figure S3 C). In addition to this regulation, we made an observation that has not been previously reported in the literature, that IFM expresses a particular isoform of *Mef2* that uses a short promoter encoded by Exon 17. We see both tissue-specific use of Exon 17 (Figure 7 F) as well as developmental regulation of Exon 17 use in IFM (Figure S7 C). Surprisingly, we saw that use of exon 17 in the *Mef2* promoter is altered in *Rbfox1* knockdown muscle. We now provide a quantification of this data, to show the change is statistically significant. We also provide a scheme of the *Mef2* locus and RT-PCR primers with exons 17, 20 and 21 labelled (Figure 7 E). We have also rewritten this section of the text to increase the impact and clarity of our finding.

7. For *Salm*, there apparently are no *Rbfox* motifs in the gene, and there are statistically significant but apparently inconsistent changes in *Salm* expression when it is knocked down in IFM by *Rbfox1-RNAi* (*Salm* increases) vs knockdown by *Rbfox1-IR27286* or *Rbfox1-IRKK110518* (*Salm* decreases). These are potentially interesting observations but more data would be needed to make stronger conclusions. How would regulation occur in the absence of *Rbfox* motifs?

The best explanation we can provide for why *salm* expression is increased with the weak hypomorph *Rbfox1-RNAi* condition, but decreased with the stronger hypomorph *Rbfox1-IR^{KK110518}* or *Dcr2*, *Rbfox1-IR²⁷²⁸⁶* conditions is that *salm* regulation is sensitive to *Rbfox1* expression or activity level. We now discuss this in a new section of the discussion. We further attempted several experiments to address this question, including obtaining an endogenously tagged *Salm-GFP* line, as well as a UAS-*Salm* line (kindly provided by F. Schnorrer). Disappointingly, there is no GFP expressed in the *Salm-GFP* line, either live, by immunostaining or in Western Blot of multiple developmental stages, indicating that the line has fallen apart and we have not yet redone the CRISPR targeting to generate a new line. The UAS-*Salm* construct works (too well), in that overexpression with *Mef2-Gal4* results in early lethality and we have not yet managed to optimize the experiment and obtain enough pupal muscle where we can evaluate the effect on *Bru1* or *Rbfox1* levels.

Our new bioinformatic analysis further revealed possible *Rbfox1* motifs in a *salm* exon and a site in an intron. Previously, we had focused on introns and UTR regions. Now, using the *in vitro* determined PWM, we can recover *Rbfox1* binding sites of the canonical TGCATGA as well as AGCATGA sites. The intron site in *salm* is an AGCATGA site. Further experiments will be required to determine if *Rbfox1* directly binds to *salm* pre-mRNA, if it interacts with the transcriptional machinery to regulate *salm* expression, or if this regulation occurs through yet a different mechanism. We feel the many required experiments are beyond the scope of the current manuscript. Our data provides an experimental basis for future studies on this topic.

****MINOR COMMENTS****

1. In several figures there is a misalignment of the transcriptional driver information with the phenotype data in the bar graphs above. Please correct the alignments to make interpretation easier.

We have revised the layout of labels for many plots throughout the manuscript to avoid a category label associated with a genotype label at a 45-degree angle, and to make interpretation easier.

2. On p. 14 Brudno *et al.* is cited as ref for *Fox* motifs near muscle exons, but this paper only focused on brain-specific exons.

In addition to brain-specific exons, Brudno *et al.* also analyzed a set of muscle-specific exons, and thus this is the appropriate reference. For instance, from the Brudno paper, "As an additional control in some experiments we analyzed a smaller sample of muscle-specific alternative exons that were collected exactly as described above for the brain-specific exons" and "UGCAUG was also found at a high frequency downstream of a smaller group of muscle-specific exons." Further details of the muscle-specific exon analysis can be found in (Brudno *et al.*, 2001).

3. For *Mef2*, why do exons described as 5'UTR have numbers 17, 20, and 21? One would normally expect these to be exon 1, 2 or 1A, 1B, etc.

We rely on the Flybase annotation and numbering system to refer to exons. Per Flybase, all exons are labeled in the 5' to 3' direction of the sequenced genome, even for genes, such as *Mef2* or *wupA*, that are encoded on the reverse strand. We strongly believe in following the numbering provided in the annotation, to increase reproducibility and transparency in working with complex gene loci for many different genes. Another researcher can go to Flybase, look-up the exon number from a given gene from a specific annotation, and get the exact location and sequence of the exons we name. It is incredibly challenging and time intensive to go through older papers and figure out which exon or splice event corresponds to those in the current annotation. We illustrate this in Figure 4 A for the *wupA* locus, where we verified exon numbers in annotation FB2021_05 by BLASTing each individual sequence and primer provided in (Barbas *et al.*, 1993). The *Mhc* locus is even more complex, in particular regarding alternative 3'-UTR regions and historic versus current exon designations (Nikonova *et al.*, 2020). For clarity and reproducibility, we therefore rely on the current Flybase designations.

4. Fig 8: "regulation of regulators" seems to imply the *Rbfox1* is impacting transcription?? Is there precedence for this type of regulation by *Rbfox1*?

Yes, indeed, there is precedence for *Rbfox1* impacting transcription, as we presented in the Discussion. *Rbfox2* is

reported to interact with the Polycomb repressive complex 2 to regulate gene transcription in mouse (Wei et al., 2016) and in flies Rbfox1 interacts with transcription factors including Cubitus interruptus and Suppressor of Hairless to regulate transcription downstream of Hedgehog and Notch signaling (Shukla et al., 2017; Usha and Shashidhara, 2010). In addition, Rbfox1 regulates splicing of Mef2A and Rbfox1 and Rbfox1 cooperatively regulate splicing of Mef2D during C2C12 cell differentiation (Gao et al., 2016). Our results provide a further piece of evidence implicating Rbfox1 either directly or indirectly in transcriptional regulation as well as regulation of alternative splicing.

Reviewer #3 (Significance (Required)):

****SIGNIFICANCE****

These studies of a major tissue-specific RNA binding protein, Rbfox1, are definitely important for our understanding of functional differences between muscle subtypes, and between muscle and nonmuscle tissues. The broad outlines of Rbfox1 alternative splicing regulation are known, but there is very little specific detail about the important targets in muscle subtypes that might help explain functional differences between subtypes. If more experimental validation can be obtained for regulation of transcript levels by binding 3'UTRs, this would also represent new information.

We thank the reviewer for recognizing the significance of our work and our detailed analysis of Rbfox1 phenotypes in different muscle fiber-types. Experimental validation of 3'-UTR binding will be a significant time investment in terms of building and testing in-vivo reporter constructs, assaying NMD and translation effects and performing the CLIP studies necessary for identification of directly-bound 3'-UTR regions, extending beyond the scope of this manuscript and the time allotted for revision. The data we present here represent an important advance in our understanding how Rbfox1 contributes to muscle-type specific differentiation, and form the basis for future experiments to explore the molecular and biochemical mechanisms underlying this regulation.

I am reviewing based on my experience studying alternative splicing in vertebrate systems, with an emphasis on Rbfox genes. Therefore I am unable to evaluate the functional data on different subtypes of muscle in Drosophila.

Reviewer Response References

Barbas, J. A., Galceran, J., Torroja, L., Prado, A. and Ferrús, A. (1993). Abnormal muscle development in the heldup3 mutant of *Drosophila melanogaster* is caused by a splicing defect affecting selected troponin I isoforms. *Mol Cell Biol* **13**, 1433–1439.

Benoit Bouvrette, L. P., Bovaird, S., Blanchette, M. and Lécuyer, E. (2020). oRNAment: a database of putative RNA binding protein target sites in the transcriptomes of model species. *Nucleic Acids Research* **48**, D166–D173.

Brudno, M., Gelfand, M. S., Spengler, S., Zorn, M., Dubchak, I. and Conboy, J. G. (2001). Computational analysis of candidate intron regulatory elements for tissue-specific alternative pre-mRNA splicing. *Nucleic Acids Res* **29**, 2338–2348.

Damianov, A., Ying, Y., Lin, C.-H., Lee, J.-A., Tran, D., Vashisht, A. A., Bahrami-Samani, E., Xing, Y., Martin, K. C., Wohlschlegel, J. A., et al. (2016). Rbfox Proteins Regulate Splicing as Part of a Large Multiprotein Complex LASR. *Cell* **165**, 606–619.

Gao, C., Ren, S., Lee, J.-H., Qiu, J., Chapski, D. J., Rau, C. D., Zhou, Y., Abdellatif, M., Nakano, A., Vondriska, T. M., et al. (2016). RBFOX1-mediated RNA splicing regulates cardiac hypertrophy and heart failure. *J Clin Invest* **126**, 195–206.

Nikonova, E., Kao, S.-Y. and Spletter, M. L. (2020). Contributions of alternative splicing to muscle type development and function. *Semin. Cell Dev. Biol.*

Nongthomba, U., Cummins, M., Clark, S., Vigoreaux, J. O. and Sparrow, J. C. (2003). Suppression of muscle hypercontraction by mutations in the myosin heavy chain gene of *Drosophila melanogaster*. *Genetics* **164**, 209–222.

- Schnorrer, F., Schönbauer, C., Langer, C. C. H., Dietzl, G., Novatchkova, M., Schernhuber, K., Fellner, M., Azaryan, A., Radolf, M., Stark, A., et al.** (2010). Systematic genetic analysis of muscle morphogenesis and function in *Drosophila*. *Nature* **464**, 287–291.
- Shukla, J. P., Deshpande, G. and Shashidhara, L. S.** (2017). Ataxin 2-binding protein 1 is a context-specific positive regulator of Notch signaling during neurogenesis in *Drosophila melanogaster*. *Development* **144**, 905–915.
- Usha, N. and Shashidhara, L. S.** (2010). Interaction between Ataxin-2 Binding Protein 1 and Cubitus-interruptus during wing development in *Drosophila*. *Dev Biol* **341**, 389–399.
- Wei, C., Xiao, R., Chen, L., Cui, H., Zhou, Y., Xue, Y., Hu, J., Zhou, B., Tsutsui, T., Qiu, J., et al.** (2016). RBFox2 Binds Nascent RNA to Globally Regulate Polycomb Complex 2 Targeting in Mammalian Genomes. *Mol Cell* **62**, 875–889.

December 20, 2021

RE: Life Science Alliance Manuscript #LSA-2021-01342-T

Dr. Maria Lynn Spletter
Ludwig-Maximilians-Universität München
Physiological Chemistry
Großhaderner Str. 9
Martinsried-Planegg, Bayern 82152
Germany

Dear Dr. Spletter,

Thank you for submitting your revised manuscript entitled "Rbfox1 is required for myofibril development and maintaining fiber-type specific isoform expression in *Drosophila* muscles". We would be happy to publish your paper in Life Science Alliance pending final revisions necessary to meet our formatting guidelines.

- please add a Running Title to our system
- please add a Summary Blurb/Alternate Abstract and a Category in our system
- please add the Twitter handle of your host institute/organization as well as your own or/and one of the authors in our system
- please add your main, supplementary figure, and table legends to the main manuscript text after the references section; all figure legends should only appear in the main manuscript file
- please add Author Contributions to our system
- supplemental references should be part of the main references
- please add callouts for Figures 6 B, C, E, G, I, M; S1F-G, K-L; S2L; S6A-B, E-F, Q to your main manuscript text
- please indicate molecular weight next to each protein blot
- The References listed in Supplemental Information should be incorporated into the main Reference list

A. FINAL FILES:

B. MANUSCRIPT ORGANIZATION AND FORMATTING:

Sincerely,

December 23, 2021

RE: Life Science Alliance Manuscript #LSA-2021-01342-TR

Dr. Maria Lynn Spletter
Ludwig-Maximilians-Universität München
Physiological Chemistry
Großhaderner Str. 9
Martinsried-Planegg, Bayern 82152
Germany

Dear Dr. Spletter,

Thank you for submitting your Research Article entitled "Rbfox1 is required for myofibril development and maintaining fiber-type specific isoform expression in *Drosophila*". It is a pleasure to let you know that your manuscript is now accepted for publication in Life Science Alliance. Congratulations on this interesting work.

DISTRIBUTION OF MATERIALS:

Again, congratulations on a very nice paper. I hope you found the review process to be constructive and are pleased with how the manuscript was handled editorially. We look forward to future exciting submissions from your lab.

Sincerely,
